# Bivariate Causal Discovery with Proxy Variables: Integral Solving and Beyond

Yong Wu [1 2 3 4]  Yanwei Fu [5]  Shouyan Wang [1 2 3 4 6 7]  Xinwei Sun ✉ [5]

## Abstract

Bivariate causal discovery is challenging when unmeasured confounders exist. To adjust for the bias, previous methods employed the proxy variable (*i.e.*, negative control outcome (NCO)) to test the treatment-outcome relationship through integral equations – and assumed that violation of this equation indicates the causal relationship. Upon this, they could establish asymptotic properties for causal hypothesis testing. However, these methods either relied on parametric assumptions or required discretizing continuous variables, which may lead to information loss. Moreover, it is unclear when this underlying integral-related assumption holds, making it difficult to justify the utility in practice. To address these problems, we first consider the scenario where only NCO is available. We propose a novel non-parametric procedure, which enjoys asymptotic properties and preserves more information. Moreover, we find that when NCO affects the outcome, the above integral-related assumption may not hold, rendering the causal relation unidentifiable. Informed by this, we further consider the scenario when the negative control exposure (NCE) is also available. In this scenario, we construct another integral restriction aided by this proxy, which can discover causation when NCO affects the outcome. We demonstrate these findings and the effectiveness of our proposals through comprehensive numerical studies.

## 1. Introduction

When the causal sufficiency is violated (Spirtes et al., 2001), constraint-based causal discovery is challenging due to the potential bias introduced by unmeasured confounders in conditional independence testing. As a foundational step, this paper focuses on the bivariate case, aiming to identify causal relationships in the presence of unobserved variables. To adjust for the bias, (Miao et al., 2018) leveraged observed proxy variables (Kuroki & Pearl, 2014; Tchetgen et al., 2024) to examine the existence of causal relations over discrete variables. Later, (Miao et al., 2023; Liu et al., 2023) extended them to continuous variables, by investigating the existence of solutions to integral equations.

Specifically, the integral equation examines whether the relationship between the treatment and the outcome can be fully explained by the proxy, *a.k.a*, negative control outcome (NCO) (Lipsitch et al., 2010). If there is no causal relation, the relationship remains consistent across changes in treatment. Upon this, (Miao et al., 2023) tested the causal null hypothesis by goodness-of-fit, which relied on parametric assumptions. To allow for nonparametric testing, (Liu et al., 2023) proposed a discretization approach to approximate the equation, and established asymptotic properties. However, it may require a large number of bins and substantial data to control the approximation error.

To address these issues when only the NCO is available, we propose a non-parametric testing procedure based on a kernel estimator called *Proxy Maximum Characteristic Restriction* (PMCR), to examine the integral equation. Without discretization, our approach preserves more information from the original data. After computing the least-square residues via PMCR, we construct the statistics and establish its asymptotic properties. Compared to other first-order moment restriction methods (Mastouri et al., 2021), our procedure leverages the characteristic function to capture all order moments, offering greater power in identifying the causal relation.

Nevertheless, we proceed to note that, regarding the power analysis, both our method and others, are built upon a basic assumption that the integral equation does not hold when the causal relation exists. However, by studying the solvability of the integral equation, we surprisingly find that this assumption may not hold, if the NCO's effect on the outcome

---

[1]Institute of Science and Technology for Brain-Inspired Intelligence, Fudan University [2]Key Laboratory of Computational Neuroscience and Brain-Inspired Intelligence (Fudan University), Ministry of Education [3]State Key Laboratory of Medical Neurobiology and MOE Frontiers Center for Brain Science, Fudan University [4]Zhangjiang Fudan International Innovation Center [5]School of Data Science, Fudan University [6]Shanghai Engineering Research Center of AI & Robotics, Fudan University [7]Engineering Research Center of AI & Robotics, Ministry of Education, Fudan University. Correspondence to: Xinwei Sun <sunxinwei@fudan.edu.cn>.

*Proceedings of the $42^{nd}$ International Conference on Machine Learning*, Vancouver, Canada. PMLR 267, 2025. Copyright 2025 by the author(s).

is strong enough to account for the effect of the treatment, making it fail to discover the causal relation with only the NCO. This inspires us to leverage the additional negative control exposure (NCE), that was commonly used in causal inference (Miao et al., 2018; Tchetgen et al., 2024). In this scenario, we introduce another integral equation aided by such NCE, which can effectively discriminate the alternative hypothesis from the null, thereby enabling causal identification when NCO has a strong effect on the outcome.

To demonstrate our findings and the effectiveness of our approach, we test it in two scenarios–one in which the NCO influences the outcome and one in which it does not. We find that the procedure using only NCO performs well when the NCO has no effect on the outcome, but fails when it does influence the outcome. In this case, our procedure that additionally leverages the NCE can successfully discover the causal relationship.

We summarize our contributions as follows:

1. We propose a non-parametric procedure to solve the integral equation, which is more efficient and effective in learning causal relations when only NCO is available.
2. We study the solvability of the integral equation and find that it may not be able to identify the causal relation when the NCO's effect on the outcome is strong enough.
3. When NCE is available, we leverage it to construct another integral restriction, which can recover causal relationships when NCO strongly affects the outcome.
4. We thoroughly verify our findings, utility, and effectiveness of our proposed methods on synthetic data.

## 2. Related works

To adjust for the bias caused by unmeasured confounding, (Miao et al., 2018) leveraged proxy variables (Kuroki & Pearl, 2014; Tchetgen et al., 2024) to test the causal null hypothesis over discrete variables. Later, (Miao et al., 2023; Liu et al., 2023) extended their procedure to continuous variables by examining the integral equation. While their methods established asymptotic properties with only a single proxy, their methods either relied on parametric assumptions or the discretization process that may lead to information loss. To address these issues, we propose a novel non-parametric procedure to examine the integral equation without discretization, hence is more sample efficient in learning causal relationships when only the NCO is available. **Additionally**, by comprehensively studying the solvability of the integral equation, we surprisingly find that previous methods may not be able to identify the causal relation when NCO influences the outcome. When the NCE is available, we further construct another integral equation that can discriminate the alternative hypothesis from the null, enabling causal identification when NCO has a strong influence on the outcome.

## 3. Preliminary

**Problem setup.** We consider the problem of testing whether the causal relation $X \to Y$ exists, under the unmeasured confounder $U$. Under Markovian and faithfulness conditions, this is equivalent to testing the causal null hypothesis $\mathbb{H}_0 : X \perp\!\!\!\perp Y|U$. To adjust for the confounding bias, we assume the availability of a proxy variable $W$ such that $X \perp\!\!\!\perp W|U$ (Kuroki & Pearl, 2014), which also serves as the negative control outcome in causal inference. Fig. 1 (a) shows the causal diagram over $X, Y, U, W$. Besides, in some scenarios, we may have access to an additional proxy variable $Z$ (*i.e.*, negative control exposure), which satisfies $Z \perp\!\!\!\perp (W, Y)|\{U, X\}$ as illustrated in Fig. 1 (b). Throughout, we assume $X, Y, U, W, Z$ are continuous variables.

**Notations.** Suppose $X, Y, U, W, Z$ are continuous random variables over the probability space $(\Omega, \mathcal{F}, \mathbb{P})$, with domains $\mathcal{X}, \mathcal{Y}, \mathcal{U}, \mathcal{W}, \mathcal{Z}$, respectively. For any variable $U$, we denote $\mathcal{L}^2\{F(u)\}$ as the space of square-integrable functions with respect to the cumulative distribution function $F(u)$. For any space $\mathcal{W}$, let $k_W$ be its positive semi-definite kernel. We denote $\phi_W$ as its associated canonical feature map, *i.e.*, $\phi_W(w) := k_W(w, \cdot)$ for any $w \in \mathcal{W}$. Besides, we denote $\mathcal{H}_W$ as the corresponding reproducing kernel Hilbert space (RKHS). For any operator $A : \mathcal{H}_W \to \mathcal{H}_X$, we denote $A^*$ as its adjoint operator. For any discrete variables $X, Y$ with respectively $i, j$ categories, we denote $\mathbb{P}(y|X) := \{p(y|x_1), ..., p(y|x_i)\}$, the probability matrix $\mathbb{P}(Y|X) := \{p(y_1|X)^\top, ..., p(y_j|X)^\top\}^\top$.

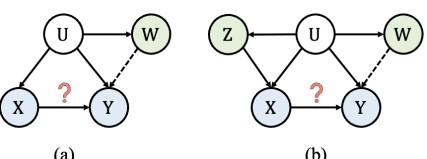

(a)            (b)

*Figure 1.* Causal diagrams over $X, Y, U, W, Z$. $W$ (*resp. Z*) denote the negative control outcome (*resp.* exposure). The dotted line indicates its potential presence or absence.

**Previous studies and limitations.** For discrete variables, (Miao et al., 2018) proposed to test $\mathbb{H}_0$ by examining whether $\mathbb{P}(W|Z, x)$ can fully explain the variability of $\mathbb{P}(y|Z, x)$ as $x$ varies. Specifically, given $X, Z, W$ with respectively $i, j, k$ categories, they performed a linear regression of $q_y := \{\mathbb{P}(y|Z, x_1), ..., \mathbb{P}(y|Z, x_i)\}^\top$ on $Q^\top := \{\mathbb{P}(W|Z, x_1), ..., \mathbb{P}(W|Z, x_i)\}^\top$, and tested the linearity based on the least-square residues. If $Q^\top$ is the full-column rank when $ij > k$, they derived the null-limiting distribution of the statistics based on these residues.

Later, (Liu et al., 2023) noticed that the rank constraint satisfies when $i > k$, allowing to test the linearity between $\mathbb{P}(y|X)$ and $\mathbb{P}(W|X)$ without $Z$. Inspired by this, they extended to continuous variables by discretizing $X, Y, W$

and performed the linearity testing. While it could establish asymptotic properties for hypothesis testing, the method was constrained by the need for both a large number of bins and a substantial sample size to ensure accurate probability estimation. Similarly, (Miao et al., 2023) used goodness-of-fit to examine the following integral equation:

$$p(y|x) = \int h(w,y)p(w|x)dw \text{ for some } h(w,y), \quad (1)$$

which serves as the theoretical foundation of the discretization approach in (Liu et al., 2023). However, its goodness-of-fit procedures relied on the parametric assumption.

*Does* (1) *holds under* $\mathbb{H}_1$? To discover the causal relationship, these methods assumed that (1) does not hold under $\mathbb{H}_1$. However, it is unclear when this condition holds, making it hard to justify the validity during practical implementations.

**Overview of our contributions.** To address problems of existing works when only $W$ is available, Sec. 4 propose a novel non-parametric approach based on *Proxy Maximum Characteristic Restriction* (PMCR), which can test (1) without discretization, thereby avoiding information loss in causal discovery.

Further, through comprehensively studying the solvability of (1) in Sec. 5, we find that when $W \to Y$ (the dotted edge in Fig. 1 (a)), the above integral may have a solution even under $\mathbb{H}_1$, making all procedures fail to discover the causal relation using only $W$. This inspires us to consider the scenario where the proxy $Z$ is available, as commonly adopted in causal inference (Tchetgen et al., 2024). In this scenario, Sec. 6 leveraged $Z$ to construct another integral equation to examine the solution's property that involves $U$. Thanks to this auxiliary information, we can discover the causal relation that may not be identifiable with only $W$.

## 4. Hypothesis testing with a single proxy

We first consider the scenario when only the proxy $W$ (*i.e.* NCO) is available. To test the causal null hypothesis, we propose to examine whether the integral equation (1) exists. To this end, Sec. 4.1 first shows that under $\mathbb{H}_0$, the solution exists under some completeness conditions. To solve the equation, Sec. 4.2 introduces a novel estimation method and constructs the testing statistics. Sec. 4.3 establishes its asymptotic level and power for such statistics. Finally, Sec. 4.4 summarizes the algorithm for implementations.

### 4.1. Solution existence under the null hypothesis

We show that $\mathbb{H}_0$ can be tested by examining the integration equation (1), which can hold under $\mathbb{H}_0$ as the absence of direct effect from $X$ to $Y$ allows $p(w|x)$ to fully explain away the variability of $p(y|x)$. To formally claim this statement, we require the completeness condition.

**Assumption 4.1** (Completeness of $\mathbb{P}(U|W)$). For any square-integrable function $g$, we assume $\mathbb{E}\{g(u)|w\} = 0$ almost surely if and only if $g(u) = 0$ almost surely.

Completeness 4.1 is a standard assumption in causal hypothesis testing (Miao et al., 2018; 2023; Liu et al., 2023). This condition is widely applicable, as shown by examples provided in (Newey & Powell, 2003; D'Haultfoeuille, 2011; Hu & Shiu, 2018; Andrews, 2017). Here, it means $W$ carries all the variability of $U$. When $W, U$ are discrete with $i, j$ categories $(i > j)$, it reduces to the full-column rank of $\mathbb{P}(W|U)$, as used in (Liu et al., 2023) for identification.

**Proposition 4.2.** *Under assumption 4.1 and some regularity conditions in B.1, there exists a $g(w,u) \in \mathcal{L}^2\{F(w)\}$ for all $u$, such that it solves the following integral equation for all $(u,x)$:*

$$p(u|x) = \int g(w,u)p(w|x). \quad (2)$$

*Remark* 4.3. When $W, X, U$ are discrete, this result reduces to $\mathbb{P}(U|X) = \mathbb{P}(W|U)^{-1}\mathbb{P}(W|X)$ to establish linear relation under $\mathbb{H}_0$ (Liu et al., 2023).

By $p(y|x) = \int p(y|u)p(u|x)du$ under $\mathbb{H}_0$, $h(w,y) := \int g(w,u)p(y|u)du$ solves (1). To further ensure $h(w,y)$ is square integrable, we require the bounded likelihood ratio condition (Kato et al., 2021) for $p(u|y)/p(u)$, which holds except for the extreme dependency between $U$ and $Y$.

**Assumption 4.4** (Bounded likelihood ratio). There exists $C > 0$ such that $0 < \frac{p(u|y)}{p(u)} \leq C$ almost surely for $u$ and $y$.

Under these conditions, Theorem 4.5 establishes the existence of a solution to the integral equation in (1).

**Theorem 4.5.** *Suppose assumptions 4.1, 4.4 and the regularity condition B.1 hold. If $g(w,u)$ in (2) satisfies $\iint |\frac{g(w,u)}{p(u)}|^2 p(w)p(u)dwdu < \infty$, then under $\mathbb{H}_0$, there exists a $h(w,y) \in \mathcal{L}^2\{F(w)\}$ for all $y$, such that it solves the integral equation in (1) for all $(x,y)$.*

*Remark* 4.6. Condition $g(W,U)/p(U) \in \mathcal{L}^2\{F(w)F(u)\}$ imposes a regularity requirement on the solution $g(W,U)$ in (2). In Appx. B, we show that this condition easily holds under linear models.

This result suggests that we can reject $\mathbb{H}_0$ when the discrepancy between $p(y|u,x)$ and $p(y|u)$ is sufficiently large to make $p(w|x)$ fails to account for all the variability encoded in $p(y|x)$. It has been similarly employed for testing $\mathbb{H}_0$ (Miao et al., 2023; Liu et al., 2023). However, (Miao et al., 2023) additionally required the equivalence condition for identification, which may not hold beyond factor models.

In particular, (Liu et al., 2023) proposed a discrete approximation of (1) and tested the linearity between $\mathbb{P}(y|X)$ and $\mathbb{P}(W|X)$. However, this method may suffer from information loss due to the discretization. In the subsequent section, we propose a novel estimation method to solve the integral equation for testing.

## 4.2. Testing statistics via integral solving

In this section, we construct the test statistics to examine whether the solution exists in (1). To this end, we propose to solve the equation and measure its residue. When the equation (1) holds, previous studies proposed *Maximum Moment Restriction* (MMR) (Mastouri et al., 2021; Kallus et al., 2021) for integral solving. In our scenario, it involves solving $\overline{h}(W)$ from the following moment restriction:

$$\mathbb{E}_{Y,W}\left\{Y - \overline{h}(W)|X\right\} = 0. \tag{3}$$

However, as it only leverages the first-order moment information, it will lose the testing power, as illustrated by example B.3 in Appx. B, where (3) holds under $\mathbb{H}_1$. To address this, we propose a novel estimation method, which leverages the information of all order moments.

**Proxy Maximum Characteristic Restriction.** To test whether $p(y|x)$ equals to $\int h(w,y)p(w|x)dw := q(y|x)$, we consider the following equation:

$$\mathbb{E}_{Y,W}\{\varphi(Y,t) - H(W,t)|X\} = 0 \,\forall\, t \in \mathcal{T}, \tag{4}$$

where we choose $\varphi(Y,t)$ to contain more information than the first-order moment (Stinchcombe & White, 1998), and set $H(W,t)$ as $\int \varphi(y,t)h(w,y)p(y|x)dy$ to make (4) holds. A common choice for $\varphi(Y,t)$ is $\exp(ity)$, where $\mathcal{T}$ can be an arbitrarily chosen neighborhood around 0. In this case, $\mathbb{E}_Y\{\varphi(Y,t)\}$ is the characteristic function, and we hence call (4) the *Proxy Maximum Characteristic Restriction*. Since the characteristic function can uniquely determine the probability density and therefore all order moments, solving (4) offers greater utility to identify causal relations. In practice, we can set $\varphi(Y,t) = \sin(ty)$ and $\cos(ty)$, and test whether (4) holds for these choices.

Corollary 4.7 shows that $H(w,t)$ belongs to $\mathcal{L}^2\{F(w)\}$ for all $t$, which ensures that (4) is solvable.

**Corollary 4.7.** *Suppose conditions in Theorem 4.5 hold. For any $t$, $H(w,t)$ in (4) exists and belongs to $\mathcal{L}^2\{F(w)\}$.*

As demonstrated by (Horowitz, 2012), achieving uniform consistency in testing the existence of a solution to the conditional equation is impossible. Therefore, certain smoothness conditions must be imposed to ensure the feasibility of solving the equation. Following existing studies (Mastouri et al., 2021; Ghassami et al., 2022), we solve the solution in the reproducing kernel Hilbert space (RKHS) denoted by $\mathcal{H}_W$.

Formally, let $k_W$ be the reproducing kernel for the RKHS $\mathcal{H}_W$. By the spectral theorem, we can rewrite $k_W(w,w')$ in terms of the eigenvalues and continuous eigenfunctions as $k_W(w,w') = \sum_{j=1}^{\infty} \eta_j \varphi_j(w)\varphi_j(w')$, where $\{\varphi_j\}_j$ is the orthonormal basis of $\mathcal{L}^2\{F(w)\}$. We can therefore characterize $\mathcal{H}_W$ as $\mathcal{H}_W := \left\{H \in \mathcal{L}^2\{F(w)\} \middle| \sum_{i=1}^{\infty} \frac{\langle H,\varphi_i\rangle^2_{\mathcal{L}^2\{F(w)\}}}{\eta_i} < \infty\right\}$.

**Assumption 4.8** (Smoothness). For $H(W,t)$ in (4), we assume $H(W,t) \in \mathcal{H}_W$ for all $t$. This implies that there exists a solution within the RKHS that satisfies (4).

To solve $H(w,t)$ from (4), we employ recently developed nonparametric methods designed to estimate such conditional restrictions (Zhang et al., 2020; Mastouri et al., 2021; Kallus et al., 2021; Ghassami et al., 2022). Unlike these methods, we do not require the completeness of $W|X$ to ensure the uniqueness of the solution. This distinction arises because most of these methods focus on causal inference, where the goal is to accurately identify the bridge function to compute the causal effect. In contrast, our objective is to determine whether a solution exists. Therefore, it suffices for our estimate to approximate any valid solution to achieve this goal. In this paper, we estimate the least norm solution:

$$H^0(W,t) := \underset{H(W,t)\in\mathcal{H}_{W,0}}{\arg\min} \|H(W,t)\|_{\mathcal{H}_W},$$

where $\mathcal{H}_{W,0} \subset \mathcal{H}_W$ contains all solutions in (4). We leave more details in Appx. C.4.

Equivalently speaking, given RKHS $\mathcal{H}_W$ and $\mathcal{H}_X \subset \mathcal{L}^2\{F(x)\}$ with kernels $k_W$ and $k_X$, respectively, the goal is to solve $H(W,t)$ from $AH(\cdot,t) = b(\cdot,t)$, where $A : \mathcal{H}_W \mapsto \mathcal{H}_X$ is a compact operator such that $AH(W,t)(\cdot) := \mathbb{E}\{H(W,t)\phi_X(X)\}$, and $b(\cdot,t) := \mathbb{E}\{\varphi(Y,t)\phi_X(X)\}$. To solve $H$, we first note that (4) means, for any $g \in \mathcal{H}_X$, we have $\mathbb{E}_{Y,W,X}\left[\{\varphi(Y,t) - H(W,t)\}g(X)\right] = 0$ for almost all $t$. Therefore, similar to (Mastouri et al., 2021), we take $g$ over a unit-ball of $\mathcal{H}_X$, and minimizes:

$$R(H) = \sup_{g\in\mathcal{H}_X, \|g\|_{\mathcal{H}_X}\leq 1} \left(\mathbb{E}\left[\{\varphi(Y,t) - H(W,t)\}g(X)\right]\right)^2.$$

Let $\Delta(W,Y,t) := \varphi(Y,t) - H(W,t)$. (Mastouri et al., 2021) provided an equivalent form of the risk:

$$R(H) = \mathbb{E}\{\Delta(W,Y,t)\Delta(W',Y',t)k_X(X,X')\},$$

where $X',Y',W'$ are independent copies of $X,Y,W$. Further, (Zhang et al., 2020) demonstrated that under some conditions for $k_X$, minimizing $R(H)$ ensures us to find the solution. To implement, we propose to minimize the regularized empirical risk:

$$\widehat{R}^\lambda(H) := \sum_{i,j=1}^{n} \frac{\Delta_i\Delta_j}{n^2}K_{X,ij} + \lambda\|H\|_{\mathcal{H}_W}, \tag{5}$$

where $\Delta_i := \varphi(y_i,t) - H(w_i,t)$ and $K_{X,ij} := k_X(x_i,x_j)$. Using the representer theorem (Schölkopf et al., 2001), the estimated function $\widehat{H}^\lambda(w,t)$ for a fixed $t$ can be written as $\widehat{H}^\lambda(w,t) = \boldsymbol{\alpha}^\top \boldsymbol{k}_W(w)$, where $\boldsymbol{k}_W(w) := \{k_W(w_i,w)\}_i \in \mathbb{R}^n$ and the coefficient $\boldsymbol{\alpha}$ is given by $\boldsymbol{\alpha} := (K_W K_X K_W + n^2\lambda K_X)^{-1}K_X K_W \varphi(\boldsymbol{y},t)$. Here, $K_X := \{k_X(x_i,x_j)\}_{ij} \in \mathbb{R}^{n\times n}$ and $K_W :=$

$\{k_W(w_i, w_j)\}_{ij} \in \mathbb{R}^{n \times n}$ are Gram matrices, and $\varphi(\boldsymbol{y}, t) := (\varphi(y_1, t), \ldots, \varphi(y_n, t))^\top$. The optimal $\lambda$ is chosen via cross-validation.

**Constructing the testing statistics.** Ideally, if $\widehat{H}^\lambda$ can well approximate the solution, the equation (4) approximately holds for $\widehat{H}^\lambda$. Therefore, we can assess the validity of $\mathbb{H}_0$ by evaluating the residue of the equation. To this end, we employ the conditional moment test procedure (Bierens, 1982; Bierens & Ploberger, 1997).

To enhance the power, we choose a weight function $m(\cdot, s)$ that transforms the conditional restriction to unconditional one. Commonly chosen weight function includes characteristic function, exponential function, sine and cosine functions. According to (Stinchcombe & White, 1998), these functions ensure that for any $U(W, Y, t) := \varphi(Y, t) - H^0(W, t)$ with $\mathbb{E}\{U(W, Y, t)|X\} \neq 0$, the set of $s \in \mathcal{T}$ such that $\mathbb{E}\{U(W, Y, t)m(X, s)\} = 0$ has Lebesgue measure zero. Let $\widehat{U}(W, Y, t) := \varphi(Y, t) - \widehat{H}^\lambda(W, t)$, we define

$$T_n(s, t) = \frac{1}{\sqrt{n}} \sum_{i=1}^{n} \widehat{U}(w_i, y_i, t)m(x_i, s), \ s, t \in \mathcal{T}. \quad (6)$$

Since testing (4) for all $t \in \mathcal{T}$ is equivalent to evaluating the maximum of the residuals over $\mathcal{T}$, we define the final statistics for testing $\mathbb{H}_0$ as:

$$\Delta_{\varphi, m} = \max_{t \in \mathcal{T}} \int_{\mathcal{T}} |T_n(s, t)|^2 d\mu(s), \quad (7)$$

where $\mu$ denotes the measure of $\mathcal{T}$ (*e.g.*, Lebesgue measure).

### 4.3. Asymptotic behavior

We study asymptotic properties of $\Delta_{\varphi, m}$. We first introduce some regularity conditions.

**Assumption 4.9.** We assume $\mathbb{E}_X\{m(X, s)|W\}$ and $\mathbb{E}_X\{|m(X, s)|^2|W\}$ are uniformly bounded for all $s$.

**Assumption 4.10.** $n\lambda \to \infty, n\lambda^2 \to 0$.

**Assumption 4.11.** For any $s, t \in \mathcal{T}$, $\mathbb{E}\{U(W, Y, t)^4|X\} < \infty$ and $\mathbb{E}(|m(X, s) - \{A(A^*A)^{-1}g_s\}(X)|^4) < \infty$, where $g_s(\cdot) := \mathbb{E}[m(X, s)\phi_W(W)](\cdot)$.

Assumptions 4.9–4.10 are standard in kernel estimation methods (Darolles et al., 2011; Babii & Florens, 2020; Beyhum et al., 2024). Asm. 4.9 imposes regularity conditions on the weight function $m$, while Asm. 4.10 ensures that the regularization bias vanishes asymptotically. Additionally, Asm. 4.11 is required to control the asymptotic variance of the test statistic, which has been similarly assumed in (Huang et al., 2022).

**Theorem 4.12.** *Let* $\eta_{s,t}(O) := U(W, Y, t)m(X, s) - U(W, Y, t)\{A(A^*A)^{-1}g_s\}(X)$*, with* $O := (W, Y, X)$*. Suppose assumptions 4.9–4.11, C.2–C.4, and D.1–D.2 hold. Under* $\mathbb{H}_0$*, we have (i).* $T_n(s, t)$ *converges weakly to* $\mathbb{G}(s, t)$

such that $\iint |\mathbb{G}(s, t)|^2 d\mu(s)d\mu(t) < \infty$, where $\mathbb{G}(s, t)$ is a Gaussian process with zero-mean and covariance:

$$\Sigma\{(s, t), (s', t')\} = \mathbb{E}\{\eta_{s,t}(O)\eta_{s',t'}(O'))\},$$

*where* $O' := (W', Y', X')$ *is an independent copy of* $O$*; and (ii).* $\Delta_{\varphi, m}$ *converges weakly to* $\max_{t \in \mathcal{T}} \int |\mathbb{G}(s, t)|^2 d\mu(s)$.

*Remark* 4.13. For brevity, we only present asymptotic results for $T_n(s, t)$ being a real-valued function, or as the real and imaginary parts of a complex-valued function, although they can be trivially extended to complex-valued functions.

**Power analysis.** We consider the power performance under two alternatives, under which (4) has no solution. First, we consider the global alternative that has been similarly considered in proximal causal discovery (Liu et al., 2023). That is, for any $H(w, t) \in \mathcal{H}_W$ for all $t$, the global alternative $\mathbb{H}_1^{\text{fix}}$ satisfies the following:

$$\mathbb{H}_1^{\text{fix}} : \mathbb{E}\{\varphi(Y, t) - H(W, t)|X\} \neq 0 \text{ for some } t \in \mathcal{T}.$$

Besides, we consider a sequence of local alternatives $\mathbb{H}_{1n}^\alpha$. There exists $H^0(w, t) \in \mathcal{H}_W$ for all $t$, such that:

$$\mathbb{H}_{1n}^\alpha : \mathbb{E}\{\varphi(Y, t)|X\} = \mathbb{E}\{H^0(W, t)|X\} + \frac{r(X, t)}{n^\alpha}, \ \forall t$$

where $0 < \alpha \leq \frac{1}{2}$ and $r(X, t) \in \mathcal{H}_X$. To be a valid alternative, $r(X, t)/n^\alpha$ can not be written as $\mathbb{E}\{H - H^0|X\}$ for any $H \in \mathcal{H}_W$. Theorem 4.14 suggests that our statistics has asymptotic power of one under $\mathbb{H}_1^{\text{fix}}$ and $\mathbb{H}_{1n}^\alpha$ when $\alpha < \frac{1}{2}$, and has nontrivial power when $\alpha = \frac{1}{2}$.

**Theorem 4.14.** *Suppose assumptions in Theorem 4.12 hold. Besides, we assume* $\mathbb{E}\{r(X, t)^4\} < \infty$*. Then, we have:*

  (i) *Global alternative.* $\lim_{n \to \infty} \max_{t \in \mathcal{T}} |T_n(s, t)| = \infty$ *for almost all $s$ under* $\mathbb{H}_1^{\text{fix}}$*.*
  (ii) *Local alternative ($\alpha < 1/2$).* $\lim_{n \to \infty} \max_{t \in \mathcal{T}} |T_n(s, t)| = \infty$ *for almost all $s$ under* $\mathbb{H}_{1n}^\alpha$*.*
  (iii) *Local alternative ($\alpha = 1/2$).* $T_n(s, t)$ *converges weakly to* $\mathbb{G}(s, t) + \mu(X, s, t)$ *such that* $\iint |\mathbb{G}(s, t) + \mu(X, s, t)|^2 d\mu(s)d\mu(t) < \infty$ *under* $\mathbb{H}_{1n}^\alpha$*, where* $\mathbb{G}(s, t)$ *is defined in Theorem 4.12 and* $\mu(X, s, t) := \mathbb{E}\left[\{r(X, t) - (A^*A)^{-1}A^*r(X, t)\}m(X, s)\right]$*.*

### 4.4. Implementations

To implement the testing, we need to compute the statistics $\Delta_{\varphi, m}$ in (7) and determine the critical value.

**Computing $\Delta_{\varphi, m}$.** Since $\Delta_{\varphi, m}$ involves the integration, generally we should employ Monte-Carlo methods for approximation. For computational convenience, we can set $m(\cdot, s)$ to the characteristic function and $\mu$ to be symmetric around the origin (e.g., Lebesgue measure). Such choices enable the integration to be computed in closed form. Besides, according to (Stinchcombe & White, 1998), this choice of

$m$ can preserve power when transforming the conditional restriction to the unconditional one. To compute the maximal value of $\int_{\mathcal{T}} |T_n(s,t)|^2 d\mu(s)$ over $\mathcal{T}$, we evaluate the process at a grid of equi-distant indices $\{t_i, i \in [K]\}$ and estimate $\widehat{\Delta}_{\varphi,m} := \max_{k \in [K]} \int_{\mathcal{T}} |T_n(s, t_k)|^2 d\mu(s)$. Corollary 4.16 shows that when $K$ is sufficiently large, $\widehat{\Delta}_{\varphi,m}$ converges to $\max_{t \in \mathcal{T}} \int_{\mathcal{T}} |\mathbb{G}(s,t)|^2 d\mu(s)$.

*Remark* 4.15. If $\widehat{U}(W, Y, t)$ is a complex function, we can respectively compute $\mathrm{Re}(\widehat{\Delta}_{\varphi,m})$ and $\mathrm{Im}(\widehat{\Delta}_{\varphi,m})$. Our test statistic is given by $\widehat{S} := \max\{\mathrm{Re}(\widehat{\Delta}_{\varphi,m}), \mathrm{Im}(\widehat{\Delta}_{\varphi,m})\}$.

**Critical value.** Since it is difficult to obtain the explicit form of $\mathbb{G}(s,t)$, we employ the residue-based wild bootstrap procedure for approximation under the null-limiting distribution. We repeat the procedure for $B$ times. For the $b$-th time, we first employ the empirical process $\widehat{T}_n^b(s,t) = \frac{1}{\sqrt{n}} \sum_{i=1}^n \omega_i^b \widehat{U}(w_i, y_i, t) m(x_i, s)$ to approximate $T_n(s,t)$ for each $(s,t)$, where $\{\omega_i^b\}_{i=1}^n$ is a sequence of zero-mean, unit variance variables. Here, we follow (Mammen, 1993) to set $\mathbb{P}(\omega_i = 1 - \kappa) = \kappa/\sqrt{5}$ and $\mathbb{P}(\omega_i = \kappa) = 1 - \kappa/\sqrt{5}$ with $\kappa = \frac{\sqrt{5}+1}{2}$. The bootstrapped statistic is given by:

$$\widehat{\Delta}_{\varphi,m}^b = \max_{k \in [K]} \int_{\mathcal{T}} |\widehat{T}_n^b(s, t_k)|^2 d\mu(s). \tag{8}$$

Given the level of significance $\alpha$, the critical value is computed as the $(1-\alpha)$-quantile of $\left\{\widehat{\Delta}_{\varphi,m}^1, ..., \widehat{\Delta}_{\varphi,m}^B\right\}$, denoted by $\widetilde{\Delta}_{\varphi,m}^{1-\alpha}$. We then reject the null hypothesis if $\widehat{\Delta}_{\varphi,m} \geq \widetilde{\Delta}_{\varphi,m}^{1-\alpha}$. Corollary 4.16 shows that the bootstrap statistics $\widehat{\Delta}_{\varphi,m}^b$ converges to $\max_{t \in \mathcal{T}} \int_{\mathcal{T}} |\mathbb{G}(s,t)|^2 d\mu(s)$.

**Corollary 4.16.** *Suppose assumptions in Theorem 4.12 hold. If $\varphi(y,t)$ is continuous with respect to $t$ for each $y$, then $\widehat{\Delta}_{\varphi,m}$ is weakly convergent to $\max_{t \in \mathcal{T}} \int_{\mathcal{T}} |\mathbb{G}(s,t)|^2 d\mu(s)$ under $\mathbb{H}_0$, as $n, K \to \infty$. Besides, conditional on the original sample $\{y_i, w_i, x_i\}_{i=1}^n$, the bootstrapped statistics (8) is also weakly convergent to the $\max_{t \in \mathcal{T}} \int_{\mathcal{T}} |\mathbb{G}(s,t)|^2 d\mu(s)$.*

*Remark* 4.17. Many choices of $\varphi(y,t)$ satisfy the continuity condition, including characteristic function, sine and cosine functions, *etc*.

## 5. Nonidentifiability with only NCO

To identify the causal relation, our method and previous studies in proximal causal discovery (Miao et al., 2023; Liu et al., 2023) rely on the assumption that the integral equation (1) has no solution under $\mathbb{H}_1$. However, it remains unclear when this condition holds. To clarify this, we explore the condition under the linear model.

Proposition 5.1 suggests that the direct effect needs to be sufficiently large for the condition to hold. Additionally, we surprisingly find that when the effect from $W$ to $Y$ is strong enough, even if the effect from $X$ to $Y$ is also strong,

equation (1) will also have a solution, rendering the causal relationship non-identifiable.

**Proposition 5.1.** *Suppose $U, X, Y, W$ follow from the linear Gaussian model, i.e. $U = \varepsilon_U, X = \alpha_U U + \alpha_0 + \varepsilon_X, W = \beta_U U + \beta_0 + \varepsilon_W, Y = \gamma_U U + \gamma_X X + \gamma_X W + \gamma_0 + \varepsilon_Y$, where $\varepsilon_U, \varepsilon_X, \varepsilon_W, \varepsilon_Y \sim \mathcal{N}(0,1)$. When $\gamma_W = 0$, as long as $|\gamma_X| > g_X(\alpha_U, \beta_U, \gamma_U)$, the integration equation (1) has no solution. Further, if $|\gamma_W| > g_W(\alpha_U, \beta_U, \gamma_U)$[1], (1) has a solution.*

*Remark* 5.2. In the proof of Prop. 5.1, we also discuss when the solution exists under $\mathbb{H}_0$. We find that the confounding strength between $W$ and $U$, specifically $\beta_U$, must be sufficiently large to make the solution exist.

To illustrate, we consider example 5.3. We find that when $\gamma_W$ is sufficiently large, the solution exists even when $X \to Y$, leading to a significant drop of power as shown in Fig. 2.

**Example 5.3.** *Suppose that $X, Y, U, W$ satisfy the linear Gaussian model, i.e. $U = \varepsilon_U, X = 2U + \varepsilon_X, W = -2U + \varepsilon_W, Y = X + U + \gamma_W W + \varepsilon_Y$, where $\varepsilon_U, \varepsilon_Y, \varepsilon_W, \varepsilon_X \sim \mathcal{N}(0,1)$ and $W, X$ are standardized. The integral equation (1) has a solution if and only if $\gamma_W > \frac{-15+36\sqrt{5}}{72+16\sqrt{5}} \approx 0.61$.*

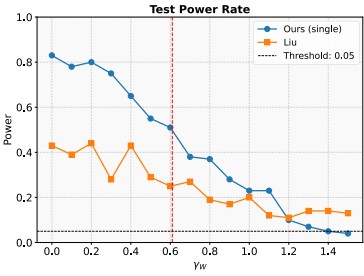

*Figure 2.* The change of power across $\gamma_W$ in example 5.3.

The result suggests that, when the effect from $W \to Y$ is strong enough, using only the NCO (*i.e.*, $W$) may fail to identify the causal relationship. In the next section, we show that when the additional proxy $Z$ (*i.e.*, NCE) is available, the identification becomes possible again.

## 6. Hypothesis testing with two proxies

To discover the causal relation when (1) has a solution under the alternative hypothesis, we assume $Z$ (*i.e.*, NCE) is available, which has been commonly employed in proximal causal inference (Miao et al., 2018; Cui et al., 2024). When $Z$ is available, we analyze the properties of the solution under the null hypothesis to identify the causal relation.

Specifically, when $h(w, y)$ satisfies (1), by $W \perp\!\!\!\perp X | U$, we have $p(y|x) = \int \left\{\int h(w, y)p(w|u)dw\right\} p(u|x)du$ for all $(x, y)$. We will examine the property of the solution $\int h(w, y)p(w|u)dw$ to test $\mathbb{H}_0$. To this end, we require the

---

[1]We leave the detailed form of $g_X, g_W$ in Appx. E.3.

following completeness condition.

**Assumption 6.1** (Completeness of $\mathbb{P}(U|X)$)**.** For any square-integrable function $g$, we assume $\mathbb{E}\{g(u)|x\} = 0$ almost surely if and only if $g(u) = 0$ almost surely.

This assumption was also made in (Liu et al., 2023; Miao et al., 2023). It ensures that for any $h(w, y)$ that satisfies (1), we must have

$$p(y|u) = \int h(w, y)p(w|u)dw \text{ under } \mathbb{H}_0. \quad (9)$$

**Testing the null hypothesis through** $Z$**.** To examine (9) that involves the unobserved confounder $U$, we employ $Z$ to introduce another restriction. We require the following completeness condition that is standard in proximal causal inference (Miao et al., 2018; Tchetgen et al., 2024).

**Assumption 6.2** (Completeness of $\mathbb{P}(U|Z, x)$)**.** For any $g \in \mathcal{L}^2\{F(u)\}$ and any $x$, we assume $\mathbb{E}\{g(u)|z, x\} = 0$ almost surely if and only if $g(u) = 0$ almost surely.

Taking expectation over $p(u|z, x)$ on both sides of (9), and obtain the following for any $x$:

$$p(y|z, x) = \int h(w, y)p(w|z, x)dw \quad (10)$$

for all $(y, z)$. Under assumption 6.2, we can check $\mathbb{H}_0$ by first solving $h$ from (1) and examine whether it satisfies $p(y|z, x) = \int h(w, y)p(w|z, x)dw$. We summarize it into the following theorem.

**Theorem 6.3.** *Suppose assumptions 6.1, 6.2 hold. For any $h(w, y)$ that satisfies (1), $\mathbb{H}_0$ holds if and only if $h(w, y)$ also satisfies the integral equation (10) for any fixed $x$.*

*Remark* 6.4. Solving $h(w, y)$ from (10) is different from that of $p(y|z, x) = \int h(w, y, x)p(w|z, x)dw$ in (Miao et al., 2018). The goal of (Miao et al., 2018) is to solve $h(w, y, x)$ for identifying $p\{y|do(x)\} = \int h(w, y, x)p(w)dw$, which thus allows $h$ to depend on $x$. In contrast, our goal is testing whether $X$ directly affects $Y$, so the solution $h$ should be independent of $X$ while ensuring that (10) hold as $x$ varies.

When $X$, $Z$, and $W$ are discrete, (Miao et al., 2018) proposed testing the linear relationship between $\{\mathbb{P}(y|Z, x_i)\}_i$ and $\{\mathbb{P}(W|Z, x_i)\}_i$ across the values of $Z, X$. Our procedure can be seen as the continuous counterpart to this approach, with the difference being that after solving $h(w, y)$ from (1) for all $x$, we only need to examine (10) for all $z$ with a single $x$, rather than solving $h(w, y)$ from (10) for all pairs of $(x, z)$.

*Remark* 6.5. One might argue that when $W, Z$ are available, the average causal effect is identifiable. In this regard, our analysis seems redundant as the effect of causal relation can be quantified. However, we would like to mention that the causal discovery conceptually differs from the causal

effect. In particular, Appx. E.2 provides a counterexample to illustrate that the causal relation may still exist even when there is no causal effect. A more comprehensive discussion is also provided in Appx. E.2.

Similar to Sec. C, if $\widehat{H}^\lambda$ can well approximate the solution of (4), the equation

$$\mathbb{E}_{Y,W}\{\varphi(Y, t) - H(W, t)|Z, x\} = 0 \,\forall\, t \in \mathcal{T}, \quad (11)$$

also approximately holds for all $t \in \mathcal{T}$. Therefore, we can assess the validity of $\mathbb{H}_0$ based on the residue $\widehat{U}$. Based on $\widehat{U}$, we can construct the statistics $T_n^{(Z)}(s, t), \Delta_{\varphi,m}^{(Z)}$, with the weight function $m(Z, x, s)$ over $Z$.

**Asymptotic behavior.** Theorem 6.6 establishes the weak convergence of $T_n^{(Z)}(s, t)$ and $\Delta_{\varphi,m}^{(Z)}$ under $\mathbb{H}_0$.

**Theorem 6.6.** *Denote* $\overline{\eta}_{s,t}(O, x) := U(W, Y, t)$ $[\{m(Z, x, s) - \{A(A^*A)^{-1}g_s\}(X, x)]$, *where* $g_s(\cdot, x) :=$ $\mathbb{E}\{m(Z, x, s)\phi_W(W)\}(\cdot)$ *and* $O := (W, Z, Y, X)$. *Suppose assumptions in Theorem 4.12 hold. If Asm. 6.1-6.2, and E.6-E.8 hold, under $\mathbb{H}_0$ we have, (i). $T_n^{(Z)}(s, t)$ converges weakly to $\mathbb{G}(s, t)$ s.t. $\iint |\mathbb{G}(s, t)|^2 d\mu(s)d\mu(t) < \infty$, where $\mathbb{G}(s, t)$ is a mean-zero Gaussian process with covariance $\Sigma\{(s, t), (s', t')\} = \mathbb{E}\{\overline{\eta}_{s,t}(O, x)\overline{\eta}_{s',t'}(O', x)\}$, where $O' := (W', Z', Y', X')$ is an independent copy of $O$; (ii). $\Delta_{\varphi,m}^{(Z)}$ converges weakly to $\max_{t \in \mathcal{T}} \int_\mathcal{T} |\mathbb{G}(s, t)|^2 d\mu(s)$.*

Similarly, we can establish the asymptotic power for $\Delta_{\varphi,m}^{(Z)}$, where the global and alternatives are defined accordingly in terms of $\mathbb{E}\{\varphi(Y, t) - H(W, t)|Z, x\}$. Due to space limit, we leave the result and its proof in Appx. E.4.

## 7. Experiments

In this section, we evaluate our methods on synthetic data. We consider two settings: **(i)** (Sec. 7.1) $W \not\to Y$ where only the NCO $W$ is available; **(ii)** (Sec. 7.2) $W \to Y$ where the additional proxy $Z$ (*i.e.*, NCE) is provided[2].

**Compared baselines.** For the single-proxy setting, we compare our methods with: **(i) Liu** (Liu et al., 2023) that designed a discretization method for bivariate causal discovery over continuous variables; **(ii) KCI** the Kernel-based Conditional Independence test (KCI) (Zhang et al., 2012) that tested the null hypothesis of $X \perp\!\!\!\perp Y|W$ using kernel matrices. For the two-proxy setting, we also conduct **(iii) Miao** (Miao et al., 2018) that was designed for causal hypothesis testing over discrete variables using $W$ and $Z$.

**Implementation details.** We set the significance level $\alpha$ to 0.05. We choose $\varphi$ and $m$ to be complex exponential functions. For PMCR estimation, we set $K = 100$ and follow (Mastouri et al., 2021) to select the optimal $\lambda$ from a

---

[2]Code is available at https://github.com/yezichu/proximal_causal_discovery_cv.

sequence ranging from $4.9 \times 10^{-6}$ to 0.25, with a step size chosen to ensure the sequence contains 50 values. Besides, we use Gaussian kernels with the bandwidth parameters being initialized using the median distance heuristic. For the procedure of **Liu** (Liu et al., 2023), we follow the paper to set the bin numbers of $W$ and $X$ to $l_X = 14, l_W = 12$, respectively. For the procedure described in **Miao**, we implement the R code released in the paper and set $l_X = 3, l_W = 2, l_Z = 2$ by default. For **KCI**, we adopt the implementations provided in the causallearn packages https://causal-learn.readthedocs.io/.

## 7.1. Single proxy with $W \not\rightarrow Y$

We first evaluate our method in Sec. 4 to the setting where only $W$ is provided.

**Data generation.** We follow (Liu et al., 2023) to generate $V \in \{X, Y, U, W\}$ via $V = f_V(\mathbf{PA}_V) + \varepsilon_V$, where $\mathbf{PA}_V$ and $\varepsilon_V$ respectively denotes the parent set and the noise of $V$. For each $V$, $f_V$ is randomly selected from $\{\text{linear}, \tanh, \sin, \text{sqrt}\}$. Besides, the distribution of $\varepsilon_V$ is randomly chosen from $\{\text{Gaussian}, \text{uniform}, \text{exponential}, \text{gamma}\}$. To mitigate the effect of randomness, we repeat the process 20 times. At each time, we generate 100 replications under each $\mathbb{H}_0$ and $\mathbb{H}_1$, and record the type-I error rate and power rate.

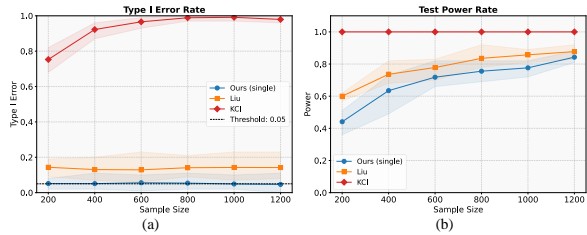

*Figure 3.* Type-I error rate (left) and power rate (right) of our testing procedure and baseline methods in the single-proxy setting. The solid line reports the average value over 20 times, and the shaded area denotes the region $(\text{mean} - \text{std}, \text{mean} + \text{std})$.

**Type-I error and power.** In Fig. 3, we report the average type-I error rate and power rate for our testing procedure and others. As shown, the type-I error rate of our method closely approximates $\alpha = 0.05$ as $n$ increases, while other methods fail to control the type-I error. Specifically, conditioning on the proxy $W$, **KCI** cannot eliminate the confounding bias, leading to uncontrollable type-I errors; while the additional error in **Liu** (Liu et al., 2023) may arise from discretization errors with finite bin number or probability estimation error due to limited sample size. Besides, our power approximates to one as $n$ increases. Compared to previous baselines **Liu**, these results demonstrate the utility and its ability to make better use of available data in causal discovery.

**Comparisons with MMR.** To further demonstrate the effec-

tiveness of our estimation method PMCR over the MMR, we apply PMCR to the data generated in Example B.3, where we have shown that the solution of the first-moment equation exists under the alternative hypothesis. As shown in Fig. 4, although both methods can asymptotically control the type-I error as $n \rightarrow \infty$, the power of our procedure approaches 1 while the first-moment method (*i.e.*, MMR) still approximates $\alpha = 0.05$ under $\mathbb{H}_1$.

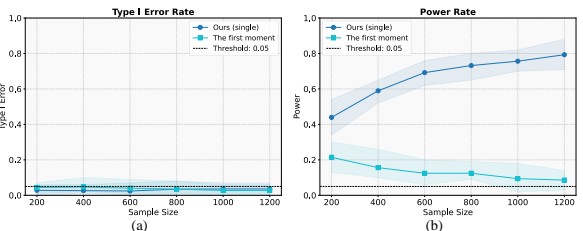

*Figure 4.* Type-I error rate (left) and power rate (right) of our procedure with PMCR and the first-moment method in example B.3.

## 7.2. Two proxies with $W \rightarrow Y$

In this section, we apply our method in Sec. 6 to the setting $W \rightarrow Y$ with both $W$ and $Z$ are available.

**Data generation.** Following example 5.3[3], we set $\gamma_W = 1$, which implies there exists $h$ that satisfies the integral equation (1). Similar to the single-proxy setting, we repeat the process 20 times, where at each time we generate 100 replications under $\mathbb{H}_0$ and $\mathbb{H}_1$.

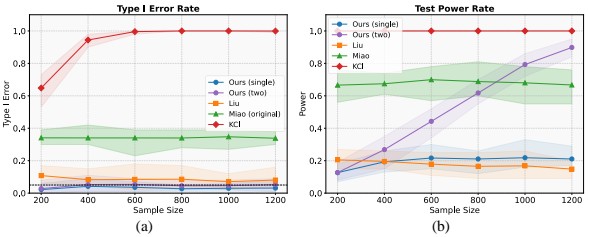

*Figure 5.* Type-I error rate (left) and power rate (right) of our procedure and baselines on synthetic data with two proxies.

**Type-I error and power.** We report the average results in Fig. 5. As shown, although our single-proxy procedure can control the type-I error, it suffers from low power in learning the causal relation, due to the existence of solution under $\mathbb{H}_1$ in this example. With additional information provided by $Z$, the power significantly improves and approaches one as $n$ increases. This verifies our findings in Sec. 5, and demonstrates the utility of employing $Z$ (*i.e.*, NCE) in discovering the causal relation when the effect of $W$ on $Y$ is strong enough to invalidate the procedure with only $W$.

---

[3]We also consider a nonlinear setting, as detailed in Appx. F.

## 8. Conclusions and discussions

We introduce a non-parametric procedure for causal hypothesis testing and establish its asymptotic properties. Additionally, we show that causal relationships may not be identifiable through examining the integral equation with only NCO. We then leverage the additional NCE that can effectively recover the causation when it exists. We believe our findings, supported by theoretical justifications, provide new insights into proximal causal discovery.

Currently, we only analyze the integral-related assumption when NCO has a direct effect on the outcome. **In the future**, we will extend our analysis to the case when there is a bidirectional edge between them.

## Impact Statement

This paper presents work whose goal is to advance the field of Machine Learning. There are many potential societal consequences of our work, none of which we feel must be specifically highlighted here.

## Acknowledgements

This work was supported by Young Scientists Fund of the National Natural Science Foundation of China (Grant No. KRH2305058); the State Key Program of National Natural Science Foundation of China under Grant No. 12331009; STI 2030—Major Projects(No. 2021ZD0200407); National Key Research and Development Program of China(No. 2022YFC2405100). The computations in this research were performed using the CFFF platform of Fudan University.

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

# Appendix

## A. Notations

We introduce notations used throughout the appendix.

Table 1: Notations.

| Notation | Definition |
|---|---|
| $Z, W, U$ | Negative control exposure, negative control outcome and unobserved confounder |
| $\mathbb{P}(X)$ | $\{p(x_1), ..., p(x_k)\}^\top$ for any discrete variables $X$ with $k$ categories |
| $\mathbb{P}(Y\|X)$ | $\begin{pmatrix} p(y_1\|x_1) & \cdots & p(y_1\|x_k) \\ \vdots & \ddots & \vdots \\ p(y_l\|x_1) & \cdots & p(y_l\|x_k) \end{pmatrix}$ for any discrete variables $Y, X$ with $l, k$ categories |
| $\mathbb{P}(Y = y\|X, Z)$ | $\begin{pmatrix} p(y\|x_1, z_1) & \cdots & p(y\|x_1, z_m) \\ \vdots & \ddots & \vdots \\ p(y\|x_k, z_1) & \cdots & p(y\|x_k, z_m) \end{pmatrix}$ for any discrete variables $X, Z$ with $k, m$ categories |
| $\mathcal{H}_W, \mathcal{H}_X$ | The reproducing kernel Hilbert spaces (RKHS) defined on the domains of $W$ and $X$ |
| $\phi_W(w), \phi_X(x)$ | The canonical feature map defined on the domains of $W$ and $X$ |
| $k_W(w, w'), k_X(x, x')$ | The reproducing kernel for the RKHS $\mathcal{H}_W$ and $\mathcal{H}_X$ |
| $R(H)$ | The population loss function defined in Eq. (22) |
| $\widehat{R}^\lambda(H)$ | The regularized empirical risk defined in Eq. (5) |
| $A, b_t(x) = b(x, t)$ | The operator and right term defined in Eq. (24) |
| $\widehat{A}, \widehat{b}_t(x) = \widehat{b}(x, t)$ | The plugging operator and right term defined in Eq. (26) |
| $A^*, \widehat{A}^*$ | The adjoint operator of $A$ and $\widehat{A}$ defined in Eq. (27) and (28) |
| $(\lambda_j, \varphi_j, \phi_j)_j$ | The singular value decomposition of the operator $A$ |
| $\mathcal{H}_{W,0}$ | The set of all solutions of Eq. (22) defined in Eq. (29) |
| $H_t^\lambda(w) = H^\lambda(w, t)$ | The population Tikhonov regularization solution defined in Eq. (31) |
| $\widehat{H}_t^\lambda(w) = \widehat{H}^\lambda(w, t)$ | The empirical Tikhonov regularization solution defined in Eq. (32) |
| $H_t^0(w) = H^0(w, t)$ | The least norm solution in Eq. (4) |
| $\mathrm{Ker}(A)$ | $\mathrm{Ker}(A) = \{H : Ah = 0\}$ is the null space of the operator $A$ |
| $\mathrm{Ran}(A)$ | $\mathrm{Ran}(A) = \{f : Ah = f\}$ is the ranged space of the operator $A$ |
| $\mathcal{L}^2\{F(w)\}, \mathcal{L}^2\{F(x)\}$ | The space of square-integrable functions with respect to the cumulative distribution function $F(w)$ and $F(x)$ |
| $\mathcal{L}^2\{\mathcal{S} \times \mathcal{T}, \mu \times \mu\}$ | We say $\mathbb{G}(s, t) \in \mathcal{L}^2\{\mathcal{S} \times \mathcal{T}, \mu \times \mu\}$ if $\iint \|\mathbb{G}(s, t)\|^2 d\mu(s) d\mu(t) < \infty$ |
| $\varphi(\cdot, t), m(\cdot, s)$ | The weight function |
| $g_s$ | $g_s = \mathbb{E}\{m(X, s)\phi_W(W)\}$ |
| $U(W, Y, t), \widehat{U}(W, Y, t)$ | The residual $\varphi(Y, t) - H^0(W, t)$ and estimated version $\varphi(Y, t) - \widehat{H}^\lambda(W, t)$ |
| $T_n(s, t)$ | The statistics defined in Eq. (6) |
| $\Delta_{\varphi, m}$ | The statistics defined in Eq. (7) |

| Notation | Definition |
|---|---|
| $\mathbb{E}(\cdot)$ | The expectation with respect to both a random variable and data |
| $\mathbb{P}(\cdot)$ | The expectation with respect to a random variable alone |
| $\mathbb{P}_n(\cdot)$ | The empirical expectation with respect to a random variable given data |
| $\|\cdot\|_{\mathcal{F}}$ | The norm with respect to space $\mathcal{F}$ |

## B. Existence of solutions with a single proxy

Let $\mathcal{L}^2\{F(x)\}$ denote the space of all square-integrable functions of $x$ with respect to a cumulative distribution function $F(x)$, which is a Hilbert space with inner product $\langle g_1, g_2 \rangle = \int g_1(x)g_2(x)p(x)dx$. Let $T$ denote the operator: $\mathcal{L}^2\{F(w)\} \to \mathcal{L}^2\{F(x)\}$, $Tg = \mathbb{E}\{g(W)|X = \cdot\}$ and let $(\lambda_n, \varphi_n, \phi_n)_{n=1}^\infty$ denote a singular value decomposition of $T$.

**Assumption B.1.** Assume the following conditions for all $u$:

(1) $\iint p(x|w)p(w|x)dwdx < \infty$ and $\int \{p(u|x)\}^2 p(x)dx < \infty$;

(2) $\sum_{n=1}^\infty \lambda_n^{-2}|\langle p(u|x), \phi_n \rangle|^2 < \infty$.

### B.1. Proof of Theorem 4.5

We first prove that under the conditions Theorem 4.5, there exists a solution $g(w, u) \in \mathcal{L}^2\{F(w)\}$ for all $u$, such that $p(u|x) = \int g(w, u)p(w|x)dw$. Our proof is based on Picard's Theorem, which is presented below.

**Lemma B.2** (Theorem 15.18 in (Kress, 1989)). *Given Hilbert spaces $\mathcal{H}_1$ and $\mathcal{H}_2$, a compact operator $T : H_1 \to H_2$ and its adjoint operator $T^* : \mathcal{H}_2 \to \mathcal{H}_1$, there exists a singular system $(\lambda_n, \varphi_n, \phi_n)_{n=1}^{+\infty}$ of $K$ with nonzero singular values $\{\lambda_n\}$ and orthogonal sequences $\{\varphi_n \in \mathcal{H}_1\}, \{\phi_n \in \mathcal{H}_2\}$. Then the equation of the first kind $Th = f$ with $f \in \mathcal{H}_2$, has a solution if and only if*

*1. $f \in \text{Ker}(T^*)^\perp$, where $\text{Ker}(T^*) = \{h : T^*h = 0\}$ is the null space of the adjoint operator $T^*$;*

*2. $\sum_{n=1}^{+\infty} \lambda_n^{-2}|\langle f, \phi_n \rangle|^2 < +\infty$.*

**Proposition 4.2.** *Under assumption 4.1 and some regularity conditions in B.1, there exists a $g(w, u) \in \mathcal{L}^2\{F(w)\}$ for all $u$, such that it solves the following integral equation for all $(u, x)$:*

$$p(u|x) = \int g(w, u)p(w|x). \tag{2}$$

*Proof.* Our goal is to show the solution exists for the following estimator:

$$T : \mathcal{L}^2\{F(w)\} \to \mathcal{L}^2\{F(x)\} : \ Tf = \mathbb{E}\{f(W)|X = \cdot\}, \ f \in \mathcal{L}^2\{F(w)\}.$$

Besides, we consider the following operator:

$$S : \mathcal{L}^2\{F(x)\} \to \mathcal{L}^2\{F(w)\} : \ Sg = \mathbb{E}\{g(X)|W = \cdot\}, \ f \in \mathcal{L}^2\{F(x)\}.$$

By Lemma B.2 and assumption B.1 (2) for $p(u|x)$, the conclusion holds if we can show that $S$ is the adjoint operator of $T$, $T$ is compact, and that $p(u|x) \in \text{Ker}(S)^\perp$.

**(i). $S$ is the adjoint operator of $T$.**

For the operator $T$, $\forall f \in \mathcal{L}^2\{F(w)\}$ and $\forall g \in \mathcal{L}^2\{F(x)\}$, we have

$$
\begin{aligned}
\langle Tf, g \rangle_{\mathcal{L}^2\{F(x)\}} &= \mathbb{E}_X[\mathbb{E}\{f(W)|X\}g(X)] \\
&= \mathbb{E}_X[\mathbb{E}_{U|X}\mathbb{E}\{f(W)|X, U\}g(X)] \\
&\overset{(1)}{=} \mathbb{E}_X[\mathbb{E}_{U|X}\mathbb{E}\{f(W)|U\}g(X)] \\
&= \mathbb{E}_{U,X}[\mathbb{E}\{f(W)|U\}g(X)] \\
&= \mathbb{E}_U[\mathbb{E}\{f(W)|U\}\mathbb{E}\{g(X)|U\}].
\end{aligned}
$$

Similarly,

$$
\begin{aligned}
\langle f, Sg \rangle_{\mathcal{L}^2\{F(w)\}} &= \mathbb{E}_W[f(W)\mathbb{E}\{g(X)|W\}] \\
&= \mathbb{E}_W[f(W)\mathbb{E}_{U|W}\mathbb{E}\{g(X)|W, U\}] \\
&\overset{(2)}{=} \mathbb{E}_W[f(W)\mathbb{E}_{U|W}\mathbb{E}\{g(X)|U\}] \\
&= \mathbb{E}_{U,W}[f(W)\mathbb{E}\{g(X)|U\}] \\
&= \mathbb{E}_U[\mathbb{E}\{f(W)|U\}\mathbb{E}\{g(X)|U\}],
\end{aligned}
$$

where (1) and (2) follow from $W \perp\!\!\!\perp X | U$. Therefore, we have

$$\langle Tf, g \rangle_{\mathcal{L}^2\{F(x)\}} = \langle f, Sg \rangle_{\mathcal{L}^2\{F(w)\}}.$$

**(ii). $T$ is compact.**

We define the integral kernel

$$K(w, x) = \frac{p(w, x)}{p(w)p(x)}. \tag{12}$$

Then for the operator defined previously, we have the following form:

$$Tf = \int K(w, x)f(w)d\mathbb{P}(w) = \mathbb{E}\{f(W)|X\}, \ f \in \mathcal{L}^2\{F(w)\}, \tag{13}$$

$$Sg = \int K(w, x)g(x)d\mathbb{P}(x) = \mathbb{E}\{g(X)|W\}, \ g \in \mathcal{L}^2\{F(x)\}. \tag{14}$$

By the definition of $K$ in (12), we have:

$$\iint |K(w, x)|^2 p(w)p(x)dwdx = \iint p(w|x)p(x|w)dwdx.$$

Under assumption B.1 (1), we can apply Theorem 2.34 in (Carrasco et al., 2007) to obtain that the operator $T$ is a Hilbert-Schmidt operator, which implies that $T$ is a compact operator, according to Theorem 2.32 in (Carrasco et al., 2007).

**(iii). $p(u|x) \in \mathrm{Ker}(S)^\perp$ for any $u$.**

For each $g \in \mathcal{N}(S)$, by iterated expectations, we have

$$\mathbb{E}\{g(X)|W\} = \mathbb{E}_{U|W}[\mathbb{E}\{g(X)|W, U\}]$$
$$\overset{(1)}{=} \mathbb{E}_{U|W}[\mathbb{E}\{g(X)|U\}] = 0,$$

where (1) follows from $W \perp\!\!\!\perp X | U$. By the completeness assumption, we have

$$\mathbb{E}\{g(X)|u\} = 0 \text{ a.s.} \tag{15}$$

Then we have

$$\langle g, p(u|X) \rangle_{L^2\{F(x)\}} = \mathbb{E}_X\{g(X)p(u|X)\} \qquad (p(u|x) \in \mathcal{L}^2\{F(x)\} \text{ is used here.})$$
$$= \int g(x)p(u, x)dx = \int p(u)\mathbb{E}\{g(X)|u\}du = 0,$$

which implies $p(u|x) \in \mathrm{Ker}(S)^\perp$. Combining the above three steps together, we obtain the conclusion. $\square$

**Theorem 4.5.** *Suppose assumptions 4.1, 4.4 and the regularity condition B.1 hold. If $g(w, u)$ in (2) satisfies $\iint |\frac{g(w,u)}{p(u)}|^2 p(w)p(u)dwdu < \infty$, then under $\mathbb{H}_0$, there exists a $h(w, y) \in \mathcal{L}^2\{F(w)\}$ for all $y$, such that it solves the integral equation in (1) for all $(x, y)$.*

*Proof.* By proposition 4.2, we have $g(w, u)$ satisfies the integral equation $p(u|x) = \int g(w, u)p(w|x)dw$. Then under $\mathbb{H}_0$, we have:

$$p(y|x) = \int p(y|u)p(u|x)du$$
$$= \int \left\{ \int g(w, u)p(y|u)du \right\} p(w|x)dw.$$

Therefore, we write an expression $h(w, y) := \int g(w, u)p(y|u)du$ that satisfies a solution of the integral Eq. (1).

We first prove that $h(w, y)$ is well-defined for any fixed $y$. To be specific,

$$\left| \int g(w, u)p(y|u)du \right| \leq \int |g(w, u)|\, p(y|u)du$$

$$= \int \left| \frac{g(w, u)}{p(u)} \right| p(u) \cdot \frac{p(u|y)}{p(u)}p(y)du$$

$$\overset{(1)}{\leq} Cp(y) \cdot \int \left| \frac{g(w, u)}{p(u)} \right| p(u)du,$$

where (1) follows from assumption 4.4. Since we have assumed $\iint |g(w, u)/p(u)|^2 p(w)p(u)dwdu < \infty$, there holds $\int |g(w, u)/p(u)|^2 p(u)du < \infty$ for a.e $w$ by Fubini's theorem. Applying the Cauchy-Schwarz inequality, we have:

$$\int \left| \frac{g(w, u)}{p(u)} \right| p(u)du \leq \left\{ \int \left| \frac{g(w, u)}{p(u)} \right|^2 p(u)du \right\}^{1/2} \left\{ \int p(u)du \right\}^{1/2} < \infty. \tag{16}$$

Thus, $h(w, y)$ is well-defined for any $y$.

Next, we show that $h(w, y) \in \mathcal{L}^2\{F(w)\}$ for any $y$. By Cauchy-Schwarz inequality, we have

$$\int |h(w, y)|^2 p(w)dw = \int \left\{ \int g(w, u)p(y|u)du \right\}^2 p(w)dw$$

$$= \int \left\{ \int \frac{g(w, u)}{p(u)}p(y|u)p(u)du \right\}^2 p(w)dw$$

$$\leq \int \left\{ \int \left| \frac{g(w, u)}{p(u)} \right|^2 p(u)du \right\} \left\{ \int |p(y|u)|^2 p(u)du \right\} p(w)dw$$

$$= \left\{ \iint \left| \frac{g(w, u)}{p(u)} \right|^2 p(w)p(u)dwdu \right\} \left\{ \int \left| \frac{p(u|y)}{p(u)} \right|^2 p(y)^2 p(u)du \right\}$$

$$\leq \left\{ \iint \left| \frac{g(w, u)}{p(u)} \right|^2 p(w)p(u)dwdu \right\} \left\{ \int C^2 p(y)^2 p(u)du \right\}.$$

Since $\iint \left| \frac{g(w,u)}{p(u)} \right|^2 p(w)p(u)dwdu < \infty$, we complete the proof. $\qquad \square$

## B.2. Proof of Corollary 4.7

**Corollary 4.7.** *Suppose conditions in Theorem 4.5 hold. For any t, $H(w, t)$ in (4) exists and belongs to $\mathcal{L}^2\{F(w)\}$.*

*Proof.* **(i).** We first prove that $H(w, t)$ is well-defined. To be specific, we take $\varphi$ to be the complex exponential function $e^{ity}$. Then, by $H(w, t) = \int \varphi(y, t)h(w, y)dy$, we have

$$\left| \int e^{ity}h(w, y)dy \right| \leq \int |e^{ity}| \cdot \left| \int g(w, u)p(y|u)du \right| dy \leq \int |e^{ity}| \cdot \int |g(w, u)|\, p(y|u)dudy$$

$$\leq \int |e^{ity}| \cdot \left\{ \int \left| \frac{g(w, u)}{p(u)} \right| p(u) \cdot \frac{p(u|y)}{p(u)}p(y)du \right\} dy$$

$$\overset{(1)}{\leq} C \cdot \int |e^{ity}| \cdot p(y) \left\{ \int \left| \frac{g(w, u)}{p(u)} \right| p(u)du \right\} dy,$$

where (1) follows from assumption 4.4. Since $|e^{ity}| = 1$, $\int p(y)\, dy = 1$, and by Eq. (16), it follows that $H(w, t)$ is well-defined.

**(ii).** We proof $H(w,t) \in \mathcal{L}^2\{F(w)\}$. By Cauchy-Schwarz inequality, we have:

$$
\begin{aligned}
\int |H(w,t)|^2 p(w)dw &= \int \left\{ \int g(w,u)e^{ity}p(y|u)dudy \right\}^2 p(w)dw \\
&= \int \left\{ \int \frac{g(w,u)}{p(u)}e^{ity}p(y|u)p(u)dudy \right\}^2 p(w)dw \\
&\le \int \left\{ \int \left| \frac{g(w,u)}{p(u)} \right|^2 p(u)du \right\} \left\{ \int \left| \int e^{ity}p(y|u)dy \right|^2 p(u)du \right\} p(w)dw \\
&= \left\{ \iint \left| \frac{g(w,u)}{p(u)} \right|^2 p(w)p(u)dwdu \right\} \left\{ \int \left| \int e^{ity}p(y|u)dy \right|^2 p(u)du \right\}.
\end{aligned}
\tag{17}
$$

The proof is completed as $\left| \int e^{ity}p(y|u)dy \right| \le 1$ and $\iint \left| \frac{g(w,u)}{p(u)} \right|^2 p(w)p(u)dwdu < \infty$. $\qquad\square$

### B.3. Counter-example to the solvability of the first-order moment equation under $\mathbb{H}_1$

**Example B.3.** Suppose that $X, Y, U, W$ satisfy the linear Gaussian model, *i.e.* $X = \varepsilon_X, U = \alpha_X X + \alpha_0 + \varepsilon_U, W = \beta_U U + \beta_0 + \varepsilon_W, Y = \gamma_U U + \gamma_X X + \gamma_0 + \varepsilon_Y$, where $\varepsilon_U, \varepsilon_X, \varepsilon_W, \varepsilon_Y$ are Gaussian noises. Then the solution of $\mathbb{E}(Y|X) = \mathbb{E}\{h(W)|X\}$ is given by $h(w) := \frac{\gamma_X + \gamma_U \alpha_X}{\mu_U \alpha_X} w + \gamma_0 + \gamma_U \alpha_0 - (\beta_0 + \mu_U \alpha_0)\frac{\gamma_X + \gamma_U \alpha_X}{\mu_U \alpha_X}$.

*Proof.* We show that $\mathbb{E}\{Y|X\} = \mathbb{E}(b_w W + b_0 | X)$, where

$$
(b_0, b_w) = \left\{ \gamma_0 + \gamma_U \alpha_0 - (\beta_0 + \mu_U \alpha_0)\frac{\gamma_X + \gamma_U \alpha_X}{\mu_U \alpha_X}, \frac{\gamma_X + \gamma_U \alpha_X}{\mu_U \alpha_X} \right\}.
$$

In fact, this can be achieved by solving $(b_0, b_w)$ in the following integral equation:

$$
\mathbb{E}(Y|X) = \mathbb{E}(b_w W + b_0 | X).
$$

For the left-hand side, we have

$$
\begin{aligned}
\mathbb{E}(Y|X) &= \gamma_0 + \gamma_X X + \gamma_U \mathbb{E}(U|X) \\
&= \gamma_0 + \gamma_X X + \gamma_U \mathbb{E}(\alpha_0 + \alpha_X X|X) \\
&= (\gamma_0 + \gamma_U \alpha_0) + (\gamma_X + \gamma_U \alpha_X)X.
\end{aligned}
$$

For the right-hand side, we have

$$
\begin{aligned}
\mathbb{E}\{g(W)|X\} &= \mathbb{E}(b_0 + b_w W|X) \\
&= b_0 + b_w \mathbb{E}(\beta_0 + \mu_U U|X) \\
&= b_0 + b_w\{\beta_0 + \mu_U \mathbb{E}(\alpha_0 + \alpha_X X|X)\} \\
&= (b_0 + b_w \beta_0 + b_w \mu_U \alpha_0) + (b_w \mu_U \alpha_X)X,
\end{aligned}
$$

$$
\begin{cases}
b_w \mu_U \alpha_X - (\gamma_X + \gamma_U \alpha_X) = 0 \\
(b_0 + b_w \beta_0 + b_w \mu_U \alpha_0) - (\gamma_0 + \gamma_U \alpha_0) = 0
\end{cases}
\implies
\begin{cases}
b_w = \dfrac{\gamma_X + \gamma_U \alpha_X}{\mu_U \alpha_X} \\
b_0 = \gamma_0 + \gamma_U \alpha_0 - (\beta_0 + \mu_U \alpha_0)\dfrac{\gamma_X + \gamma_U \alpha_X}{\mu_U \alpha_X}
\end{cases}.
$$

$\qquad\square$

### B.4. Verification of $\iint \frac{g(w,u)}{p(u)}|^2 p(w)p(u)dwdu < \infty$ in Theorem 4.5

**Lemma B.4** ((Miao et al., 2018)). *If $W|X \sim \mathcal{N}(\beta'_0 + \beta'_1 X, \sigma_2^2)$ and $Y|X \sim \mathcal{N}(\gamma'_0 + \gamma'_1 X, \sigma_3^2)$, then one can verify integral equation $p(y|x) = \int h(w,y)p(w|x)dw$ has a unique solution $h(w,y)$:*

$$
h(w,y) = \frac{1}{\sigma_{wx}}\phi\left(\frac{y - \gamma_{wx} - \gamma'_1/\beta'_1 w}{\sigma_{wx}}\right),
\tag{18}
$$

*where $\phi$ is the probability density function (pdf) of the standard normal distribution, $\gamma_{wx} = \gamma_0' - \gamma_1'\beta_0'/\beta_1'$ and $\sigma_{wx}^2 = \sigma_3^2 - (\gamma_1')^2\sigma_2^2/(\beta_1')^2$.*

**Example B.5.** We consider the linear Gaussian generation mechanism:

$$
\begin{cases}
U \sim \mathcal{N}(0, \sigma^2) \\
X = \alpha_0 + \alpha_U U + \mathcal{N}(0, 1) \\
W = \beta_0 + \beta_U U + \mathcal{N}(0, 1) \\
Y = \gamma_0 + \gamma_U U + \mathcal{N}(0, 1).
\end{cases}
$$

Then $h(w, y)$ given by the following has a unique solution to the integral equation (1):

$$
h(w, y) = \frac{1}{\sqrt{1 - \left(\frac{\gamma_U}{\beta_U}\right)^2}} \phi \left( \frac{y - \frac{\gamma_U}{\beta_U} + \frac{\gamma_U}{\beta_U}\beta_0 - \gamma_0}{\sqrt{1 - \left(\frac{\gamma_U}{\beta_U}\right)^2}} \right).
$$

Besides, at this time, the assumption in Theorem 4.5 has

$$
\int \left\{ \frac{g(w, u)}{p(u)} \right\}^2 p(w)p(u)dwdu = \frac{\sigma^2 - \sigma^2 k + k^2/\alpha_U^2}{1 - \sigma^2 k - k\sigma^2\beta_U^2} < \infty, \tag{19}
$$

where $k := (\sigma\alpha_U)^2/(\sigma^2\alpha_U^2 + 1)$.

*Proof.* Based on the data generation structure, we can obtain joint distribution

$$
\begin{pmatrix} U \\ X \\ W \\ Y \end{pmatrix} \sim \mathcal{N} \left\{ \begin{pmatrix} 0 \\ \alpha_0 \\ \beta_0 \\ \gamma_0 \end{pmatrix}, \begin{pmatrix} \sigma^2 & \sigma^2\alpha_U & \sigma^2\beta_U & \sigma^2\gamma_U \\ \sigma^2\alpha_U & \sigma^2\alpha_U^2 + 1 & \sigma^2\alpha_U\beta_U & \sigma^2\alpha_U\gamma_U \\ \sigma^2\beta_U & \sigma^2\alpha_U\beta_U & \sigma^2\beta_U^2 + 1 & \sigma^2\beta_U\gamma_U \\ \sigma^2\gamma_U & \sigma^2\alpha_U\gamma_U & \sigma^2\beta_U\gamma_U & \sigma^2\gamma_U^2 + 1 \end{pmatrix} \right\}.
$$

We now get the conditional distributions $p(w|x)$ and $p(y|x)$ according to the joint distribution

$$
W|X = x \sim \mathcal{N} \left\{ \mu_W + \frac{\text{Cov}(W, X)}{\text{Var}(X)}(x - \mu_x), \text{Var}(W)\left(1 - \frac{\text{Cov}^2(W, X)}{\text{Var}(X) \cdot \text{Var}(W)}\right) \right\}
$$

$$
\sim \mathcal{N} \left\{ \frac{\sigma^2\alpha_U\beta_U}{\sigma^2\alpha_U^2 + 1}x - \frac{\sigma^2\alpha_U\beta_U}{\sigma^2\alpha_U^2 + 1}\alpha_0 + \beta_0, \sigma^2\beta_U^2 + 1 - \frac{(\sigma^2\alpha_U\beta_U)^2}{\sigma^2\alpha_U^2 + 1} \right\}
$$

$$
Y|X = x \sim \mathcal{N} \left\{ \mu_y + \frac{\text{Cov}(Y, X)}{\text{Var}(X)}(x - \mu_x), \text{Var}(Y)\left(1 - \frac{\text{Cov}^2(Y, X)}{\text{Var}(X) \cdot \text{Var}(Y)}\right) \right\}
$$

$$
\sim \mathcal{N} \left\{ \frac{\sigma^2\alpha_U\gamma_U}{\sigma^2\alpha_U^2 + 1}x - \frac{\sigma^2\alpha_U\gamma_U}{\sigma^2\alpha_U^2 + 1}\alpha_0 + \gamma_0, \sigma^2\gamma_U^2 + 1 - \frac{(\sigma^2\alpha_U\gamma_U)^2}{\sigma^2\alpha_U^2 + 1} \right\}.
$$

By Lemma B.4, the solution to (1) with $h(w, y)$ given above is

$$
h(w, y) = \frac{1}{\sqrt{1 - \left(\frac{\gamma_U}{\beta_U}\right)^2}} \phi \left( \frac{y - \frac{\gamma_U}{\beta_U} + \frac{\gamma_U}{\beta_U}\beta_0 - \gamma_0}{\sqrt{1 - \left(\frac{\gamma_U}{\beta_U}\right)^2}} \right).
$$

Next, we prove (19). We will first obtain $g(w, u)$. Specifically, we first get $p(u|x)$:

$$
U|X = x \sim \mathcal{N} \left\{ \mu_U + \frac{\text{Cov}(U, X)}{\text{Var}(X)}(x - \mu_x), \text{Var}(U)\left(1 - \frac{\text{Cov}^2(U, X)}{\text{Var}(X) \cdot \text{Var}(U)}\right) \right\}
$$

$$
\sim \mathcal{N} \left\{ \frac{\sigma^2\alpha_U}{\sigma^2\alpha_U^2 + 1}x - \frac{\sigma^2\alpha_U}{\sigma^2\alpha_U^2 + 1}\alpha_0, \sigma^2 - \sigma^2\frac{(\sigma\alpha_U)^2}{\sigma^2\alpha_U^2 + 1} \right\}.
$$

Similarly, by applying Lemma B.4 with the form of $Y|X$ replaced by that of $U|X$ above, we have:

$$g(w,u) = \frac{1}{\sqrt{\frac{(\sigma^2\alpha_U)^2}{\sigma^2\alpha_U^2+1}\left(1-\sigma^2\right)-\left(\frac{1}{\beta_U}\right)^2}}\phi\left(\frac{u-\frac{1}{\beta_U}w+\frac{\beta_0}{\beta_U}}{\sqrt{\frac{(\sigma^2\alpha_U)^2}{\sigma^2\alpha_U^2+1}\left(1-\sigma^2\right)-\left(\frac{1}{\beta_U}\right)^2}}\right).$$

We can easily verify that the following integration is finite, by the following:

$$\int\left\{\frac{g(w,u)}{p(u)}\right\}^2 p(w)p(u)dwdu = \frac{\sigma^2-\sigma^2 k+\frac{k^2}{\alpha_U^2}}{1-\sigma^2 k-k\sigma^2\beta_U^2},$$

where $k := \frac{(\sigma\alpha_U)^2}{\sigma^2\alpha_U^2+1}$. $\qquad\square$

# C. Proxy Maximum Characteristic Restriction

For the sake of completeness, we introduce some preliminary concepts that are necessary to understand the theoretical analysis of our PMCR method. First, in section C.1–C.3, we introduce some background knowledge of the linear operators and Reproducing Kernel Hilbert Spaces required in this article. Upon this, we provide details on the derivation of our empirical loss (5) in section C.4. Section C.5 rewrites the loss into the Tikhonov regularized form, which serves as the foundation of our theoretical analysis for Theorem 4.12.

## C.1. Bounded linear operator

For two normed linear spaces $\mathcal{F}$ and $\mathcal{G}$ over $\mathbb{R}$, a function $A : \mathcal{F} \to \mathcal{G}$ (where $\mathcal{F}$ and $\mathcal{G}$ are both normed linear spaces over $\mathbb{R}$) is called a linear operator if it satisfies the following properties:

1. Homogeneity: $A(\alpha f) = \alpha(Af)$, for any $\alpha \in \mathbb{R}, f \in \mathcal{F}$;

2. Additivity: $A(f + g) = Af + Ag$, for any $f, g \in \mathcal{F}$.

**Operator Norm and Boundedness.** The operator norm of a linear operator $A : \mathcal{F} \to \mathcal{G}$ is defined as

$$\|A\|_{\mathrm{op}} = \sup_{f \in \mathcal{F}} \frac{\|Af\|_{\mathcal{G}}}{\|f\|_{\mathcal{F}}}.$$

A linear operator $A$ is called bounded if there exists a finite constant $C$ such that for all $f \in \mathcal{F}$, we have

$$\|Af\|_{\mathcal{G}} \leq C\|f\|_{\mathcal{F}}.$$

In terms of the operator norm, this condition is equivalent to saying that $\|A\|_{\mathrm{op}} < \infty$.

## C.2. Hilbert space

We begin by introducing definitions and basic properties of an inner product space. Based on this, we introduce the Hilbert space.

A function $\langle \cdot, \cdot \rangle_{\mathcal{F}} : \mathcal{F} \times \mathcal{F} \to \mathbb{R}$ is said to be an inner product on $\mathcal{F}$ if it satisfies the following three properties

1. $\langle \alpha_1 f_1 + \alpha_2 f_2, g \rangle_{\mathcal{F}} = \alpha_1 \langle f_1, g \rangle_{\mathcal{F}} + \alpha_2 \langle f_2, g \rangle_{\mathcal{F}}.$

2. $\langle f, g \rangle_{\mathcal{F}} = \langle g, f \rangle_{\mathcal{F}}.$

3. $\langle f, f \rangle_{\mathcal{F}} \geq 0$ and $\langle f, f \rangle_{\mathcal{F}} = 0$ if and only if $f = 0$.

One can always define a norm induced by the inner product: $\|f\|_{\mathcal{F}} = \langle f, f \rangle_{\mathcal{F}}^{1/2}$. For this norm, the following Cauchy-Schwarz inequality holds, *i.e.*, $|\langle f, g \rangle_{\mathcal{F}}| \leq \|f\|_{\mathcal{F}} \cdot \|g\|_{\mathcal{F}}$.

A Hilbert space is a complete inner product space. This means, a Hilbert space is an inner product space in which every Cauchy sequence (a sequence where the elements get arbitrarily close to each other) converges to an element within the space. An orthonormal basis of a Hilbert space $\mathcal{H}$ is a set of vectors $\{e_i\}$, such that $\|e_i\|_{\mathcal{H}} = 1$ for each $i$ and $\langle e_i, e_j \rangle_{\mathcal{H}} = 0$ for each $i \neq j$. Besides, each $f \in \mathcal{H}$ can be expanded in a Fourier series:

$$\varphi = \sum_j \langle f, e_i \rangle_{\mathcal{H}} e_i.$$

**Hilbert adjoint operator.** In the context of Hilbert spaces, we can define the adjoint operator. Let $\mathcal{H}_1$ and $\mathcal{H}_2$ be Hilbert spaces, and let $A : \mathcal{H}_1 \to \mathcal{H}_2$ be a linear operator. The adjoint operator $A^* : \mathcal{H}_2 \to \mathcal{H}_1$ is defined by the property that for all

$$\langle Af, g \rangle_{\mathcal{H}_2} = \langle f, A^* g \rangle_{\mathcal{H}_1}.$$

The operator enjoys a number of important properties:

1. If $A$ is bounded, so is $A^*$, and $\|A\|_{\mathrm{op}} = \|A^*\|_{\mathrm{op}}$;

2. $(A^*)^* = A$;

3. If $A$ is invertible, so is $A^*$, and $(A^*)^{-1} = (A^{-1})^*$.

## C.3. Reproducing Kernel Hilbert Space

For any space $\mathcal{W}$, let $k_W : \mathcal{W} \times \mathcal{W} \to \mathbb{R}$ be a positive semi-definite kernel. A kernel is called *characteristic* if $\mathbb{P} \mapsto \mathbb{E}_{W \sim \mathbb{P}}[k_W(W, \cdot)]$ is injective (Fukumizu et al., 2004). We denote by $\phi_W$ its associated canonical feature map $\phi_W(w) = k_W(w, \cdot)$ for any $w \in \mathcal{W}$, and $\mathcal{H}_W$ its corresponding RKHS of real-valued functions on $W$. The space $\mathcal{H}_W$ is a Hilbert space with inner product $\langle \cdot \rangle_{\mathcal{H}_W}$ and norm $\| \cdot \|_{\mathcal{H}_W}$. It satisfies two important properties:

1. $k_W(w, \cdot) \in \mathcal{H}_W$ for all $w \in \mathcal{W}$;

2. reproducing property: for all $f \in \mathcal{H}_W$ and $w \in \mathcal{W}$, $f(w) = \langle f, k_W(w, \cdot) \rangle_{\mathcal{H}_W}$.

Since the Reproducing Kernel Hilbert Space (RKHS) is a Hilbert space, it satisfies all properties in secion C.2. Besides, we can define the kernel mean embedding, which helps to take the expectation of a function. Suppose we wish to calculate $\mathbb{E}\{f(W)\}$ for any $f \in \mathcal{H}_W$. By the reproducing property and linearity of the inner product, we have

$$\mathbb{E}\{f(W)\} = \int f(w)d\mathbb{P}(w) = \int \langle f, \phi_W(w) \rangle_{\mathcal{H}_W} d\mathbb{P}(w) = \left\langle f, \int \phi_W(w)d\mathbb{P}(w) \right\rangle_{\mathcal{H}_W} = \langle f, \mu_W \rangle_{\mathcal{H}_W}.$$

The object $\mu_W := \int \phi_W(w)d\mathbb{P}(w)$ is called the mean embedding of the distribution $\mathbb{P}(w)$. This property of RKHS implies that, to calculate the expectation of a function, it suffices to take the inner product between the function and the mean embedding of the corresponding distribution. Following this property, we can also calculate the expectation $\mathbb{E}\{f(W)g(X)\}$ for any $f \in \mathcal{H}_W$:

$$\mathbb{E}\{f(W)m(X)\} = \int f(w)m(x)d\mathbb{P}(w,x) = \int \langle f, \phi_W(w) \rangle_{\mathcal{H}_W} m(x)d\mathbb{P}(w,x) = \left\langle f, \int m(x)\phi_W(w)d\mathbb{P}(w,x) \right\rangle_{\mathcal{H}_W}.$$
$$(20)$$

Finally, we introduce properties for the norm $\| \cdot \|_{\mathcal{H}_W}$. A function $f \in \mathcal{H}_W$ if and only if $\|f\|_{\mathcal{H}_W}^2 = \langle f, f \rangle_{\mathcal{H}_W} < \infty$. Further, if $k_W(w, \cdot)$ is bounded, we have $\|f\|_{\mathcal{L}^2\{F(w)\}} \lesssim \|f\|_{\mathcal{H}_W}$. To see this, note that by Cauchy-Schwarz inequality, for any $f \in \mathcal{H}_W$, we get:

$$|f(w)|^2 = \langle k_W(w, \cdot), f \rangle_{\mathcal{H}_W}^2 \leq \|k_W(w, \cdot)\|_{\mathcal{H}_W}^2 \|f\|_{\mathcal{H}_W}^2.$$

Therefore, we have

$$\|f\|_{\mathcal{L}^2\{F(w)\}} \lesssim \|f\|_{\mathcal{H}_W}. \tag{21}$$

## C.4. Validity of optimizing (5)

Since (4) implies $\mathbb{E}[\{\varphi(Y, t) - H(W, t)\}g(X)] = 0$ holds for any measurable functions $g : \mathcal{X} \to \mathbb{R}$, we follow (Zhang et al., 2020; Mastouri et al., 2021) to take $g$ over a unit-ball of RKHS $\mathcal{H}_X$ with a fixed kernel $k^g$, and minimizes

$$R(H) = \sup_{g \in \mathcal{H}_X, \|g\| \leq 1} \left( \mathbb{E}\left[ \{\varphi(Y, t) - H(W, t)\}g(X) \right] \right)^2. \tag{22}$$

(Mastouri et al., 2021) provides an equivalent form of this risk, which is the population version of our empirical loss (5).

**Lemma C.1** (Lemma 2 in (Mastouri et al., 2021)). *Assume that* $\mathbb{E}[\{\varphi(Y, t) - H(W, t)\}^2 k_X(X, X')] < \infty$ *and denote by* $X'$ *an independent copy of the random variable* $X$. *Then* $R(H) = \mathbb{E}[\{\varphi(Y, t) - H(W, t)\}\{\varphi(Y', t) - H(W', t)\}k_X(X, X')]$.

(Zhang et al., 2020; Mastouri et al., 2021) demonstrated that if the kernel function $k_X$ derived from the conditional variable $X$ in the conditional moment equation (4) is integrally strictly positive definite (ISPD defined in Asm. C.4), continuous, and bounded, then the conditional moment equation (4) shares the same solution with $R(H)$. That means, optimizing $R(H)$ ensures us to find the right solution.

## C.5. Tikhonov regularization

In this section, we rewrite our loss (5) into the following Tikhonov regularized form, which serves as the foundation to prove Theorem 4.12.

$$\widehat{R}_\lambda(H) = \|\widehat{b}(x, t) - \widehat{A}H(w, t)(x)\|_{\mathcal{H}_X}^2 + \lambda\|H(w, t)\|_{\mathcal{H}_W}^2. \tag{23}$$

This can be achieved by reformulating the PMCR into a linear ill-posed inverse problem in the RKHS. Specifically, let $\phi_X(x) := k_X(x, \cdot)$ and $\phi_W(w) := k_W(w, \cdot)$ be the canonical feature maps. Then, by $\langle \phi_X(x), \phi_X(x') \rangle_{\mathcal{H}_X} = k_X(x, x')$, $R(H)$ of Lemma C.1 can be rewritten in terms of mean square error:

$$
\begin{aligned}
R(H) &= \|\mathbb{E}[\{\varphi(Y, t) - H(W, t)\}\phi_X(X)]\|_{\mathcal{H}_X}^2 \\
&= \|\mathbb{E}\{\varphi(Y, t)\phi_X(X)\} - \mathbb{E}\{H(W, t)\phi_X(X)\}\|_{\mathcal{H}_X}^2 \\
&= \|b(X, t) - AH(W, t)(X)\|_{\mathcal{H}_X}^2,
\end{aligned}
$$

where

$$
b(\cdot, t) := \int \varphi(y, t)\phi_X(x)p(x, y)dxdy, \quad AH(W, t)(\cdot) := \int H(w, t)\phi_X(x)p(x, w)dxdw. \tag{24}
$$

According $\phi_X(x) = k_X(x, \cdot)$, we can treat $\phi_X(x)$ as $\phi_X(x)(\cdot)$. Therefore, we have $b(x', t) := \int \varphi(y, t)k_X(x, x')p(x, y)dxdy$ and $AH(w, t)(x') := \int H(w, t)k_X(x, x')p(x, w)dxdw$ for all $x'$.

Thus, we can treat PMCR as a linear ill-posed inverse problem in the RKHS by the operator $A$. To ensure that $A$ is a bounded linear operator, we require some standard assumptions (Zhang et al., 2020; Mastouri et al., 2021):

**Assumption C.2.** $\exists c_Y < \infty, |Y| < c_Y$ a.s. and $\mathbb{E}(Y) < c_Y$.

**Assumption C.3.** **(i).** $k_X(x, \cdot)$ and $k_W(w, \cdot)$ are continuous and bounded, *i.e.*, there exists $\kappa > 0$ such that:

$$
\sup_w \|\phi_W(w)\|_{\mathcal{H}_W} \leq \kappa, \sup_x \|\phi_X(x)\|_{\mathcal{H}_X} \leq \kappa.
$$

**(ii).** Feature maps $\phi_W(W)$ and $\phi_X(X)$ are measurable. **(iii).** $\phi_W(W)$ and $\phi_X(X)$ are characteristic kernels.

**Assumption C.4.** The kernel $k_X(x, x')$ is integrally strictly positive definite (ISPD), *i.e.*, for any function $f$ that satisfies $0 < \|f\|_{\mathcal{L}^2\{F(x)\}}^2 < \infty$, we have $\iint f(x)k_X(x, x')f(x')dxdx' > 0$.

By assumptions C.2 and C.3, $b(x, t) \in \mathcal{H}_X$ and $A$ is a bounded linear operator from $\mathcal{H}_W$ to $\mathcal{H}_X$. Based on the above formulation, we can rewrite $R(H)$ of Lemma C.1 with regularized term as follow:

$$
R_\lambda(H) = \|b(x, t) - AH(w, t)(x)\|_{\mathcal{H}_X}^2 + \lambda \|H(w, t)\|_{\mathcal{H}_W}^2. \tag{25}
$$

Plugging the estimates of $\widehat{b}(x, t)$ and $\widehat{A}$ into the loss, we have (23). Based on the i.i.d. samples $(x_i, w_i, y_i)_{i=1}^n$ and $\phi_X(x_i)$, the estimates $\widehat{A}(\cdot, t)$ and $\widehat{A}$ are given by:

$$
\widehat{b}(x, t) := \frac{1}{n} \sum_{i=1}^n \varphi(y_i, t)k_X(x, x_i), \quad \widehat{A}H(W, t)(x) := \frac{1}{n} \sum_{i=1}^n H(w_i, t)k_X(x, x_i). \tag{26}
$$

In the following, we derive the minimizer of the risk (25).

Let $A^* : \mathcal{H}_X \to \mathcal{H}_W$ be an adjoint operator of $A$ such that $\langle Au, v \rangle_{\mathcal{H}_X} = \langle u, A^*v \rangle_{\mathcal{H}_W}$ for all $u \in \mathcal{H}_W$ and $v \in \mathcal{H}_X$. And we denote $\widehat{A}^*$ as an adjoint operator of $\widehat{A}$. By (Mastouri et al., 2021), for any $m(w, t) \in \mathcal{H}_W$, we have:

$$
A^*m(X, t)(w') := \int m(x, t)k_W(w, w')p(x, w)dxdw. \tag{27}
$$

Since $\phi_W(w) = k_W(w, \cdot)$, we have $A^*m(x, t)(w') := \int m(x, t)k_W(w, w')p(x, w)dxdw$ for all $w'$. The estimate $\widehat{A}$ is given by its empirical form:

$$
\widehat{A}^*m(X, t)(w') := \frac{1}{n} \sum_{i=1}^n m(x_i, t)k_W(w_i, w'). \tag{28}
$$

### C.6. Ill-posed inverse problem and solutions

Solving $R(H)$ is generally an ill-posed inverse problem, as it may not have a unique solution (Carrasco et al., 2007). We allow the *Conditional Characteristic Restrictions* (4) to be ill-posed and have non-unique solutions. Thus, the set of all solutions is given by

$$
\mathcal{H}_{W,0} = \{H(\cdot, t) \in \mathcal{H}_W : AH(\cdot, t) = b(\cdot, t)\} = H^0(\cdot, t) + \text{Ker}(A), \tag{29}
$$

where $\mathrm{Ker}(A) = \{H(\cdot, t) : AH(\cdot, t) = 0\}$ is the null space of the adjoint operator $A$. This solution consists of two parts, one is a special solution $H_t^0 \in \mathrm{Ran}(A)$, and the other is the elements of the null space.

If the solution exists, we can express the solution in the form of the singular value decomposition of $A$. Let $(\lambda_j, \varphi_j, \phi_j)_j$ be the singular value decomposition of the operator $A$. Then, if we define the orthogonal projection operator $Q : \mathcal{H}_W \to \mathrm{Ker}(A)$, we have:

$$H(\cdot, t) = \sum_j \langle H(\cdot, t), \varphi_j \rangle_{\mathcal{H}_W} \varphi_j + QH(\cdot, t) = \sum_j \frac{1}{\lambda_j} \langle b(\cdot, t), \phi_j \rangle_{\mathcal{H}_X} \varphi_j + QH(\cdot, t).$$

Thus, we target at the special solution $H^0(W, t)$, which achieves the least norm, *i.e.*,

$$H^0(W, t) = \underset{H(W,t) \in \mathcal{H}_{W,0}}{\arg\min} \|H(W, t)\|_{\mathcal{H}_W}. \tag{30}$$

By solving for $R^\lambda(H)$ of Eq. (25), we attempt to estimate the minimum norm solution $H^0(W, t)$ in (30) via the Tikhonov regularization solutions in respectively the population and in the finite sample regime:

$$H^\lambda(W, t) \quad := \quad \underset{H(W,t) \in \mathcal{H}_W}{\arg\min} \ R_\lambda(H) = \{(A^*A + \lambda I)^{-1} A^* b\}(W, t), \tag{31}$$

$$\widehat{H}^\lambda(W, t) \quad := \quad \underset{H(W,t) \in \mathcal{H}_W}{\arg\min} \ \widehat{R}_\lambda(H) = \{(\widehat{A}^* \widehat{A} + \lambda I)^{-1} \widehat{A}^* \widehat{b}\}(W, t). \tag{32}$$

# D. Proofs of Asymptotic Properties

In this section, we study the asymptotic properties of the testing statistics $\Delta_{\varphi,m}$. Since $\Delta_{\varphi,m}$ depends on $T_n(s,t)$ through (7), we first study the asymptotic properties of $T_n(s,t)$.

**Notations.** For a generic random vector $W \in \mathcal{W}$, we use $\mathcal{L}^2\{F(w)\}$ to denote the space of square integrable functions of $W$ with respect to the cumulative distribution of $W$. For any $f(W), g(W) \in \mathcal{L}^2\{F(w)\}$, we denote the $\mathcal{L}^2$-norm by $\|f\|_{\mathcal{L}^2\{F(w)\}} = \sqrt{\mathbb{E}\{f(W)^2\}}$ and inner product by $\langle f, g \rangle_{\mathcal{L}^2\{F(w)\}} = \mathbb{E}\{f(W)g(W)\}$. We use $\mathcal{H}_W$ to denote the reproducing kernel Hilbert spaces of $W$. For any $f(W), g(W) \in \mathcal{H}_W$, we denote the $\mathcal{H}_W$-norm by $\|f\|_{\mathcal{H}_W}$ and inner product by $\langle f, g \rangle_{\mathcal{H}_W}$. We let $\mathbb{P}$ denote the probability. We let $\mathbb{P}\{f(W)\} = \int f(w)d\mathbb{P}(w)$ be the expectation with respect to $W$ alone. We differentiate this from $\mathbb{E}\{f(W)\}$, which we use to denote full expectation with respect to both $W$ and data $w_1, ..., w_n$. Thus if $\widehat{H}$ depends on the data $w_1, ..., w_n$, then $\mathbb{P}\{f(W; \widehat{H})\}$ remains a function of $\widehat{H}$ (and thus the data) but $\mathbb{E}\{f(W; \widetilde{H})\}$ is a nonrandom scalar. We use both $\mathbb{P}_n$ to denote the empirical expectation with respect to $W$ given data $w_1, ..., w_n$: $\mathbb{P}_n\{f(W)\} = \frac{1}{n}\sum_{i=1}^n f(W_i)$.

For $A$, $b_t(w) := b(w,t)$ defined in (24), and $A^*$ in (28), the estimates $\widehat{A}, \widehat{b}_t(w) := \widehat{b}(w,t)$ given by (26), and the estimates $\widehat{A}^*$ is given by (28). Besides, for the operator $A$, its singular value decomposition is given by $(\lambda_n, \varphi_n, \phi_n)_{n=1}^{+\infty}$. We denote $H_t^0 = H^0(w,t)$ as the least norm solution is defined in (30). The population Tikhonov regularization solution $H_t^\lambda(w) := H^\lambda(w,t)$ and the empirical Tikhonov regularization solution $\widehat{H}_t^\lambda(w) := \widehat{H}^\lambda(w,t)$ are respectively defined in (31) and (32). Further, recall that

$$g_s = \mathbb{E}\{m(X,s)\phi_W(W)\} \text{ (assumption 4.11)}, \tag{33}$$

$$U(W,Y,t) = \varphi(Y,t) - H^0(W,t) \text{ (section 4.2)}, \tag{34}$$

$$\widehat{U}(W,Y,t) = \varphi(Y,t) - \widehat{H}^\lambda(W,t) \text{(section 4.2)}. \tag{35}$$

## D.1. Proof roadmap and key assumptions

In this section, we introduce the overview of our proof and the required assumptions to derive the asymptotic distribution of $T_n(s,t)$. Define $U(W,Y,t) = \varphi(Y,t) - H^0(W,t)$, we decompose $T_n(s,t)$ as follows:

$$
\begin{aligned}
T_n(s,t) &= \frac{1}{\sqrt{n}}\sum_{i=1}^n \widehat{U}(w_i, y_i; t)m(x_i, s) \\
&= \sqrt{n}\mathbb{P}_n\left\{\widehat{U}(W,Y,t)m(X,s)\right\} \\
&= \sqrt{n}\mathbb{P}_n\left[\left\{\varphi(Y,t) - \widehat{H}^\lambda(W,t)\right\}m(X,s)\right] \\
&= \sqrt{n}\mathbb{P}_n\left[\{\varphi(Y,t) - H^0(W,t) + H^0(W,t) - \widehat{H}^\lambda(W,t)\}m(X,s)\right] \\
&= \sqrt{n}\mathbb{P}_n\{U(W,Y,t)m(X,s)\} + \sqrt{n}\mathbb{P}\underbrace{\left[\{H^0(W,t) - \widehat{H}^\lambda(W,t)\}m(X,s)\right]}_{\text{Expected risk difference}} \\
&\quad + \sqrt{n}(\mathbb{P}_n - \mathbb{P})\underbrace{\left[\{H^0(W,t) - \widehat{H}^\lambda(W,t)\}m(X,s)\right]}_{\text{Empirical process}}.
\end{aligned}
\tag{36}
$$

To derive the asymptotic distribution of $T_n(s,t)$, we first investigate the last two terms in (36):

- **Empirical process** (Proposition D.3): $(\mathbb{P}_n - \mathbb{P})\left[\{H^0(W,t) - \widehat{H}^\lambda(W,t)\}m(X,s)\right] = o_p(n^{-1/2})$.

- **Expected risk difference** (Proposition D.4):

$$\sqrt{n}\mathbb{P}\left[\{H^0(W,t) - \widehat{H}^\lambda(W,t)\}m(X,s)\right] = -\frac{1}{\sqrt{n}}\sum_{i=1}^n U(w_i, y_i, t)\{A(A^*A)^{-1}g_s\}(x_i) + o_p(1),$$

where $g_s$ is defined in (46).

Lastly, we show that $-\frac{1}{\sqrt{n}}\sum_{i=1}^{n} U(w_i, y_i, t)\{A(A^*A)^{-1}g_s\}(x_i)$ plus the remaining term $\sqrt{n}\mathbb{P}_n\{U(W, Y, t)m(X, s)\}$ converges to the zero-man Gaussian process $\mathbb{G}_{s,t}$, *i.e.*,

$$\lim_{n\to\infty} \sqrt{n}\mathbb{P}_n\{U(W, Y, t)m(X, s)\} - \frac{1}{\sqrt{n}}\sum_{i=1}^{n} U(w_i, y_i, t)\{A(A^*A)^{-1}g_s\}(x_i) \to_d \mathbb{G}_{s,t}.$$

Since $\Delta_{\varphi,m} = \max_{t\in\mathcal{T}} \int_{\mathcal{S}} |T_n(s,t)|^2 d\mu(s)$ in (7), we show that $\Delta_{\varphi,m}$ converges to $\max_{t\in T}\int |\mathbb{G}_{s,t}|^2 d\mu(s)$ in Theorem 4.12.

Before proving these properties, we first introduce some regularity conditions. Let $\mathcal{H}_W$ denote the function space such that $H^0(W, t) \in \mathcal{H}_W$ for each $t$.

**Assumption D.1.** Let $N_{[\cdot]}(\epsilon, \mathcal{H}_W, \|\cdot\|_{\mathcal{H}_W})$ be the bracketing number of size $\epsilon$ of $\mathcal{H}_W$. We assume $\int_0^1 \sqrt{\log N_{[\cdot]}(\epsilon, \mathcal{H}_W, \|\cdot\|_{\mathcal{L}^2\{F(w)\}})}d\epsilon < \infty$ and $\mathbb{P}(\widehat{H}^\lambda \in \mathcal{H}_W) \to 1$.

**Assumption D.2.** Let $(\lambda_j, \varphi_j, \phi_j)_j$ be the singular value decomposition of the operator $A$ described in section C. Then we assume: (a). For some $\eta > 2$, $\sum_j \lambda_j^{-2\eta}|\langle g_s, \varphi_j\rangle_{\mathcal{H}_W}|^2 < \infty$; (b) For some $\theta \geq 2$, $\sum_j \lambda_j^{-2\theta}|\langle H_t^0, \varphi_j\rangle_{\mathcal{H}_W}|^2 < \infty$.

Assumption D.1 restricts the complexity of $\mathcal{H}_W$ and ensures $\mathcal{H}_W$ is a $P$-Donsker class (vd Vaart, 1998), which was a standard assumption to analyze the empirical process (Beyhum et al., 2024; Lapenta & Lavergne, 2022).

Assumption D.2 is the source condition that is commonly assumed in nonparametric regression (Carrasco et al., 2007; Florens et al., 2012). These have also been employed in (Florens et al., 2012; Beyhum et al., 2024) to obtain a faster convergence rate for nonparametric instrumental regression. Here, we require $g_s$ and $H_t^0$ to satisfy the source condition, for establishing the asymptotic properties of the statistic in examining the integral equation. Since $g_s := \mathbb{E}\{m(X, s)\phi_W(W)\}$, the source condition for $g_s$ puts requirement on the smoothness for the space $\mathcal{H}_W$ when $m(\cdot, s)$ is chosen properly.

Compared to (Beyhum et al., 2024; Mastouri et al., 2021) that allowed $\eta = 2$, we require $\eta > 2$ in the source condition for $g_s$. This is because, after investigating the proof of (Beyhum et al., 2024; Mastouri et al., 2021), we find that their conclusion may not hold for $\eta = 2$. Specifically, similar to our analysis, a key step in their proofs that related to the source condition is $\|(\lambda I + A^*A)^{-1}A^*A\widetilde{g} - \widetilde{g}\|_{\mathcal{H}_W} = o_p(1)$ (in our scenario, it refers to (51)) for $\widetilde{g} \in \mathcal{H}_W$. Here, $\widetilde{g} := \sum_j \lambda_j^{-2}\langle g, \varphi_j\rangle\varphi_j$ ($\widetilde{g}$ refers to $\widetilde{g}_s$ in our scenario), where the source condition was assumed on $g$ ($g_s$ in our scenario). To prove the property, they leveraged the property in Lemma 2.5 (d) of (Beyhum et al., 2024) that $\|(\lambda I + A^*A)^{-1}A^*A\widetilde{g} - \widetilde{g}\|_{\mathcal{H}_W} = O_p\left\{\lambda^{\frac{\min(\gamma, 2)}{2}}\right\}$ for $\widetilde{g}$ such that $\|\widetilde{g}\|_{W,\gamma}^2 := \sum_j \lambda_j^{-2\gamma}|\langle g, \varphi_j\rangle|^2 < \infty$. Since $g$ satisfies the source condition and $\widetilde{g} := \sum_j \lambda_j^{-2}\langle g, \varphi_j\rangle\varphi_j$, to apply the above property, we should take $\gamma \leq \eta - 2$ to ensure that $\|\widetilde{g}\|_{W,\gamma}^2 < \infty$. That means, we should take $\eta > 2$ to make that $\gamma = \eta - 2 > 0$, in order to ensure that $O_p\left\{\lambda^{\frac{\min(\gamma, 2)}{2}}\right\} = o_p(1)$. Besides, while this condition is stronger than the case of $\eta = 2$, we emphasize that requiring $\eta > 2$ only slightly increases the smoothness requirement of the $\mathcal{H}_W$, making it a modestly stronger condition.

## D.2. Empirical process

**Proposition D.3.** *Under assumptions 4.9, C.2-C.4, and D.1-D.2, the empirical process* $\sqrt{n}(\mathbb{P}_n - \mathbb{P})[\{H^0(W, t) - \widehat{H}^\lambda(W, t)\}m(X, s)] = o_p(1)$.

*Proof.* We first proof $\|\{H^0(W, t) - \widehat{H}^\lambda(W, t)\}m(X, s)\|_{\mathcal{L}^2\{F(x,w)\}}^2 = o_p(1)$. In fact, we have

$$\|\{H^0(W, t) - \widehat{H}^\lambda(W, t)\}m(X, s)\|_{\mathcal{L}^2\{F(x,w)\}}^2 = \int \{H^0(W, t) - \widehat{H}^\lambda(W, t)\}^2|m(X, s)|^2 d\mathbb{P}(W, X)$$

$$= \int \{H^0(W, t) - \widehat{H}^\lambda(W, t)\}^2\mathbb{E}\{|m(X, s)|^2|W\}d\mathbb{P}(W)$$

$$\overset{(1)}{\leq} C\|H^0(\cdot, t) - \widehat{H}^\lambda(\cdot, t)\|_{\mathcal{L}^2\{F(w)\}}^2 \overset{(2)}{\lesssim} \|H^0(\cdot, t) - \widehat{H}^\lambda(\cdot, t)\|_{\mathcal{H}_W}^2,$$

where (1) follows from assumption 4.9 and (2) follows from (21) by assumption C.3. Since assumptions C.2-C.4 and D.2 are satisfied, we have $\|H^0(w, t) - \widehat{H}^\lambda(w, t)\|_{\mathcal{H}_W}^2 = o_p(1)$ by Lemma D.17. Therefore, all conditions in Lemma D.14 are satisfied and we obtain

$$\sqrt{n}(\mathbb{P}_n - \mathbb{P})[\{H^0(W, t) - \widehat{H}^\lambda(W, t)\}m(X, s)] = o_p(1).$$

The proof is completed. $\square$

### D.3. Expected risk difference

Proposition D.4 is our main result in this section, whose proof is decomposed into Lemmas D.5-D.9.

**Proposition D.4.** *Under assumptions 4.10, C.2–C.3, and D.2, the expected risk difference term has:*

$$\sqrt{n}\mathbb{P}\left[\left\{H^0(W,t) - \widehat{H}^\lambda(W,t)\right\}m(X,s)\right] = -\frac{1}{\sqrt{n}}\sum_{i=1}^{n}U(w_i,y_i,t)\{A(A^*A)^{-1}g_s\}(x_i) + o_p(1).$$

*Proof.* For simplicity, we denote $H_t^0(w) := H^0(w,t)$, $b_t(x) := b(x,t)$. Based on the interpretation of PMCR as a linear ill-posed problem and the form of Tikhonov regularization solutions in (31)–(32), we have the following decomposition (Babii & Florens, 2017; 2020):

$$\widehat{H}^\lambda(w,t) - H^0(w,t) = G_1 + G_2 + G_3 + G_4 + G_5, \tag{37}$$

where

$$G_1 := (\lambda I + A^*A)^{-1}A^*(\widehat{b}_t - \widehat{A}H_t^0); \tag{38}$$

$$G_2 := (\lambda I + A^*A)^{-1}(\widehat{A}^* - A^*)(\widehat{b}_t - \widehat{A}H_t^0); \tag{39}$$

$$G_3 := \left\{(\lambda I + \widehat{A}^*\widehat{A})^{-1} - (\lambda I + A^*A)^{-1}\right\}\widehat{A}^*(\widehat{b}_t - \widehat{A}H_t^0); \tag{40}$$

$$G_4 := (\lambda I + \widehat{A}^*\widehat{A})^{-1}\widehat{A}^*\widehat{A}H_t^0 - (\lambda I + A^*A)^{-1}A^*b_t; \tag{41}$$

$$G_5 := (\lambda I + A^*A)^{-1}A^*b_t - H_t^0. \tag{42}$$

Therefore, we have

$$\sqrt{n}\mathbb{P}[\{\widehat{H}^\lambda(W,t) - H^0(W,t)\}m(X,s)] = \sum_{i=1}^{5}S_{ni}(s,t),$$

where $S_{ni}(s,t)$ is define as $\sqrt{n}\mathbb{P}\{G_i m(X,s)\}$. By applying Lemmas D.9, D.5, D.6, D.7 and D.8 to $S_{n1}(s,t)$, $S_{n2}(s,t)$, $S_{n3}(s,t)$, $S_{n4}(s,t)$ and $S_{n5}(s,t)$, respectively, we have:

$$\sqrt{n}\mathbb{P}[\{H^0(W,t) - \widehat{H}^\lambda(W,t)\}m(X,s)] = -\frac{1}{\sqrt{n}}\sum_{i=1}^{n}U(w_i,y_i,t)\{A(A^*A)^{-1}g_s\}(x_i) + o_p(1).$$

The proof is completed. $\square$

Next, we provide proofs for Lemmas D.5–D.8.

**Lemma D.5.** *Under assumptions C.2, C.3 and D.2, $S_{n2}(s,t) = o_p(1)$ as $n \to \infty$.*

*Proof.* By the reproducing property that $f(w) = \langle f, k_W(w,\cdot)\rangle_{\mathcal{H}_W}$ for each $f \in \mathcal{H}_W$, we have $(\lambda I + A^*A)^{-1}(\widehat{A}^* - A^*)(\widehat{b}_t - \widehat{A}H_t^0)(w) = \langle(\lambda I + A^*A)^{-1}(\widehat{A}^* - A^*)(\widehat{b}_t - \widehat{A}H_t^0), k_W(w,\cdot)\rangle_{\mathcal{H}_W}$. Therefore, we have the following for $S_{n2}(s,t) := \sqrt{n}\mathbb{P}\{G_2 m(X,s)\}$:

$$
\begin{aligned}
|\mathbb{P}\{G_2 m(X,s)\}| &= \left|\mathbb{E}\left\{(\lambda I + A^*A)^{-1}(\widehat{A}^* - A^*)(\widehat{b}_t - \widehat{A}H_t^0)(W) \cdot m(X,s)\right\}\right| \\
&= \left|\mathbb{E}\left\{\langle(\lambda I + A^*A)^{-1}(\widehat{A}^* - A^*)(\widehat{b}_t - \widehat{A}H_t^0), \phi_W(W)\rangle_{\mathcal{H}_W} \cdot m(X,s)\right\}\right| \\
&\overset{(1)}{=} \left|\langle(\lambda I + A^*A)^{-1}(\widehat{A}^* - A^*)(\widehat{b}_t - \widehat{A}H_t^0), \mathbb{E}\{m(X,s)\phi_W(W)\}\rangle_{\mathcal{H}_W}\right| \\
&\overset{(2)}{=} \left|\langle(\widehat{A}^* - A^*)(\widehat{b}_t - \widehat{A}H_t^0), (\lambda I + A^*A)^{-1}\mathbb{E}\{m(X,s)\phi_W(W)\}\rangle_{\mathcal{H}_W}\right|,
\end{aligned}
$$

where (1) follows from (20) and (2) follows from $\{(\lambda I + A^*A)^{-1}\}^* = (\lambda I + A^*A)^{-1}$ in Sec. C.2. Thus, we have:

$$
\begin{aligned}
|\mathbb{P}\{G_2 m(X,s)\}| &\overset{(1)}{\leq} \|(\widehat{A}^* - A^*)(\widehat{b}_t - \widehat{A}H_t^0)\|_{\mathcal{H}_W} \cdot \|(\lambda I + A^*A)^{-1}\mathbb{E}\{m(X,s)\phi_W(W)\}\|_{\mathcal{H}_W} \\
&\leq \|\widehat{A}^* - A^*\|_{\mathrm{op}} \cdot \|\widehat{b}_t - \widehat{A}H_t^0\|_{\mathcal{H}_X} \cdot \|(\lambda I + A^*A)^{-1}\mathbb{E}\{m(X,s)\phi_W(W)\}\|_{\mathcal{H}_W} \\
&\overset{(2)}{=} \|\widehat{A} - A\|_{\mathrm{op}} \cdot \|\widehat{b}_t - \widehat{A}H_t^0\|_{\mathcal{H}_X} \cdot \|(\lambda I + A^*A)^{-1}\mathbb{E}\{m(X,s)\phi_W(W)\}\|_{\mathcal{H}_W},
\end{aligned}
$$

where (1) follows from the Cauchy-Schwartz inequality and (2) follows from $\|A^*\|_{\mathrm{op}} = \|A\|_{\mathrm{op}}$ in Sec. C.4. By Lemmas D.12, we have $\|\widehat{b}_t - b_t\|_{\mathcal{H}_X} = O_p(1/\sqrt{n})$ and $\|A - \widehat{A}\|_{\mathrm{op}} = O_p(1/\sqrt{n})$. With these properties, we have:

$$
\begin{aligned}
\|\widehat{b}_t - \widehat{A}H_t^0\|_{\mathcal{H}_X} &= \|\widehat{b}_t - b_t + b_t - AH_t^0 + AH_t^0 - \widehat{A}H_t^0\|_{\mathcal{H}_X} \\
&= \|\widehat{b}_t - b_t + AH_t^0 - \widehat{A}H_t^0\|_{\mathcal{H}_X} \\
&\leq \|\widehat{b}_t - \widehat{A}H_t^0\|_{\mathcal{H}_X} + \|AH_t^0 - \widehat{A}H_t^0\|_{\mathcal{H}_X} \\
&\leq \|\widehat{b}_t - b_t\|_{\mathcal{H}_X} + \|A - \widehat{A}\|_{\mathrm{op}} \cdot \|H_t^0\|_{\mathcal{H}_W} = O_p(1/\sqrt{n}).
\end{aligned}
\tag{43}
$$

By assumption D.2 (a) with $g_s := \mathbb{E}\{m(X,s)\phi_W(W)\}$, we apply Lemma D.10 (d) to obtain that $\|(\lambda I + A^*A)^{-1}\mathbb{E}\{m(X,s)\phi_W(W)\}\|_{\mathcal{H}_W} = O_p\{\lambda^{\frac{\min(\eta,2)}{2}-1}\}$. Combining all the inequalities, we get

$$
\sqrt{n}\,|\mathbb{P}\{G_2 m(X,s)\}| \leq \sqrt{n} \cdot O_p\left(\frac{1}{\sqrt{n}}\right) \cdot O_p\left(\frac{1}{\sqrt{n}}\right) \cdot O_p\{\lambda^{\frac{\min(\eta,2)}{2}-1}\}.
\tag{44}
$$

Since we require $\eta \geq 2$ in assumption D.2 (a), the last term is $o_p(1)$. $\qquad\square$

**Lemma D.6.** *Under assumptions 4.10, C.2, C.3 and D.2, $S_{n3}(s,t) = o_p(1)$ as $n \to \infty$.*

*Proof.* By Lemma D.16, we have:

$$
\begin{aligned}
G_3 &= \left\{(\lambda I + \widehat{A}^*\widehat{A})^{-1} - (\lambda I + A^*A)^{-1}\right\}\widehat{A}^*(\widehat{b}_t - \widehat{A}H_t^0) \\
&= (\lambda I + A^*A)^{-1}(A^*A - \widehat{A}^*\widehat{A})(\lambda I + \widehat{A}^*\widehat{A})^{-1}\widehat{A}^*(\widehat{b}_t - \widehat{A}H_t^0).
\end{aligned}
$$

By the reproducing property that $f(w) = \langle f, k_W(w,\cdot)\rangle_{\mathcal{H}_W}$ for any $f \in \mathcal{H}_W$, we have $(\lambda I + A^*A)^{-1}(A^*A - \widehat{A}^*\widehat{A})(\lambda I + \widehat{A}^*\widehat{A})^{-1}\widehat{A}^*(\widehat{b}_t - \widehat{A}H_t^0)(w) = \langle(\lambda I + A^*A)^{-1}(A^*A - \widehat{A}^*\widehat{A})(\lambda I + \widehat{A}^*\widehat{A})^{-1}\widehat{A}^*(\widehat{b}_t - \widehat{A}H_t^0), k_W(w,\cdot)\rangle_{\mathcal{H}_W}$. Therefore, we have the following for $S_{n3}(s,t) := \sqrt{n}\mathbb{P}\{G_3 m(X,s)\}$:

$$
\begin{aligned}
|\mathbb{P}\{G_3 m(X,s)\}| &= \left|\mathbb{E}\left[(\lambda I + A^*A)^{-1}(A^*A - \widehat{A}^*\widehat{A})(\lambda I + \widehat{A}^*\widehat{A})^{-1}\widehat{A}^*(\widehat{b}_t - \widehat{A}H_t^0)(W)m(X,s)\right]\right| \\
&= \left|\mathbb{E}\left\{\langle(\lambda I + A^*A)^{-1}(A^*A - \widehat{A}^*\widehat{A})(\lambda I + \widehat{A}^*\widehat{A})^{-1}\widehat{A}^*(\widehat{b}_t - \widehat{A}H_t^0), \phi_W(W)\rangle_{\mathcal{H}_W} \cdot m(X,s)\right\}\right| \\
&\overset{(1)}{=} \left|\langle(\lambda I + A^*A)^{-1}(A^*A - \widehat{A}^*\widehat{A})(\lambda I + \widehat{A}^*\widehat{A})^{-1}\widehat{A}^*(\widehat{b}_t - \widehat{A}H_t^0), \mathbb{E}\{m(X,s)\phi_W(W)\}\rangle_{\mathcal{H}_W}\right| \\
&\overset{(2)}{=} \left|\langle\widehat{b}_t - \widehat{A}H_t^0, \widehat{A}(\lambda I + \widehat{A}^*\widehat{A})^{-1}(A^*A - \widehat{A}^*\widehat{A})(\lambda I + A^*A)^{-1}\mathbb{E}\{m(X,s)\phi_W(W)\}\rangle_{\mathcal{H}_X}\right|,
\end{aligned}
$$

where (1) follows from (20), and (2) follows from $\{(\lambda I + A^*A)^{-1}\}^* = (\lambda I + A^*A)^{-1}$ and $\{(\lambda I + \widehat{A}^*\widehat{A})^{-1}\}^* = (\lambda I + \widehat{A}^*\widehat{A})^{-1}$. Thus, we have:

$$
\begin{aligned}
|\mathbb{P}\{G_3 m(X,s)\}| &\overset{(1)}{\leq} \|\widehat{b}_t - \widehat{A}H_t^0\|_{\mathcal{H}_X} \cdot \|\widehat{A}(\lambda I + \widehat{A}^*\widehat{A})^{-1}(A^*A - \widehat{A}^*\widehat{A})(\lambda I + A^*A)^{-1}\mathbb{E}\{m(X,s)\phi_W(W)\}\|_{\mathcal{H}_X} \\
&\overset{(2)}{\leq} \|\widehat{b}_t - \widehat{A}H_t^0\|_{\mathcal{H}_X} \cdot \|\widehat{A}(\lambda I + \widehat{A}^*\widehat{A})^{-1}\|_{\mathrm{op}} \cdot \|A^*A - \widehat{A}^*\widehat{A}\|_{\mathrm{op}} \cdot \|(\lambda I + A^*A)\mathbb{E}\{m(X,s)\phi_W(W)\}\|_{\mathcal{H}_W},
\end{aligned}
$$

where (1) follows from the Cauchy-Schwartz inequality and (2) follows from operator norm inequality.

According to the paragraph above equation (97) in (Mastouri et al., 2021), $\widehat{A}$ is a compact operator. By Lemma D.10 (c), we have $\|\widehat{A}(\lambda I + \widehat{A}^*\widehat{A})^{-1}\|_{\mathrm{op}} = O_p(1/\sqrt{\lambda})$, $\|(\lambda I + \widehat{A}^*\widehat{A})^{-1}\widehat{A}^*\|_{\mathrm{op}} = O_p(1/\sqrt{\lambda})$. According to assumption D.2

(a) and Lemma D.10 (d), we have $\|(\lambda I + A^*A)^{-1}\mathbb{E}\{m(X,s)\phi_W(W)\}\| = O_p\{\lambda^{\frac{\min(\eta,2)}{2}-1}\}$. Finally, By Lemmas D.12, we have $\|\widehat{b}_t - b_t\|_{\mathcal{H}_X} = O_p(1/\sqrt{n})$ and $\|A - \widehat{A}\|_{\mathrm{op}} = O_p(1/\sqrt{n})$. With these properties, we can similarly obtain that $\|\widehat{b}_t - \widehat{A}H_t^0\|_{\mathcal{H}_X} \leq \|\widehat{b}_t - b_t\|_{\mathcal{H}_X} + \|A - \widehat{A}\|_{\mathrm{op}} \cdot \|H_t^0\|_{\mathcal{H}_W} = O_p(1/\sqrt{n})$. Combining all the inequalities and $\eta \geq 2$ in assumption D.2 (a), we get

$$\sqrt{n}\,|\mathbb{P}\{G_3 m(X,s)\}| = \sqrt{n} \cdot O_p\left(\frac{1}{\sqrt{n}}\right) \cdot O_p\left(\frac{1}{\sqrt{\lambda}}\right) \cdot O_p\left(\frac{1}{\sqrt{n}}\right) \cdot O_p\{\lambda^{\frac{\min(\eta,2)}{2}-1}\}$$
$$= O_p\left(\frac{1}{\sqrt{n\lambda}}\right) \overset{(*)}{=} o_p(1),$$

(45)

where $(*)$ follows from assumption 4.10. $\qquad\square$

**Lemma D.7.** *Under assumptions 4.10, C.2, C.3 and D.2, $S_{n4}(s,t) = o_p(1)$ as $n \to \infty$.*

*Proof.* By the reproducing property, we have $\{(\lambda I + \widehat{A}^*\widehat{A})^{-1}\widehat{A}^*\widehat{A}H_t^0 - (\lambda I + A^*A)^{-1}A^*b_t\}(w) = \langle((\lambda I + \widehat{A}^*\widehat{A})^{-1}\widehat{A}^*\widehat{A}H_t^0 - (\lambda I + A^*A)^{-1}A^*b_t, k_W(w,\cdot)\rangle_{\mathcal{H}_W}$. Therefore, we have the following for $S_{n4}(s,t)$:

$$|\mathbb{P}\{G_4 m(X,s)\}| = \left|\mathbb{E}\left[\{(\lambda I + \widehat{A}^*\widehat{A})^{-1}\widehat{A}^*\widehat{A}H_t^0 - (\lambda I + A^*A)^{-1}A^*b_t\}(W) \cdot m(X,s)\right]\right|$$
$$= \left|\mathbb{E}\left[\langle(\lambda I + \widehat{A}^*\widehat{A})^{-1}\widehat{A}^*\widehat{A}H_t^0 - (\lambda I + A^*A)^{-1}A^*b_t, \phi_W(W)\rangle_{\mathcal{H}_W} \cdot m(X,s)\right]\right|$$
$$\overset{(1)}{=} \left|\langle(\lambda I + \widehat{A}^*\widehat{A})^{-1}\widehat{A}^*\widehat{A}H_t^0 - (\lambda I + A^*A)^{-1}A^*b_t, \mathbb{E}\{m(X,s)\phi_W(W)\}\rangle_{\mathcal{H}_W}\right|,$$

where (1) follows from (20). To establish the upper bound for the right-hand side, we derive the form of the operator $A^*A$. First, for the operator $A : \mathcal{H}_W \to \mathcal{H}_X$ defined in (24), its singular value decomposition given by $(\lambda_n, \varphi_n, \phi_n)_{n=1}^{+\infty}$. For $g_s(w') := \mathbb{E}[m(X,s)\phi_W(W)](w')$, we first proof $\|g_s\|_{\mathcal{H}_W} < \infty$. In fact, since $\mathbb{E}\{m(X,s)|W\}$ and kernel $k$ are assumed to bounded in assumption C.2, we have:

$$\|g_s\|_{\mathcal{H}_W} = \|\mathbb{E}\{m(X,s)\phi_W(W)\}\|_{\mathcal{H}_W} = \|\mathbb{E}\left[\mathbb{E}\{m(X,s)|W\}\phi_W(W)\right]\|_{\mathcal{H}_W}$$
$$\leq C\|\mathbb{E}\{\phi_W(W)\}\|_{\mathcal{H}_W} = C\sqrt{\langle\mathbb{E}\{\phi_W(W)\}, \mathbb{E}\{\phi_W(W)\}\rangle_{\mathcal{H}_W}}$$
$$= C\sqrt{\mathbb{E}\left\{\langle\phi_W(W), \phi_W(W')\rangle_{\mathcal{H}_W}\right\}} = C\sqrt{\mathbb{E}\{k_W(W,W')\}} < \infty.$$

(46)

Thus, $\|g_s\|_{\mathcal{H}_W} < \infty$ imply $g_s \in \mathcal{H}_W$, which can write the expansion of the basis functions as $g_s = \sum_j \langle g_s, \varphi_j\rangle_{\mathcal{H}_W}\varphi_j$ by sec. C.2. Besides, by the properties of singular value decomposition, we have $A\varphi_j = \lambda_j\phi_j$ and $A^*\phi_j = \lambda_j\varphi_j$. Therefore, for each $g_s \in \mathcal{H}_W$, we have:

$$A^*Ag_s = \sum_j \lambda_j^2\langle g_s, \varphi_j\rangle_{\mathcal{H}_W}\varphi_j.$$

(47)

Let $\widetilde{g}_s(w') := \sum_j \lambda_j^{-2}\langle g_s, \varphi_j\rangle_{\mathcal{H}_W}\varphi_j$. By assumption D.2, we have:

$$\|\widetilde{g}_s\|_{\mathcal{H}_W}^2 = \left\langle\sum_j \lambda_j^{-2}\langle g_s, \varphi_j\rangle_{\mathcal{H}_W}\varphi_j, \sum_j \lambda_j^{-2}\langle g_s, \varphi_j\rangle_{\mathcal{H}_W}\varphi_j\right\rangle_{\mathcal{H}_W} = \sum_j \lambda_j^{-4}|\langle g_s, \varphi_j\rangle_{\mathcal{H}_W}|^2 < \infty.$$

(48)

Besides, we have:

$$A^*A\widetilde{g}_s = \sum_j\sum_i \lambda_j^2\lambda_i^{-2}\langle g_s, \varphi_i\rangle_{\mathcal{H}_W}\langle\varphi_i, \varphi_j\rangle_{\mathcal{H}_W}\psi_j = \sum_j \langle g_s, \varphi_j\rangle_{\mathcal{H}_W}\varphi_j = g_s.$$

Thus, if we define $P_t := (\lambda I + \widehat{A}^*\widehat{A})^{-1}\widehat{A}^*\widehat{A}H_t^0 - (\lambda I + A^*A)^{-1}A^*b_t$, then by the Cauchy-Schwartz inequality, we have:

$$|\mathbb{P}\{G_4 m(X,s)\}| = |\langle P_t, \mathbb{E}\{m(X,s)\phi_W(W)\}\rangle_{\mathcal{H}_W}| = |\langle P_t, A^*A\widetilde{g}_s\rangle_{\mathcal{H}_W}|$$
$$\leq \left|\langle P_t, (A^* - \widehat{A}^*)A\widetilde{g}_s\rangle_{\mathcal{H}_W}\right| + \left|\langle P_t, \widehat{A}^*A\widetilde{g}_s\rangle_{\mathcal{H}_W}\right|$$
$$= \left|\langle P_t, (A^* - \widehat{A}^*)A\widetilde{g}_s\rangle_{\mathcal{H}_W}\right| + \left|\langle\widehat{A}P_t, A\widetilde{g}_s\rangle_{\mathcal{H}_W}\right|$$
$$\overset{(1)}{\leq} \|P_t\|_{\mathcal{H}_W} \cdot \|\widehat{A} - A\|_{\mathrm{op}} \cdot \|A\widetilde{g}_s\|_{\mathcal{H}_X} + \|\widehat{A}P_t\|_{\mathcal{H}_X} \cdot \|A\widetilde{g}_s\|_{\mathcal{H}_X},$$

where (1) follows from $\|\widehat{A}^* - A^*\|_{\mathrm{op}} = \|\widehat{A} - A\|_{\mathrm{op}}$. By $AH_t^0 = b_t$, we can decompose $P_t$ as follows by Lemma D.16:

$$
\begin{aligned}
P_t &= (\lambda I + \widehat{A}^*\widehat{A})^{-1}\widehat{A}^*\widehat{A}H_t^0 - (\lambda I + A^*A)^{-1}A^*AH_t^0 \\
&= \left\{ (\lambda I + \widehat{A}^*\widehat{A})^{-1}(\lambda I + \widehat{A}^*\widehat{A} - \lambda I) - (\lambda I + A^*A)^{-1}(\lambda I + A^*A - \lambda I) \right\} H_t^0 \\
&= \lambda \left\{ (\lambda I + A^*A)^{-1} - (\lambda I + \widehat{A}^*\widehat{A})^{-1} \right\} H_t^0 \\
&= \lambda(\lambda I + \widehat{A}^*\widehat{A})^{-1}\{\widehat{A}^*\widehat{A} - A^*A\}(\lambda I + A^*A)^{-1}H_t^0.
\end{aligned}
$$

Therefore, we have:

$$
\begin{aligned}
\|P_t\|_{\mathcal{H}_W} &= \|\lambda(\lambda I + \widehat{A}^*\widehat{A})^{-1}(\widehat{A}^*\widehat{A} - A^*A)(\lambda I + A^*A)^{-1}H_t^0\|_{\mathcal{H}_W} \\
&\le \|\lambda(\lambda I + \widehat{A}^*\widehat{A})^{-1}\|_{\mathrm{op}} \cdot \|A^*A - \widehat{A}^*\widehat{A}\|_{\mathrm{op}} \cdot \|(\lambda I + A^*A)^{-1}H_t^0\|_{\mathcal{H}_W}.
\end{aligned}
$$

Since $\widehat{A}$ is a compact operator as shown in the proof of Lemma D.5, by Lemma D.10 (b), we have $\|(\lambda(\lambda I + \widehat{A}^*\widehat{A})^{-1}\|_{\mathrm{op}} \le 2$. By assumption D.2 (b), we can apply Lemma D.10 (d) to obtain that $\|(\lambda I + A^*A)^{-1}H_t^0\|_{\mathcal{H}_W} = O_p\{\lambda^{\frac{\min(\theta,2)}{2}-1}\}$. Finally, by Lemma D.13, we have $\|A^*A - \widehat{A}^*\widehat{A}\|_{\mathrm{op}} = O_p(1/\sqrt{n})$. Combining all the inequalities, we get

$$
\|P_t\|_{\mathcal{H}_W} \le 2 \cdot O_p\left(\frac{1}{\sqrt{n}}\right) \cdot \lambda^{\frac{\min(\theta,2)}{2}-1}.
$$

Next, we prove $\|A\widetilde{g}_s\|_{\mathcal{H}_X} < \infty$. In fact, since $A$ is bounded linear operators, we have $\|A\|_{\mathrm{op}} < \infty$. Besides, by Eq. (48), we have $\|\widetilde{g}_s\|_{\mathcal{H}_W} < \infty$. Thus, we have $\|A\widetilde{g}_s\|_{\mathcal{H}_X} \le \|A\|_{\mathrm{op}} \cdot \|\widetilde{g}_s\|_{\mathcal{H}_W} < \infty$.

Next we provide the bound for $\|\widehat{A}P_t\|_{\mathcal{H}_X}$.

$$
\begin{aligned}
\|\widehat{A}P_t\|_{\mathcal{H}_X} &= \|\lambda\widehat{A}(\lambda I + \widehat{A}^*\widehat{A})^{-1}(\widehat{A}^*\widehat{A} - A^*A)(\lambda I + A^*A)^{-1}H_t^0\|_{\mathcal{H}_X} \\
&\le \lambda \cdot \|\widehat{A}(\lambda I + \widehat{A}^*\widehat{A})^{-1}\|_{\mathrm{op}} \cdot \|\widehat{A}^*\widehat{A} - A^*A\|_{\mathrm{op}} \cdot \|(\lambda I + A^*A)^{-1}H_t^0\|_{\mathcal{H}_W}.
\end{aligned}
$$

Since $\widehat{A}$ is a compact operator, by Lemma D.10 (c), we have $\|\widehat{A}(\lambda I + \widehat{A}^*\widehat{A})^{-1}\|_{\mathrm{op}} = O_p(1/\sqrt{\lambda})$, $\|(\lambda I + \widehat{A}^*\widehat{A})^{-1}\widehat{A}^*\|_{\mathrm{op}} = O_p(1/\sqrt{\lambda})$. By assumption D.2 (2), we can apply Lemma D.10 (d) to have $\|(\lambda I + A^*A)^{-1}H_t^0\|_{\mathcal{H}_W} = O_p\{\lambda^{\frac{\min(\theta,2)}{2}-1}\}$. Finally, by Lemma D.13, we have $\|A^*A - \widehat{A}^*\widehat{A}\|_{\mathrm{op}} = O_p(1/\sqrt{n})$. Combining all the inequalities, we get:

$$
\|\widehat{A}P_t\| \le \lambda \cdot O_p\left(\frac{1}{\sqrt{\lambda}}\right) \cdot O_p\left(\frac{1}{\sqrt{n}}\right) \cdot O_p\{\lambda^{\frac{\min(\theta,2)}{2}-1}\}.
$$

Therefore, substituting all inequalities to $S_{n4}(s,t)$ and $\theta \ge 2$ in assumption D.2 (b), we have

$$
\begin{aligned}
\sqrt{n}\,|\mathbb{P}\{G_4m(X,s)\}| &\le \sqrt{n} \cdot \|P_t\|_{\mathcal{H}_W} \cdot \|\widehat{A} - A\|_{\mathrm{op}} \cdot \|A\widetilde{g}_s\|_{\mathcal{H}_X} + \sqrt{n} \cdot \|\widehat{A}P_t\|_{\mathcal{H}_X} \cdot \|A\widetilde{g}_s\|_{\mathcal{H}_X} \\
&\le \sqrt{n} \cdot \left\{ 2 \cdot O_p\left(\frac{1}{\sqrt{n}}\right) \cdot \lambda^{\frac{\min(\theta,2)}{2}-1} \right\} \cdot O_p\left(\frac{1}{\sqrt{n}}\right) + \sqrt{n} \cdot \left\{ \lambda \cdot \lambda^{-1/2} \cdot O_p\left(\frac{1}{\sqrt{n}}\right) \cdot \lambda^{\frac{\min(\theta,2)}{2}-1} \right\} \\
&= O_p\left(\frac{1}{\sqrt{n}}\right) + \lambda^{1/2} = o_p(1) + \lambda^{1/2}.
\end{aligned}
$$

$$(49)$$

By assumption 4.10, the last term is $o_p(1)$. $\qquad\square$

**Lemma D.8.** *Under assumptions 4.10, C.2, C.3 and D.2, $S_{n5}(s,t) = o_p(1)$ as $n \to \infty$.*

*Proof.* By the reproducing property, we have $\{(\lambda I + A^*A)^{-1}A^*b_t - H_t^0\}(w) = \langle(\lambda I + A^*A)^{-1}A^*b_t - H_t^0, k_W(w,\cdot)\rangle_{\mathcal{H}_W}$. Thus, we have the following for $S_{n5}(s,t)$:

$$
\begin{aligned}
|\mathbb{P}\{G_5m(X,s)\}| &= \left| \mathbb{E}\left[ \{(\lambda I + A^*A)^{-1}A^*b_t - H_t^0\}(W) \cdot m(X,s) \right] \right| \\
&= \left| \mathbb{E}\left[ \langle(\lambda I + A^*A)^{-1}A^*b_t - H_t^0, \phi_W(W)\rangle_{\mathcal{H}_W} \cdot m(X,s) \right] \right| \\
&\overset{(1)}{=} \left| \langle(\lambda I + A^*A)^{-1}A^*b_t - H_t^0, \mathbb{E}\{m(X,s)\phi_W(W)\}\rangle_{\mathcal{H}_W} \right|,
\end{aligned}
$$

where (1) follows from (20).

By assumption D.2 (b) and $AH_t^0 = b_t$, we can apply Lemma D.11 to obtain that $\|(\lambda I + A^*A)^{-1}A^*b_t - H_t^0\|_{\mathcal{H}_W} = \|(\lambda I + A^*A)^{-1}A^*AH_t^0 - H_t^0\|_{\mathcal{H}_W} = O_p\{\lambda^{\frac{\min(\theta,2)}{2}}\}$. Combining this rate with the Cauchy-Schwartz inequality and and $\theta \geq 2$ in assumption D.2 (a), we have

$$
\begin{aligned}
\sqrt{n}\,|\mathbb{P}\{G_5 m(X,s)\}| &= \sqrt{n}\,\big|\langle(\lambda I + A^*A)^{-1}A^*b_t - H_t^0, \mathbb{E}\{m(X,s)\phi_W(W)\}\rangle_{\mathcal{H}_W}\big| \\
&\leq \sqrt{n}\cdot\|(\lambda I + A^*A)^{-1}A^*b_t - H_t^0\|_{\mathcal{H}_W}\cdot\|g_s\|_{\mathcal{H}_W} \\
&\overset{(1)}{=} O_p(\sqrt{n\lambda^2}) \overset{(2)}{=} o_p(1),
\end{aligned}
\tag{50}
$$

where (1) follows from $\|g_s\|_{\mathcal{H}_W} < \infty$ by Eq. (46) in Lemma D.7 and (2) follows from assumption 4.10. $\qquad\square$

**Lemma D.9.** *Under assumptions 4.10 C.2 and C.3, $S_{n1}(s,t) = \frac{1}{\sqrt{n}}\sum_{i=1}^{n} U(w_i, y_i, t)\{A(A^*A)^{-1}g_s\}(x_i) + o_p(1)$ as $n \to \infty$, where $g_s(\cdot) := \mathbb{E}\{m(X,s)k_W(W,\cdot)\}$.*

*Proof.* By the reproducing property, we have $(\lambda I + A^*A)^{-1}A^*(\widehat{b}_t - \widehat{A}H_t^0)(W) = \langle(\lambda I + A^*A)^{-1}A^*(\widehat{b}_t - \widehat{A}H_t^0), \phi_W(W)\rangle_{\mathcal{H}_W}$. Therefore, we have the following for $S_{n1}(s,t)$:

$$
\begin{aligned}
\mathbb{P}\{G_1 m(X,s)\} &= \mathbb{E}\left[(\lambda I + A^*A)^{-1}A^*(\widehat{b}_t - \widehat{A}H_t^0)(W)m(X,s)\right] \\
&= \mathbb{E}\left[\langle(\lambda I + A^*A)^{-1}A^*(\widehat{b}_t - \widehat{A}H_t^0), \phi_W(W)\rangle_{\mathcal{H}_W}\cdot m(X,s)\right] \\
&\overset{(1)}{=} \left\langle(\lambda I + A^*A)^{-1}A^*(\widehat{b}_t - \widehat{A}H_t^0), \mathbb{E}\{m(X,s)\phi_W(W)\}\right\rangle_{\mathcal{H}_W} \\
&\overset{(2)}{=} \langle A^*(\widehat{b}_t - \widehat{A}H_t^0), (\lambda I + A^*A)^{-1}g_s\rangle_{\mathcal{H}_W} \\
&= \langle A^*(\widehat{b}_t - \widehat{A}H_t^0), \{(\lambda I + A^*A)^{-1} - (A^*A)^{-1}\}g_s\rangle_{\mathcal{H}_W} + \langle A^*(\widehat{b}_t - \widehat{A}H_t^0), (A^*A)^{-1}g_s\rangle_{\mathcal{H}_W},
\end{aligned}
$$

where (1) follows from (20) and (2) follows from $\{(\lambda I + A^*A)^{-1}\}^* = (\lambda I + A^*A)^{-1}$.

We first analyze the second term in RHS. By (26), we obtain:

$$
\begin{aligned}
(\widehat{b}_t - \widehat{A}H_t^0)(X) &= \left\{\frac{1}{n}\sum_{i=1}^{n}\varphi(y_i,t)\phi_X(x_i) - \frac{1}{n}\sum_{i=1}^{n}H^0(w_i,t)\phi_X(x_i)\right\}(X) \\
&= \frac{1}{n}\sum_{i=1}^{n}U(w_i,y_i,t)k_X(x_i,X),
\end{aligned}
$$

where $U$ is defined in (34). Since $A^*m_t := \int m(X,t)\phi_W(W)d\mathbb{P}(X,W)$ in (27), we have:

$$
\begin{aligned}
\left\{A^*(\widehat{b}_t - \widehat{A}H_t^0)\right\}(\cdot) &= \frac{1}{n}\sum_{i=1}^{n}U(w_i,y_i,t)\int k_X(x_i,X)\phi_W(W)d\mathbb{P}(X,W) \\
&= \frac{1}{n}\sum_{i=1}^{n}U(w_i,y_i,t)A^*\{k_X(x_i,X)\}(\cdot),
\end{aligned}
$$

Therefore, we obtain:

$$\sqrt{n} \left\langle A^*(\widehat{b}_t - \widehat{A}H_t^0), (A^*A)^{-1}g_s \right\rangle_{\mathcal{H}_W} = \sqrt{n} \left\langle (A^*A)^{-1}A^*(\widehat{b}_t - \widehat{A}H_t^0), \mathbb{E}\{m(X,s)\phi_W(W)\} \right\rangle_{\mathcal{H}_W}$$

$$\overset{(1)}{=} \sqrt{n}\mathbb{E}\left\{(A^*A)^{-1}A^*(\widehat{b}_t - \widehat{A}H_t^0)(W)m(X,s)\right\}$$

$$= \sqrt{n}\mathbb{E}\left[(A^*A)^{-1}\left\{\frac{1}{n}\sum_{i=1}^n U(w_i, y_i, t)A^*\{k_X(x_i, X)\}(W)\right\}m(X,s)\right]$$

$$= \frac{1}{\sqrt{n}}\sum_{i=1}^n U(w_i, y_i, t)\int (A^*A)^{-1}A^*\{k_X(x_i, X)\}(W)m(X,s)d\mathbb{P}(X, W)$$

$$\overset{(2)}{=} \frac{1}{\sqrt{n}}\sum_{i=1}^n U(w_i, y_i, t)\left\langle (A^*A)^{-1}A^*\{k_X(x_i, X)\}, \mathbb{E}\{m(X,s)\phi_W(W)\} \right\rangle_{\mathcal{H}_W}$$

$$= \frac{1}{\sqrt{n}}\sum_{i=1}^n U(w_i, y_i, t)\left\langle k_X(x_i, \cdot), A(A^*A)^{-1}g_s \right\rangle_{\mathcal{H}_X}$$

$$\overset{(3)}{=} \frac{1}{\sqrt{n}}\sum_{i=1}^n U(w_i, y_i, t)\{A(A^*A)^{-1}g_s\}(x_i),$$

where $(1), (2), (3)$ follows from reproducing property $f(x) = \langle f, k_X(x, \cdot)\rangle_{\mathcal{H}_X}$ and $g(w) = \langle g, k_W(w, \cdot)\rangle_{\mathcal{H}_W}$ for each $f \in \mathcal{H}_X$ and $g \in \mathcal{H}_W$. Besides, for $\sqrt{n}\|A^*(\widehat{b}_t - \widehat{A}H_t^0)\|_{\mathcal{H}_W}$, we have:

$$\|A^*(\widehat{b}_t - \widehat{A}H_t^0)\|_{\mathcal{H}_W} = \|A^*\widehat{b}_t - A^*b_t + A^*b_t - A^*\widehat{A}H_t^0\|_{\mathcal{H}_W}$$

$$\leq \|A^*\widehat{b}_t - A^*b_t\|_{\mathcal{H}_W} + \|A^*AH_t^0 - A^*\widehat{A}H_t^0\|_{\mathcal{H}_W}$$

$$\leq \|A^*\|_{\mathrm{op}} \cdot \|\widehat{b}_t - b_t\|_{\mathcal{H}_X} + \|A^*\|_{\mathrm{op}} \cdot \|A - \widehat{A}\|_{\mathrm{op}} \cdot \|H_t^0\|_{\mathcal{H}_W}.$$

Since $H_t^0 \in \mathcal{H}_W$, we must have $\|H_t^0\|_{\mathcal{H}_W} < \infty$. Besides, according to Sec. C.2, we have $\|A^*\|_{\mathrm{op}} = \|A\|_{\mathrm{op}} < \infty$ since $A$ is a bounded linear operator. Therefore, the last term is $O_p\left(\frac{1}{\sqrt{n}}\right)$. By Lemma D.12, we have $\sqrt{n}\|A^*(\widehat{b}_t - \widehat{A}H_t^0)\|_{\mathcal{H}_W} = O_p(1)$. By the Cauchy-Schwartz inequality, we have:

$$\left|\left\langle \sqrt{n}A^*(\widehat{b}_t - \widehat{A}H_t^0), \{(\lambda I + A^*A)^{-1} - (A^*A)^{-1}\}g_s \right\rangle_{\mathcal{H}_W}\right|$$

$$\leq \sqrt{n}\|A^*(\widehat{b}_t - \widehat{A}H_t^0)\|_{\mathcal{H}_W} \cdot \left\|\{(\lambda I + A^*A)^{-1} - (A^*A)^{-1}\}g_s\right\|_{\mathcal{H}_W}$$

$$\leq O_p\left\{\left\|\{(\lambda I + A^*A)^{-1} - (A^*A)^{-1}\}g_s\right\|_{\mathcal{H}_W}\right\}.$$

By $A^*A\widetilde{g}_s = g_s$ in Lemma D.7, we have:

$$\{(\lambda I + A^*A)^{-1} - (A^*A)^{-1}\}g_s = (\lambda I + A^*A)^{-1}A^*(A\widetilde{g}_s) - \widetilde{g}_s. \tag{51}$$

Since $(\lambda I + A^*A)^{-1}A^*$ is a Tikhonov regularization scheme, the RHS of the above display converges to 0 as $\lambda \to 0$. In fact, according to Lemma D.11, we have $\|(\lambda I + A^*A)^{-1}A^*Ag - g\|_{\mathcal{H}_W}^2 \leq \lambda^{\min\{\theta, 2\}}$ since $g$ satisfies assumption D.2 (b). By (48), $\widetilde{g}_s = \sum_j \lambda_j^{-2}\langle g_s, \varphi_j\rangle_{\mathcal{H}_W}\varphi_j$. Thus, similar to the proof of Lemma D.11, we have

$$\|\{(\lambda I + A^*A)^{-1} - (A^*A)^{-1}\}g_s\|_{\mathcal{H}_W}^2 = \sum_j \left\{\left(\frac{1}{\lambda_j^2 + \lambda} - \frac{1}{\lambda_j^2}\right)\langle g_s, \varphi_j\rangle_{\mathcal{H}_W}\right\}^2$$

$$\leq \sup_j \left\{\frac{\lambda\lambda_j^\eta}{\lambda_j^2(\lambda_j^2 + \lambda)}\right\}^2 \sum_j \frac{|\langle g_s, \varphi_j\rangle_{\mathcal{H}_W}|^2}{\lambda_j^{2\eta}}.$$

If $\eta \geq 4$, since the maximum singular value of the operator equals $\|A\|_{\mathrm{op}} < \infty$, we have

$$\sup_j \left\{\frac{\lambda\lambda_j^\eta}{\lambda_j^2(\lambda_j^2 + \lambda)}\right\}^2 \leq \lambda^2\sup_j\left\{\frac{\lambda_j^\eta}{\lambda_j^2(\lambda_j^2 + \lambda)}\right\}^2 \leq \lambda^2\sup_j \lambda_j^{2\eta-8} = O(\lambda^2).$$

If $2 < \eta < 4$, we define $x := \lambda_j^2$. Plugging it into $\left\{ \frac{\lambda \lambda_j^\eta}{\lambda_j^2 (\lambda_j^2 + \lambda)} \right\}^2$, we define $f(x) = \frac{\lambda^2 x^{\eta-2}}{(x+\lambda)^2}$. Noted that $f(x)$ is maximized (by using the first order condition) at $x = \lambda(2-\eta)(\eta-4)^{-1}$. Thus, the maximum value of $f(x)$ is

$$\frac{x^{\eta-2} \lambda^2}{(x+\lambda)^2} \le \frac{(\eta-2)^{\eta-2}(4-\eta)^{4-\eta}}{4} \cdot \lambda^{\eta-2} = O(\lambda^{\eta-2}).$$

Thus, $\{(\lambda I + A^* A)^{-1} - (A^* A)^{-1}\} g_s \to 0$ by assumption 4.10. Therefore, we have

$$\sqrt{n} \mathbb{P}\{G_1 m(X, s)\} = \frac{1}{\sqrt{n}} \sum_{i=1}^{n} U(w_i, y_i, t)\{A(A^* A)^{-1} g_s\}(x_i) + o_p(1).$$

$\square$

## D.4. Proofs in section 4.3

**Theorem 4.12.** *Let $\eta_{s,t}(O) := U(W, Y, t) m(X, s) - U(W, Y, t)\{A(A^* A)^{-1} g_s\}(X)$, with $O := (W, Y, X)$. Suppose assumptions 4.9–4.11, C.2–C.4, and D.1–D.2 hold. Under $\mathbb{H}_0$, we have **(i).** $T_n(s, t)$ converges weakly to $\mathbb{G}(s, t)$ such that $\iint |\mathbb{G}(s, t)|^2 d\mu(s) d\mu(t) < \infty$, where $\mathbb{G}(s, t)$ is a Gaussian process with zero-mean and covariance:*

$$\Sigma\{(s, t), (s', t')\} = \mathbb{E}\{\eta_{s,t}(O)\eta_{s',t'}(O'))\},$$

*where $O' := (W', Y', X')$ is an independent copy of $O$; and **(ii).** $\Delta_{\varphi,m}$ converges weakly to $\max_{t \in \mathcal{T}} \int |\mathbb{G}(s, t)|^2 d\mu(s)$.*

*Proof.* By (36), we have

$$T_n(s, t) = \sqrt{n} \mathbb{P}_n \{U(W, Y, t) m(X, s)\} + (\text{Expected risk difference}) + (\text{Empirical process}).$$

By Propositions D.3 and D.4, we have:

$$T_n(s, t) = \frac{1}{\sqrt{n}} \sum_{i=1}^{n} U(w_i, y_i, t) \left[ m(x_i, s) - \left\{ A(A^* A)^{-1} g_s \right\}(x_i) \right] + o_p(1).$$

Next, we apply Lemma D.15 to $\left\{ U(w_i, y_i, t) \left[ \left\{ m(x_i, s) - A(A^* A)^{-1} g_s \right\}(x_i) \right] \right\}_i$ to obtain the result. To this end, we need to verify $U(W, Y, t)[\{A(A^* A)^{-1} g_s\}(X) + m(X, s)]$ is zero mean and

$$\mathbb{E}\left[ \left\| U(w_i, y_i, t)[m(x_i, s - \{A(A^* A)^{-1} g_s\}(x_i))] \right\|_{\mathcal{L}^2\{\mathcal{T} \times \mathcal{T}, \mu \times \mu\}} \right] < \infty. \tag{52}$$

Notice that the zero-mean is met by $\mathbb{E}\{U(W, Y, t)|X\} = \mathbb{E}\{\varphi(Y, t) - H^0(W, t)|X\} = 0$ under $\mathbb{H}_0$. Besides, by assumption 4.11, we have $\mathrm{Var}(U(w_i, y_i, t)[m(x_i, s) - \{A(A^* A)^{-1} g_s\}(x_i)]) = \mathbb{E}(U(w_i, y_i, t)[m(x_i, s) - \{A(A^* A)^{-1} g_s\}(x_i)])^2 < \mathbb{E}\{U(w_i, y_i, t)^4|X_i\} + \mathbb{E}[m(x_i, s) - \{A(A^* A)^{-1} g_s\}(x_i)]^4 < \infty$ for any $(s, t)$. Therefore, exchanging the order of integration, taking the maximum value of $t$, and setting the measure to be a probability measure, we get (52). Thus, we have $T_n(s, t)$ converges weakly to $\mathbb{G}(s, t)$ in $\mathcal{L}^2\{\mathcal{T} \times \mathcal{T}, \mu \times \mu\}$, where $\mathbb{G}(s, t)$ is a Gaussian process with zero-mean.

For any fixed $t$ and $T_n(s, t) \in \mathcal{L}^2\{\mathcal{T}, \mu\}$, we use the continuous mapping theorem (Theorem 1.3.6 of (Wellner et al., 2013)) to obtain

$$\int |T_n(s, t)|^2 d\mu(s) \xrightarrow{d} \int |\mathbb{G}(s, t)|^2 d\mu(s),$$

by the continuity of the integral functional. Next, for $\int |T_n(s, t)|^2 d\mu(s) \in \mathcal{L}^2\{\mathcal{T}, \mu\}$, noted that $t$ in $\int |T_n(s, t)|^2 d\mu(s)$ is determined by $U(w, y, t) = \varphi(y, t) - H^0(w, t)$. To ensure that taking the max operation is meaningful, we need to prove that if $U(w, y, t) \in \mathcal{L}^2\{F(w, y)\}$ for any $t$, $\max_{t \in T} |U(w, y, t)| \in \mathcal{L}^2\{F(w, y)\}$. By (17), we have:

$$\int |\varphi(y, t) - H(w, t)|^2 p(w, y) dw dy \le 2 \int |\varphi(y, t)|^2 p(y) dy + 2 \int |H(w, t)|^2 p(w) dw$$

$$\le 2 + \left\{ \iint \left| \frac{g(w, u)}{p(u)} \right|^2 p(w)p(u) dw du \right\} \left\{ \int \left| \int e^{ity} p(y|u) dy \right|^2 p(u) du \right\} < \infty,$$

where the second inequality follows from $(a-b)^2 \le 2a^2 + 2b^2$ and $|e^{ity}|^2 = 1$. Thus, taking max operation on both sides, we have $\max_{t \in T} \int |U(w, y, t)|^2 p(w, y) dw dy < \infty$.

Next, we prove the continuity of the max functional in metric $d$. Next, we prove the continuity of the max functional. If $d(f_1, f_2) < \delta$ given any $\delta > 0$, we have $\max_{t \in T}|f_1(t)| - \max_{t \in T}|f_2(t)| \le \max_{t \in T}|f_1(t) - f_2(t)| = d(f_1, f_2) < \delta$. Applying the continuous mapping theorem to such continuous metric $\max$, we have:

$$\max_{t \in T} \Delta(t) \xrightarrow{d} \max_{t \in T} \int |\mathbb{G}(s, t)|^2 d\mu(s).$$

The proof is complete. $\qquad\qquad\qquad\qquad\qquad\qquad\qquad\qquad\qquad\qquad\qquad\qquad\qquad\qquad\qquad\qquad\qquad\quad$ $\square$

**Theorem 4.14.** *Suppose assumptions in Theorem 4.12 hold. Besides, we assume $\mathbb{E}\{r(X, t)^4\} < \infty$. Then, we have:*

(i) **Global alternative**. $\lim_{n \to \infty} \max_{t \in \mathcal{T}} |T_n(s, t)| = \infty$ *for almost all $s$ under $\mathbb{H}_1^{\text{fix}}$.*
(ii) **Local alternative** *($\alpha < 1/2$)*. $\lim_{n \to \infty} \max_{t \in \mathcal{T}} |T_n(s, t)| = \infty$ *for almost all $s$ under $\mathbb{H}_{1n}^{\alpha}$.*
(iii) **Local alternative** *($\alpha = 1/2$)*. $T_n(s, t)$ *converges weakly to $\mathbb{G}(s, t) + \mu(X, s, t)$ such that $\iint |\mathbb{G}(s, t) + \mu(X, s, t)|^2 d\mu(s) d\mu(t) < \infty$ under $\mathbb{H}_{1n}^{\alpha}$, where $\mathbb{G}(s, t)$ is defined in Theorem 4.12 and $\mu(X, s, t) := \mathbb{E}\left[\{r(X, t) - (A^*A)^{-1}A^*r(X, t)\}m(X, s)\right]$.*

*Proof.* **(i). The case of $\mathbb{H}_1^{\text{fix}}$.**

We first decompose $T_n(s, t)$ as follows

$$
\begin{aligned}
T_n(s, t) &= \frac{1}{\sqrt{n}} \sum_{i=1}^{n} \widehat{U}(w_i, y_i; t) m(x_i, s) \\
&= \sqrt{n} \mathbb{P}_n \left[ \{\varphi(Y, t) - \widehat{H}^{\lambda}(W, t)\} m(X, s) \right] \\
&= \sqrt{n} \mathbb{P}_n \left[ \{\varphi(Y, t) - \widehat{H}^0(W, t) + H^0(W, t) - \widehat{H}^{\lambda}(W, t)\} m(X, s) \right] \\
&= \sqrt{n} \mathbb{P}_n \left[ \{\varphi(Y, t) - \widehat{H}^0(W, t)\} m(X, s) \right] - \sqrt{n} \mathbb{P} \left[ \{\widehat{H}^{\lambda}(W, t) - \widehat{H}^0(W, t)\} m(X, s) \right] \\
&\quad - \sqrt{n}(\mathbb{P}_n - \mathbb{P}) \left[ \{\widehat{H}^{\lambda}(W, t) - \widehat{H}^0(W, t)\} m(X, s) \right].
\end{aligned}
$$

where $\widehat{H}^0(W, t)$ is the least squares solution.

We first analyze $\mathbb{P}\left[\{\widehat{H}^{\lambda}(W, t) - \widehat{H}^0(W, t)\}m(X, s)\right]$.

For $\widehat{H}^{\lambda}(w, t) - \widehat{H}^0(w, t)$, we can decompose $b_t = b_t^{\overline{\text{ran}}} + b_t^{\text{ker}}$, where $b_t^{\text{ker}} \in \text{Ker}(A^*)$ and $b_t^{\overline{\text{ran}}} \in \text{Ker}(A^*)^{\perp} = \overline{\text{Ran}(A)}$. By using a $b_t^{\varepsilon} \in \text{Ran}(A)$ to approximate $b_t^{\overline{\text{ran}}}$, we get $b_t = b_t^{\overline{\text{ran}}} - b_t^{\varepsilon} + b_t^{\varepsilon} + b_t^{\text{ker}}$. By this decomposition, we have the following for $\widehat{H}^{\lambda}(w, t) - \widehat{H}^0(w, t)$:

$$
\begin{aligned}
\widehat{H}^{\lambda}(w, t) - \widehat{H}^0(w, t) &= (\lambda I + \widehat{A}^*\widehat{A})^{-1}\widehat{A}^*\widehat{b}_t - \widehat{H}^0(w, t) \\
&= (\lambda I + \widehat{A}^*\widehat{A})^{-1}\widehat{A}^*(\widehat{b}_t - b_t) + (\lambda I + \widehat{A}^*\widehat{A})^{-1}\widehat{A}^*b_t - (A^*A)^{-1}A^*b_t \\
&= (\lambda I + \widehat{A}^*\widehat{A})^{-1}\widehat{A}^*(\widehat{b}_t - b_t) + \{(\lambda I + \widehat{A}^*\widehat{A})^{-1}\widehat{A}^* - (A^*A)^{-1}A^*\}(b_t^{\overline{\text{ran}}} + b_t^{\text{ker}}) \\
&= (\lambda I + \widehat{A}^*\widehat{A})^{-1}\widehat{A}^*(\widehat{b}_t - b_t) + \{(\lambda I + \widehat{A}^*\widehat{A})^{-1}\widehat{A}^* - (A^*A)^{-1}A^*\}(b_t^{\overline{\text{ran}}} - b_t^{\varepsilon} + b_t^{\varepsilon} + b_t^{\text{ker}}).
\end{aligned}
$$

Thus, we have

$$
\begin{aligned}
\mathbb{P}\left[\{\widehat{H}^{\lambda}(W, t) - \widehat{H}^0(W, t)\}m(X, s)\right] &= \mathbb{P}\left(\left[\{(\lambda I + \widehat{A}^*\widehat{A})^{-1}\widehat{A}^* - (A^*A)^{-1}A^*\}(b_t^{\overline{\text{ran}}} - b_t^{\varepsilon})\right]m(X, s)\right) \\
&\quad + \mathbb{P}\left(\left[\{(\lambda I + \widehat{A}^*\widehat{A})^{-1}\widehat{A}^* - (A^*A)^{-1}A^*\}b_t^{\text{ker}}\right]m(X, s)\right) \\
&\quad + \mathbb{P}\left(\left[\{(\lambda I + \widehat{A}^*\widehat{A})^{-1}\widehat{A}^* - (A^*A)^{-1}A^*\}b_t^{\varepsilon}\right]m(X, s)\right) \\
&\quad + \mathbb{P}\left[\{(\lambda I + \widehat{A}^*\widehat{A})^{-1}\widehat{A}^*(\widehat{b}_t - b_t)\}m(X, s)\right].
\end{aligned}
$$

For the first term, since we can always make $b_t^\varepsilon$ arbitrarily close to $b_t^{\overline{\mathrm{ran}}}$, which causes $b_t^{\overline{\mathrm{ran}}} - b_t^\varepsilon < \varepsilon$. Thus, we only need to consider the last three terms.

For the second term, since $b_t^{\mathrm{ker}} \in \mathrm{Ker}(A^*)$, we have

$$\{(\lambda I + \widehat{A}^*\widehat{A})^{-1}\widehat{A}^* - (A^*A)^{-1}A^*\}b_t^{\mathrm{ker}} = (\lambda I + \widehat{A}^*\widehat{A})^{-1}\widehat{A}^*b_t^{\mathrm{ker}}.$$

Then, we have

$$\begin{aligned}
\mathbb{P}\left\{(\lambda I + \widehat{A}^*\widehat{A})^{-1}\widehat{A}^*b_t^{\mathrm{ker}} \cdot m(X,s)\right\} &\overset{(1)}{=} \langle(\lambda I + \widehat{A}^*\widehat{A})^{-1}\widehat{A}^*b_t^{\mathrm{ker}}, \mathbb{E}\left[m(X,s)\phi_W(W)\right]\rangle_{\mathcal{H}_W} \\
&= \langle(\lambda I + \widehat{A}^*\widehat{A})^{-1}(\widehat{A}^* - A^*)b_t^{\mathrm{ker}}, g_s\rangle_{\mathcal{H}_W} + \langle(\lambda I + \widehat{A}^*\widehat{A})^{-1}A^*b_t^{\mathrm{ker}}, g_s\rangle_{\mathcal{H}_W} \\
&\overset{(2)}{=} \langle(\widehat{A}^* - A^*)b_t^{\mathrm{ker}}, (\lambda I + \widehat{A}^*\widehat{A})^{-1}g_s\rangle_{\mathcal{H}_W} \\
&= -\underbrace{\langle(\widehat{A}^* - A^*)b_t^{\mathrm{ker}}, \{(\lambda I + A^*A)^{-1} - (\lambda I + \widehat{A}^*\widehat{A})^{-1}\}g_s\rangle_{\mathcal{H}_W}}_{\text{(I)}} \\
&\quad + \underbrace{\langle(\widehat{A}^* - A^*)b_t^{\mathrm{ker}}, (\lambda I + A^*A)^{-1}g_s\rangle_{\mathcal{H}_W}}_{\text{(II)}},
\end{aligned}$$

where (1) follows from (20) and (2) follow from $b_t^{\mathrm{ker}} \in \mathrm{Ker}(A^*)$.

According to Lemma D.16, for the first term, we have

$$\text{(I)} = \langle(\widehat{A}^* - A^*)b_t^{\mathrm{ker}}, \{(\lambda I + \widehat{A}^*\widehat{A})^{-1}(\widehat{A}^*\widehat{A} - A^*A)\}(\lambda I + A^*A)^{-1}g_s\rangle_{\mathcal{H}_W}.$$

By Lemma D.10 (d) and assumption D.2 (b), we have $\left\|(\lambda I + A^*A)^{-1}g_s\right\|_{\mathcal{H}_W} = O_p\{\lambda^{\frac{\min(\eta,2)}{2}-1}\}$. Since $\eta > 2$ in assumption D.2 (a), we have $\left\|(\lambda I + A^*A)^{-1}g_s\right\|_{\mathcal{H}_W} = O_p(1)$. By Lemma D.12 and D.13, we have $\|\widehat{A}^* - A^*\|_{\mathrm{op}} = O_p(1/\sqrt{n})$ and $\|\widehat{A}^*\widehat{A} - A^*A\|_{\mathrm{op}} = O_p(1/\sqrt{n})$. By Lemma D.10 (b), we have $\|(\lambda I + \widehat{A}^*\widehat{A})\|_{\mathrm{op}} = O_p(1/\lambda)$. Combining all these results, we get

$$\begin{aligned}
\text{(I)} &\leq \|\widehat{A}^* - A^*\|_{\mathrm{op}} \cdot \|b_t^{\mathrm{ker}}\| \cdot \|(\lambda I + \widehat{A}^*\widehat{A})^{-1}\|_{\mathrm{op}} \cdot \|\widehat{A}^*\widehat{A} - A^*A\|_{\mathrm{op}} \cdot \left\|(\lambda I + A^*A)^{-1}g_s\right\|_{\mathcal{H}_W} \\
&= O_p(1/\sqrt{n}) * O_p(1/\lambda) * O_p(1/\sqrt{n}) * O_p(1) = O_p\left(\frac{1}{n\lambda}\right).
\end{aligned}$$

By assumption 4.10, we have (I) $= o_p(1)$. For (II), by Lemma D.10 (d) and assumption D.2 (b), we have $\left\|(\lambda I + A^*A)^{-1}g_s\right\|_{\mathcal{H}_W} = O_p\left\{\lambda^{\frac{\min(\eta,2)}{2}-1}\right\}$. By Lemma D.12 and, we have $\|\widehat{A}^* - A^*\|_{\mathrm{op}} = O_p(1/\sqrt{n})$.

$$\begin{aligned}
\text{(II)} &\leq \|\widehat{A}^* - \widehat{A}\|_{\mathrm{op}} \cdot \|b_t^{\mathrm{ker}}\|_{\mathcal{H}_W} \cdot \left\|(\lambda I + A^*A)^{-1}g_s\right\|_{\mathcal{H}_W} \\
&= O_p(1/\sqrt{n}) * O_p\{\lambda^{\frac{\min(\eta,2)}{2}-1}\} = o_p(1).
\end{aligned}$$

Next, we consider $\{(\lambda I + \widehat{A}^*\widehat{A})^{-1}\widehat{A}^* - (A^*A)^{-1}A^*\}b_t^\varepsilon$. In fact, we have the following decomposition

$$\{(\lambda I + \widehat{A}^*\widehat{A})^{-1}\widehat{A}^* - (A^*A)^{-1}A^*\}b_t^\varepsilon = S_1 + S_2 + S_3,$$

where

$$\begin{aligned}
S_1 &:= \{(\lambda I + \widehat{A}^*\widehat{A})^{-1}\widehat{A}^* - (\lambda I + A^*A)^{-1}\widehat{A}^*\}b_t^\varepsilon; \\
S_2 &:= (\lambda I + A^*A)^{-1}(\widehat{A}^* - A^*)b_t^\varepsilon; \\
S_3 &:= \left\{(\lambda I + A^*A)^{-1} - (A^*A)^{-1}\right\}A^*b_t^\varepsilon.
\end{aligned}$$

For $S_2$. we have

$$\begin{aligned}
\mathbb{P}\{S_2 m(X,s)\} &= \langle(\lambda I + A^*A)^{-1}(\widehat{A}^* - A^*)b_t^\varepsilon, \mathbb{E}\{m(X,s)\phi_W(W)\}\rangle_{\mathcal{H}_W} \\
&= \langle(\widehat{A}^* - A^*)b_t^\varepsilon, (\lambda I + A^*A)^{-1}g_s\rangle_{\mathcal{H}_W} \\
&\leq \|\widehat{A}^* - A^*\|_{\mathrm{op}} \cdot \|b_t^\varepsilon\|_{\mathcal{H}_X} \cdot \|(\lambda I + A^*A)^{-1}g_s\|_{\mathcal{H}_W}.
\end{aligned}$$

By Lemma D.10 (d) and assumption D.2 (b), we have $\left\|(\lambda I + A^*A)^{-1}g_s\right\|_{\mathcal{H}_W} = O_p\{\lambda^{\frac{\min(\eta,2)}{2}-1}\}$. By Lemma D.12, we have $\|\widehat{A}^* - A^*\|_{\mathrm{op}} = O_p(1/\sqrt{n})$. Combining all these results and according to assumption D.2 (a) with $\eta > 2$, we get $\mathbb{P}\{S_2 m(X,s)\} = O_p(1/\sqrt{n}) = o_p(1)$.

For $S_3$, since $A^*A\widetilde{g}_s = g_s$, we have

$$\begin{aligned}
\mathbb{P}\{S_3 m(X,s)\} &= \langle\{(\lambda I + A^*A)^{-1} - (A^*A)^{-1}\}A^*b_t^\varepsilon, \mathbb{E}\{m(X,s)\phi_W(W)\}\rangle_{\mathcal{H}_W} \\
&= \langle A^*b_t^\varepsilon, \{(\lambda I + A^*A)^{-1} - (A^*A)^{-1}\}g_s\rangle_{\mathcal{H}_W} \\
&= \langle A^*b_t^\varepsilon, \{(\lambda I + A^*A)^{-1} - (A^*A)^{-1}\}A^*A\widetilde{g}_s\rangle_{\mathcal{H}_W} \\
&= \langle A^*b_t^\varepsilon, (\lambda I + A^*A)^{-1}A^*A\widetilde{g}_s - \widetilde{g}_s\rangle_{\mathcal{H}_W}.
\end{aligned}$$

By Lemma D.9, we have $\|(\lambda I + A^*A)^{-1}A^*A\widetilde{g}_s - \widetilde{g}_s\|_{\mathcal{H}_W} = o_p(1)$. Besides, since $\|A^*\|_{\mathrm{op}} = \|A\|_{\mathrm{op}} < \infty$ and $\|b_t^\varepsilon\|_{\mathcal{H}_X} < \infty$, we have

$$\mathbb{P}\{S_3 m(X,s)\} \le \|A\|_{\mathrm{op}} \cdot \|b_t^\varepsilon\|_{\mathcal{H}_X} \cdot \|(\lambda I + A^*A)^{-1}A^*A\widetilde{g}_s - \widetilde{g}_s\|_{\mathcal{H}_W} = o_p(1).$$

For $S_1$, we have

$$\begin{aligned}
\mathbb{P}\{S_1 m(X,s)\} &= \langle\{(\lambda I + \widehat{A}^*\widehat{A})^{-1}\widehat{A}^* - (\lambda I + A^*A)^{-1}\widehat{A}^*\}b_t^\varepsilon, \mathbb{E}\{m(X,s)\phi_W(W)\}\rangle_{\mathcal{H}_W} \\
&= \langle\{(\lambda I + \widehat{A}^*\widehat{A})^{-1} - (\lambda I + A^*A)^{-1}\}\widehat{A}^*b_t^\varepsilon, g_s\rangle_{\mathcal{H}_W} \\
&= \langle(\lambda I + A^*A)^{-1}\{A^*A - \widehat{A}^*\widehat{A}\}(\lambda I + \widehat{A}^*\widehat{A})^{-1}\widehat{A}^*b_t^\varepsilon, g_s\rangle_{\mathcal{H}_W} \\
&= \langle\{A^*A - \widehat{A}^*\widehat{A}\}(\lambda I + \widehat{A}^*\widehat{A})^{-1}\widehat{A}^*b_t^\varepsilon, (\lambda I + A^*A)^{-1}g_s\rangle_{\mathcal{H}_W} \\
&\le \|A^*A - \widehat{A}^*\widehat{A}\|_{\mathrm{op}} \cdot \|(\lambda I + \widehat{A}^*\widehat{A})^{-1}\widehat{A}^*\|_{\mathrm{op}} \cdot \|b_t^\varepsilon\|_{\mathcal{H}_X} \cdot \|(\lambda I + A^*A)^{-1}g_s\|_{\mathcal{H}_W}.
\end{aligned}$$

By Lemma D.10 (d), we have $\|(\lambda I + A^*A)^{-1}g_s\|_{\mathcal{H}_W} = O_p\{\lambda^{\frac{\min(\eta,2)}{2}-1}\}$. By Lemma D.10 (c), we have $\|\widehat{A}(\lambda I + \widehat{A}^*\widehat{A})^{-1}\|_{\mathrm{op}} = \|(\lambda I + \widehat{A}^*\widehat{A})^{-1}\widehat{A}^*\|_{\mathrm{op}} \lesssim 1/\sqrt{\lambda}$. By Lemma D.13, we have $\|A^*A - \widehat{A}^*\widehat{A}\|_{\mathrm{op}} = O_p(1/\sqrt{n})$. Combining these results and according to assumption D.2 (a) with $\eta > 2$, we get $\mathbb{P}\{S_1 m(X,s)\} = O_p(1/\sqrt{n\lambda}) = o_p(1)$.

Thus, we have

$$\mathbb{P}\left(\left[\{(\lambda I + \widehat{A}^*\widehat{A})^{-1}\widehat{A}^* - (A^*A)^{-1}A^*\}b_t^\varepsilon\right]m(X,s)\right) = o_p(1).$$

For the last term, we have

$$\begin{aligned}
\mathbb{P}\left[\{(\lambda I + \widehat{A}^*\widehat{A})^{-1}\widehat{A}^*(\widehat{b}_t - b_t)\}m(X,s)\right] &= \langle(\lambda I + \widehat{A}^*\widehat{A})^{-1}\widehat{A}^*(\widehat{b}_t - b_t), \mathbb{E}\{m(X,s)\phi_W(W)\}\rangle_{\mathcal{H}_W} \\
&\le \left\|(\lambda I + \widehat{A}^*\widehat{A})^{-1}\widehat{A}^*\right\|_{\mathrm{op}} \cdot \|\widehat{b}_t - b_t\|_{\mathcal{H}_X} \cdot \|g_s\|_{\mathcal{H}_W}.
\end{aligned}$$

By Lemma D.10 (c), we have $\|(\lambda I + \widehat{A}^*\widehat{A})^{-1}\widehat{A}^*\|_{\mathrm{op}} \lesssim 1/\sqrt{\lambda}$. By Lemma D.12, we have $\|\widehat{b}_t - b_t\|_{\mathrm{op}} = O_p(1/\sqrt{n})$. Combining these results, we get $\mathbb{P}\left[\{(\lambda I + \widehat{A}^*\widehat{A})^{-1}\widehat{A}^*(\widehat{b}_t - b_t)\}m(X,s)\right] = O_p(1/\sqrt{n\lambda}) = o_p(1)$.

Thus, we have

$$\mathbb{P}\left[\{\widehat{H}^\lambda(W,t) - \widehat{H}^0(W,t)\}m(X,s)\right] = o_p(1).$$

By the above proof, we similarly have $\mathbb{P}\left|\{\widehat{H}^\lambda(W,t) - \widehat{H}^0(W,t)\}m(X,s)\right| = o_p(1)$, and therefore $(\mathbb{P}_n - \mathbb{P})\left[\{\widehat{H}^\lambda(W,t) - \widehat{H}^0(W,t)\}m(X,s)\right] = o_p(1)$.

Besides, for the first term of $T_n(s,t)$, there exists $t$, we have

$$\mathbb{P}_n\left[\{\varphi(Y,t) - \widehat{H}^0(W,t)\}m(X,s)\right] = \mathbb{P}_n\left[\{\varphi(Y,t) - H^0(W,t) + H^0(W,t) - \widehat{H}^0(W,t)\}m(X,s)\right].$$

According to the definition of $\mathbb{H}_1^{\mathrm{fix}}$, there exists $r(X,t)$ such that $\mathbb{E}\{\varphi(Y,t) - H^0(W,t)|X\} = r(X,t)$, where $r(X,t)$ cannot be written as $\mathbb{E}\{H(W,t) - H^0(W,t)|X\}$ for any $t$.

We first prove $\mathbb{E}[|\{r(X,t) + H^0(W,t) - \widehat{H}^0(W,t)\}m(X,s)|] < \infty$. In fact, we only proof $\mathbb{E}[|r(X,t)m(X,s)|] < \infty$ and $\mathbb{E}[|\{H^0(W,t) - \widehat{H}^0(W,t)\}m(X,s)|] < \infty$.

For $\mathbb{E}[|r(X,t)m(X,s)|]$, we have

$$\mathbb{E}\{|r(X,t)m(X,s)|\} \leq \|r(X,t)\|_{\mathcal{L}^2\{F(x)\}}^{1/2} \cdot \|m(X,s)\|_{\mathcal{L}^2\{F(x)\}}^{1/2}$$
$$\leq \|r(X,t)\|_{\mathcal{H}_X}^{1/2} \cdot \|m(X,s)\|_{\mathcal{L}^2\{F(x)\}}^{1/2},$$

where the last inequality follows from (21) by assumption C.3. Since $\|r(X,t)\|_{\mathcal{H}_X} < \infty$ and the selected weight function $m$ satisfies $\|m(X,s)\|_{\mathcal{L}^2\{F(x)\}} < \infty$, we have $\mathbb{E}\{|r(X,t)m(X,s)|\} < \infty$.

For $\mathbb{E}[|\{H^0(W,t) - \widehat{H}^0(W,t)\}m(X,s)|]$, we have

$$\mathbb{E}[|\{H^0(W,t) - \widehat{H}^0(W,t)\}m(X,s)|] = \mathbb{E}[|\{H^0(W,t) - \widehat{H}^0(W,t)\}\mathbb{E}\{m(X,s)|W\}|]$$
$$\leq C \cdot \mathbb{E}\{|H^0(W,t) - \widehat{H}^0(W,t)|\}$$
$$\leq C \cdot \|H^0(W,t) - \widehat{H}^0(W,t)\|_{H_W},$$

where the second inequality follows from assumption 4.9 and the last inequality follows from (21) by assumption C.3. Since $H^0(W,t) - \widehat{H}^0(W,t) \in \mathcal{H}_W$, we have $\mathbb{E}\left[\{H^0(W,t) - \widehat{H}^0(W,t)\}m(X,s)\right] < \infty$.

According to the law of large numbers, we know that it converges to $\mathbb{E}\left[\{r(X,t) + H^0(W,t) - \widehat{H}^0(W,t)\}m(X,s)\right]$. We assert that if $\mathbb{H}_1^{\mathrm{fix}}$ holds, then exists $t$ satisfies $\mathbb{E}\left[\{r(X,t) + H^0(W,t) - \widehat{H}^0(W,t)\}|X\right] \neq 0$. Proof by contradiction. If $r(X,t) = \mathbb{E}\left[\{H^0(W,t) - \widehat{H}^0(W,t)\}|X\right]$ holds for any $t$, we have:

$$\mathbb{E}\left\{\varphi(Y,t) - H^0(W,t)|X\right\} = \mathbb{E}\{\widehat{H}^0(W,t) - H^0(W,t)|X\},$$

which implies $\mathbb{E}\left\{\varphi(Y,t)|X\right\} = \mathbb{E}\{\widehat{H}^0(W,t)|X\}$ for any $t$. This is not consistent with $\mathbb{H}_1^{\mathrm{fix}}$.

Combining these results, we have:

$$\lim_{n\to\infty} \max_{t\in\mathcal{T}} |T_n(s,t)| = \lim_{n\to\infty} \sqrt{n}\{O_p(1) + o_p(1)\} = \infty.$$

for almost all $s$ under $\mathbb{H}_1^{\mathrm{fix}}$.

**(ii). The case of $\mathbb{H}_{1n}^{\alpha}$ with $0 < \alpha < 1/2$.**

We first decompose the term into $\mathbb{P}\left[\{\widehat{H}^{\lambda}(W,t) - \widehat{H}^0(W,t)\}m(X,s)\right]$ into $\sum_{i=1}^{6} G_i$, where

$$G_1 := (\lambda I + A^*A)^{-1}A^*(\widehat{b}_t - \widehat{A}H_t^0)$$
$$G_2 := (\lambda I + A^*A)^{-1}(\widehat{A}^* - A^*)(\widehat{b}_t - \widehat{A}H_t^0);$$
$$G_3 := \left\{(\lambda I + \widehat{A}^*\widehat{A})^{-1} - (\lambda I + A^*A)^{-1}\right\}\widehat{A}^*(\widehat{b}_t - \widehat{A}H_t^0);$$
$$G_4 := (\lambda I + \widehat{A}^*\widehat{A})^{-1}\widehat{A}^*\widehat{A}\widehat{A}H_t^0 - (\lambda I + A^*A)^{-1}A^*b_t;$$
$$G_5 := (\lambda I + A^*A)^{-1}A^*b_t - \widehat{A}H_t^0.$$
$$G_6 := H^0(w,t) - \widehat{H}^0(w,t) = H^0(w,t) - (A^*A)^{-1}A^*b_t.$$

We define $\ell(X,t) = \mathbb{E}\{r(X,t)\phi_X(X)\}$.

For $\mathbb{P}\{G_2 m(X,s)\}$, following Lemma D.5, we have

$$
\begin{aligned}
\mathbb{P}\{G_2 m(X,s)\} &= \langle (\lambda I + A^*A)^{-1}(\widehat{A}^* - A^*)(\widehat{b}_t - \widehat{A}H_t^0), g_s \rangle_{\mathcal{H}_W} \\
&= \langle \widehat{b}_t - \widehat{A}H_t^0, (\widehat{A} - A)(\lambda I + A^*A)^{-1} g_s \rangle_{\mathcal{H}_X} \\
&= \langle \widehat{b}_t - b_t + b_t - \widehat{A}H_t^0, (\widehat{A} - A)(\lambda I + A^*A)^{-1} g_s \rangle_{\mathcal{H}_X} \\
&= \langle \widehat{b}_t - b_t + AH_t^0 + \ell(X,t)/n^\alpha - \widehat{A}H_t^0, (\widehat{A} - A)(\lambda I + A^*A)^{-1} g_s \rangle_{\mathcal{H}_X} \\
&= \langle \ell(X,t)/n^\alpha, (\widehat{A} - A)(\lambda I + A^*A)^{-1} g_s \rangle + \langle \widehat{b}_t - b_t, (\widehat{A} - A)(\lambda I + A^*A)^{-1} g_s \rangle_{\mathcal{H}_X} \\
&\quad + \langle AH_t^0 - \widehat{A}H_t^0, (\widehat{A} - A)(\lambda I + A^*A)^{-1} g_s \rangle_{\mathcal{H}_X}.
\end{aligned}
$$

By Lemma D.12, we have $\|\widehat{b}_t - b_t\|_{\mathcal{H}_X} = O_p(1/\sqrt{n})$ and $\|\widehat{A} - A\|_{\mathrm{op}} = O_p(1/\sqrt{n})$. By Lemma D.10 (d), we have $\|(\lambda I + A^*A)^{-1} g_s\|_{\mathcal{H}_W} = O_p\{\lambda^{\frac{\min(\eta,2)}{2}-1}\}$. Therefore, we have

$$
\begin{aligned}
\mathbb{P}\{G_2 m(X,s)\} &\leq \langle \ell(X,t)/n^\alpha, (\widehat{A} - A)(\lambda I + A^*A)^{-1} g_s \rangle_{\mathcal{H}_X} + \|\widehat{b}_t - b_t\|_{\mathcal{H}_X} \cdot \|\widehat{A} - A\|_{\mathrm{op}} \cdot \|(\lambda I + A^*A)^{-1} g_s\|_{\mathcal{H}_W} \\
&\quad + \|\widehat{A} - A\|_{\mathrm{op}} \cdot \|H_t^0\|_{\mathcal{H}_W} \cdot \|\widehat{A} - A\|_{\mathrm{op}} \cdot \|(\lambda I + A^*A)^{-1} g_s\|_{\mathcal{H}_W} \\
&\leq \langle \ell(X,t)/n^\alpha, (\widehat{A}^* - A^*)(\lambda I + A^*A)^{-1} g_s \rangle_{\mathcal{H}_X} + O_p(1/n).
\end{aligned}
$$

For $\mathbb{P}\{G_3 m(X,s)\}$, following Lemma D.6, we have

$$
\begin{aligned}
\mathbb{P}\{G_3 m(X,s)\} &= \left\langle \left\{ (\lambda I + \widehat{A}^*\widehat{A})^{-1} - (\lambda I + A^*A)^{-1} \right\} \widehat{A}^*(\widehat{b}_t - \widehat{A}H_t^0), g_s \right\rangle_{\mathcal{H}_W} \\
&= \left\langle (\lambda I + A^*A)^{-1}(A^*A - \widehat{A}^*\widehat{A})(\lambda I + \widehat{A}^*\widehat{A})^{-1}\widehat{A}^*(\widehat{b}_t - \widehat{A}H_t^0), g_s \right\rangle_{\mathcal{H}_W} \\
&= \langle \widehat{b}_t - b_t + b_t - \widehat{A}H_t^0, \widehat{A}(\lambda I + \widehat{A}^*\widehat{A})^{-1}(A^*A - \widehat{A}^*\widehat{A})(\lambda I + A^*A)^{-1} g_s \rangle_{\mathcal{H}_X} \\
&= \langle \widehat{b}_t - b_t + AH_t^0 + \ell(X,t)/n^\alpha - \widehat{A}H_t^0, \widehat{A}(\lambda I + \widehat{A}^*\widehat{A})^{-1}(A^*A - \widehat{A}^*\widehat{A})(\lambda I + A^*A)^{-1} g_s \rangle_{\mathcal{H}_X} \\
&= \langle \ell(X,t)/n^\alpha, \widehat{A}(\lambda I + \widehat{A}^*\widehat{A})^{-1}(A^*A - \widehat{A}^*\widehat{A})(\lambda I + A^*A)^{-1} g_s \rangle_{\mathcal{H}_X} \\
&\quad + \langle \widehat{b}_t - b_t, \widehat{A}(\lambda I + \widehat{A}^*\widehat{A})^{-1}(A^*A - \widehat{A}^*\widehat{A})(\lambda I + A^*A)^{-1} g_s \rangle_{\mathcal{H}_X} \\
&\quad + \langle AH_t^0 - \widehat{A}H_t^0, \widehat{A}(\lambda I + \widehat{A}^*\widehat{A})^{-1}(A^*A - \widehat{A}^*\widehat{A})(\lambda I + A^*A)^{-1} g_s \rangle_{\mathcal{H}_X}.
\end{aligned}
$$

By Lemma D.12, we have $\|\widehat{b}_t - b_t\|_{\mathcal{H}_X} = O_p(1/\sqrt{n})$ and $\|\widehat{A} - A\|_{\mathrm{op}} = O_p(1/\sqrt{n})$. By Lemma D.10 (d), we have $\|(\lambda I + A^*A)^{-1} g_s\|_{\mathcal{H}_W} = O_p\{\lambda^{\frac{\min(\eta,2)}{2}-1}\}$. By Lemma D.10 (c), we have $\|\widehat{A}(\lambda I + \widehat{A}^*\widehat{A})^{-1}\|_{\mathrm{op}} = O_p(1/\sqrt{\lambda})$. Therefore, we have

$$
\begin{aligned}
\mathbb{P}\{G_3 m(X,s)\} &\leq \langle \ell(X,t)/n^\alpha, \widehat{A}(\lambda I + \widehat{A}^*\widehat{A})^{-1}(A^*A - \widehat{A}^*\widehat{A})(\lambda I + A^*A)^{-1} g_s \rangle_{\mathcal{H}_X} \\
&\quad + \|\widehat{b}_t - b_t\|_{\mathcal{H}_X} \cdot \|\widehat{A}(\lambda I + \widehat{A}^*\widehat{A})^{-1}\|_{\mathrm{op}} \cdot \|A^*A - \widehat{A}^*\widehat{A}\|_{\mathrm{op}} \cdot \|(\lambda I + A^*A)^{-1} g_s\|_{\mathcal{H}_W} \\
&\quad + \|\widehat{A} - A\|_{\mathrm{op}} \cdot \|H_t^0\|_{\mathcal{H}_W} \cdot \|\widehat{A}(\lambda I + \widehat{A}^*\widehat{A})^{-1}\|_{\mathrm{op}} \cdot \|A^*A - \widehat{A}^*\widehat{A}\|_{\mathrm{op}} \cdot \|(\lambda I + A^*A)^{-1} g_s\|_{\mathcal{H}_W} \\
&\leq \langle \ell(X,t)/n^\alpha, \widehat{A}(\lambda I + \widehat{A}^*\widehat{A})^{-1}(A^*A - \widehat{A}^*\widehat{A})(\lambda I + A^*A)^{-1} g_s \rangle_{\mathcal{H}_X} + O_p\left( \frac{1}{n\sqrt{\lambda}} \right).
\end{aligned}
$$

For $G_4$ and $G_5$, noted that

$$
\begin{aligned}
G_4 + G_5 &= (\lambda I + \widehat{A}^*\widehat{A})^{-1}\widehat{A}^*\widehat{A}H_t^0 - (\lambda I + A^*A)^{-1}A^*b_t + (\lambda I + A^*A)^{-1}A^*b_t - H_t^0 \\
&= (\lambda I + \widehat{A}^*\widehat{A})^{-1}\widehat{A}^*\widehat{A}H_t^0 - (\lambda I + A^*A)^{-1}A^*(AH_t^0 + r/\sqrt{n}) + (\lambda I + A^*A)^{-1}A^*(AH_t^0 + r/\sqrt{n}) - H_t^0 \\
&= \underbrace{(\lambda I + \widehat{A}^*\widehat{A})^{-1}\widehat{A}^*\widehat{A}H_t^0 - (\lambda I + A^*A)^{-1}A^*AH_t^0}_{\overline{G}_4} + \underbrace{(\lambda I + A^*A)^{-1}A^*AH_t^0 - H_t^0}_{\overline{G}_5},
\end{aligned}
$$

where $\overline{G}_4$ and $\overline{G}_5$ are those defined in Proposition D.4. Therefore, we have $\mathbb{P}\{G_4 m(X,s)\} = O_p(1/n) + \lambda^{1/2}/\sqrt{n}$ and $\mathbb{P}\{G_5 m(X,s)\} = O_p(\lambda)$.

For $\mathbb{P}\{G_1 m(X, s)\}$, following Lemma D.9, we have

$$
\begin{aligned}
\mathbb{P}\{G_1 m(X, s)\} &= \langle (\lambda I + A^*A)^{-1} A^*(\widehat{b}_t - \widehat{A}H_t^0), g_s \rangle_{\mathcal{H}_W} \\
&= \langle A^*(\widehat{b}_t - \widehat{A}H_t^0), (\lambda I + A^*A)^{-1} g_s \rangle_{\mathcal{H}_W} \\
&= \underbrace{\langle A^*(\widehat{b}_t - \widehat{A}H_t^0), \left\{ (\lambda I + A^*A)^{-1} - (A^*A)^{-1} \right\} g_s \rangle_{\mathcal{H}_W}}_{\text{(I)}} + \underbrace{\langle A^*(\widehat{b}_t - \widehat{A}H_t^0), (A^*A)^{-1} g_s \rangle_{\mathcal{H}_W}}_{\text{(II)}}.
\end{aligned}
$$

For (II), following Lemma D.9, we have

$$
\begin{aligned}
\langle A^*(\widehat{b}_t - \widehat{A}H_t^0), (A^*A)^{-1} g_s \rangle_{\mathcal{H}_W} &= \left\langle A^* \left\{ \frac{1}{n} \sum_{i=1}^n \varphi(y_i, t) \phi_X(x_i) - \frac{1}{n} \sum_{i=1}^n H^0(w_i, t) \phi_X(x_i) \right\}, (A^*A)^{-1} g_s \right\rangle_{\mathcal{H}_W} \\
&= \left\langle A^* \left\{ \frac{1}{n} \sum_{i=1}^n \left\{ U(w_i, y_i, t) + r(x_i, t)/n^\alpha \right\} \phi_X(x_i) \right\}, (A^*A)^{-1} g_s \right\rangle_{\mathcal{H}_W} \\
&= \left\langle A^* \left\{ \frac{1}{n} \sum_{i=1}^n \left\{ U(w_i, y_i, t) \right\} \phi_X(x_i) \right\}, (A^*A)^{-1} g_s \right\rangle_{\mathcal{H}_W} \\
&\quad + \left\langle A^* \left\{ \frac{1}{n} \sum_{i=1}^n r(x_i, t)/n^\alpha \phi_X(x_i) \right\}, (A^*A)^{-1} g_s \right\rangle_{\mathcal{H}_W} \\
&= \frac{1}{n} \sum_{i=1}^n U(w_i, y_i, t) \{A(A^*A)^{-1} g_s\}(x_i) \\
&\quad + \left\langle A^* \left\{ \frac{1}{n} \sum_{i=1}^n r(x_i, t)/n^\alpha \phi_X(x_i) \right\}, (A^*A)^{-1} g_s \right\rangle_{\mathcal{H}_W}.
\end{aligned}
$$

For (I), we have

$$
\begin{aligned}
&\langle A^*(\widehat{b}_t - \widehat{A}H_t^0), \left\{ (\lambda I + A^*A)^{-1} - (A^*A)^{-1} \right\} g_s \rangle_{\mathcal{H}_W} \\
&= \langle A^*(\widehat{b}_t - b_t + b_t - \widehat{A}H_t^0), \left\{ (\lambda I + A^*A)^{-1} - (A^*A)^{-1} \right\} g_s \rangle_{\mathcal{H}_W} \\
&= \langle A^*(\widehat{b}_t - b_t + AH_t^0 + \ell(X, t)/n^\alpha - \widehat{A}H_t^0), \left\{ (\lambda I + A^*A)^{-1} - (A^*A)^{-1} \right\} g_s \rangle_{\mathcal{H}_W} \\
&= \langle A^*(\widehat{b}_t - b_t, \left\{ (\lambda I + A^*A)^{-1} - (A^*A)^{-1} \right\} g_s \rangle_{\mathcal{H}_W} + \langle A^*(AH_t^0 - \widehat{A}H_t^0), \left\{ (\lambda I + A^*A)^{-1} - (A^*A)^{-1} \right\} g_s \rangle_{\mathcal{H}_W} \\
&\quad + \langle A^*\ell(X, t)/n^\alpha, \left\{ (\lambda I + A^*A)^{-1} - (A^*A)^{-1} \right\} g_s \rangle_{\mathcal{H}_W}.
\end{aligned}
$$

By Lemma D.12, we have $\|\widehat{b}_t - b_t\|_{\mathcal{H}_X} = O_p(1/\sqrt{n})$ and $\|\widehat{A} - A\|_{\text{op}} = O_p(1/\sqrt{n})$. By Lemma D.9, we have $\| \left\{ (\lambda I + A^*A)^{-1} - (A^*A)^{-1} \right\} g_s \| = o_p(1)$. Thus, we have

$$
\langle A^*(\widehat{b}_t - \widehat{A}H_t^0), \left\{ (\lambda I + A^*A)^{-1} - (A^*A)^{-1} \right\} g_s \rangle_{\mathcal{H}_W} \leq o_p(1/\sqrt{n}) + \langle A^*\ell(X, t)/n^\alpha, \left\{ (\lambda I + A^*A)^{-1} - (A^*A)^{-1} \right\} g_s \rangle_{\mathcal{H}_W}.
$$

For $\mathbb{P}\{G_6 m(X, s)\}$, we have

$$
\begin{aligned}
\langle (A^*A)^{-1} A^* b_t - H_t^0, g_s \rangle_{\mathcal{H}_W} &= \langle (A^*A)^{-1} A^* \left( AH_t^0 + \ell(X, t)/n^\alpha \right) - H_t^0, g_s \rangle_{\mathcal{H}_W} \\
&= \langle A^* \left( \ell(X, t)/n^\alpha \right), (A^*A)^{-1} g_s \rangle_{\mathcal{H}_W}.
\end{aligned}
$$

Combining these results, we get

$$\sqrt{n}\sum_{i=1}^{6}\mathbb{P}\{G_i m(X,s)\} = \frac{1}{\sqrt{n}}\sum_{i=1}^{n}U(w_i,y_i,t)\{A(A^*A)^{-1}g_s\}(x_i) + \underbrace{\frac{\sqrt{n}}{n^\alpha}\langle\ell(X,t),(\widehat{A}^* - A^*)(\lambda I + A^*A)^{-1}g_s\rangle_{\mathcal{H}_X}}_{(I)}$$

$$+ \underbrace{\frac{\sqrt{n}}{n^\alpha}\langle\ell(X,t),\widehat{A}(\lambda I + \widehat{A}^*\widehat{A})^{-1}(A^*A - \widehat{A}^*\widehat{A})(\lambda I + A^*A)^{-1}g_s\rangle_{\mathcal{H}_X}}_{(II)}$$

$$+ \underbrace{\frac{\sqrt{n}}{n^\alpha}\langle A^*\ell(X,t),\{(\lambda I + A^*A)^{-1} - (A^*A)^{-1}\}g_s\rangle_{\mathcal{H}_W}}_{(III)}$$

$$+ \frac{\sqrt{n}}{n^\alpha}\langle A^*\left\{\frac{1}{n}\sum_{i=1}^{n}r(x_i,t)\phi_X(x_i)\right\}, (A^*A)^{-1}g_s\rangle_{\mathcal{H}_W} - \frac{\sqrt{n}}{n^\alpha}\langle A^*(\ell(X,t)),(A^*A)^{-1}g_s\rangle_{\mathcal{H}_W}.$$

By Lemma D.12, we have $\|\widehat{b}_t - b_t\|_{\mathcal{H}_X} = O_p(1/\sqrt{n})$ and $\|\widehat{A} - A\|_{\mathrm{op}} = O_p(1/\sqrt{n})$. By Lemma D.13, we have $\|A^*A - \widehat{A}^*\widehat{A}\| = O_p(1/\sqrt{n})$. By Lemma D.10 (d), we have $\|(\lambda I + A^*A)^{-1}g_s\|_{\mathcal{H}_W} = O_p\{\lambda^{\frac{\min(\eta,2)}{2}-1}\}$. By Lemma D.10 (c), we have $\|\widehat{A}(\lambda I + \widehat{A}^*\widehat{A})^{-1}\|_{\mathrm{op}} = O_p(1/\sqrt{\lambda})$. By Lemma D.9, we have $\|\{(\lambda I + A^*A)^{-1} - (A^*A)^{-1}\}g_s\| = o_p(1)$.

Thus, for the term (I), we have

$$(I) \le \frac{\sqrt{n}}{n^\alpha}\cdot\|\ell(X,t)\|\cdot\|\widehat{A}^* - A^*\|_{\mathrm{op}}\cdot\|(\lambda I + A^*A)^{-1}g_s\|_{\mathcal{H}_W}$$

$$\le \frac{\sqrt{n}}{n^\alpha}\cdot O_p(1/\sqrt{n}) = o_p\left(\frac{\sqrt{n}}{n^\alpha}\right).$$

For term (II), we have

$$(II) \le \frac{\sqrt{n}}{n^\alpha}\cdot\|\ell(X,t)\|\cdot\|\widehat{A}(\lambda I + \widehat{A}^*\widehat{A})^{-1}\|_{\mathrm{op}}\cdot\|A^*A - \widehat{A}^*\widehat{A}\|_{\mathrm{op}}\cdot\|(\lambda I + A^*A)^{-1}g_s\|_{\mathcal{H}_W}$$

$$\le \frac{\sqrt{n}}{n^\alpha}\cdot O_p(1/\sqrt{n\lambda}) = o_p\left(\frac{\sqrt{n}}{n^\alpha}\right).$$

For term (III), we have

$$(III) \le \frac{\sqrt{n}}{n^\alpha}\cdot\|A^*\|\cdot\|\ell(X,t)\|\cdot\|\{(\lambda I + A^*A)^{-1} - (A^*A)^{-1}\}g_s\|_{\mathcal{H}_W}$$

$$\le \frac{\sqrt{n}}{n^\alpha}\cdot o_p(1) = o_p\left(\frac{\sqrt{n}}{n^\alpha}\right).$$

For the last term, we have

$$\frac{\sqrt{n}}{n^\alpha}\left\langle A^*\left\{\frac{1}{n}\sum_{i=1}^{n}r(x_i,t)\phi_X(x_i)\right\}, (A^*A)^{-1}g_s\right\rangle_{\mathcal{H}_W} - \frac{\sqrt{n}}{n^\alpha}\left\langle A^*(\ell(X,t)),(A^*A)^{-1}g_s\right\rangle_{\mathcal{H}_W}$$

$$= \frac{\sqrt{n}}{n^\alpha}\left[\left\langle\left\{\frac{1}{n}\sum_{i=1}^{n}r(x_i,t)\phi_X(x_i)\right\} - \mathbb{E}[r(X,t)\phi_X(X)], A(A^*A)^{-1}g_s\right\rangle_{\mathcal{H}_W}\right].$$

By Lemma 11 of (Mastouri et al., 2021), we have $\|\frac{1}{n}\sum_{i=1}^{n}r(x_i,t)\phi_X(x_i) - \mathbb{E}[r(X,t)\phi_X(X)]\| = O_p(1/\sqrt{n})$. Combining all these results, we have:

$$\sqrt{n}\sum_{i=1}^{6}\mathbb{P}\{G_i m(X,s)\} = \frac{1}{\sqrt{n}}\sum_{i=1}^{n}U(w_i,y_i,t)\{A(A^*A)^{-1}g_s\}(x_i) + o_p\left(\frac{\sqrt{n}}{n^\alpha}\right) + o_p(1). \tag{53}$$

Besides, for the first term of $T_n(s,t)$, we have

$$\mathbb{P}_n\left[\{\varphi(Y,t) - \widehat{H}^0(W,t)\}m(X,s)\right] = \mathbb{P}_n\left[\{\varphi(Y,t) - H^0(W,t) + H^0(W,t) - \widehat{H}^0(W,t)\}m(X,s)\right]$$
$$= \mathbb{P}_n\left[\{U(W,Y,t) + r(X,t)/n^\alpha + H^0(W,t) - \widehat{H}^0(W,t)\}m(X,s)\right].$$

Therefore, combining (53), we have

$$T_n(s,t) = \sqrt{n}\mathbb{P}_n\left(U(W,Y,t)\left[m(X,s) - \{A(A^*A)^{-1}g_s\}(x_i)\right]\right)$$
$$+ \sqrt{n}\mathbb{P}_n\left[\{r(X,t)/n^\alpha + H^0(W,t) - \widehat{H}^0(W,t)\}m(X,s)\right] + o_p\left(\frac{\sqrt{n}}{n^\alpha}\right) + o_p(1).$$

For the first term, according to Theorem 4.12, we have $\sqrt{n}\mathbb{P}_n\left(U(W,Y,t)\left[m(X,s) - \{A(A^*A)^{-1}g_s\}(x_i)\right]\right)$ converges weakly to $\mathbb{G}(s,t)$. For the second term, we have

$$\sqrt{n}\mathbb{P}_n\left[\{r(X,t)/n^\alpha + H^0(W,t) - \widehat{H}^0(W,t)\}m(X,s)\right]$$
$$= \sqrt{n}\mathbb{P}_n\left[\{r(X,t)/n^\alpha + H^0(W,t) - A^*A)^{-1}A^*\{AH^0(W,t) + r(X,t)/n^\alpha\}\}m(X,s)\right]$$
$$= \frac{\sqrt{n}}{n^\alpha}\mathbb{P}_n\left[\{r(X,t) - (A^*A)^{-1}A^*r(X,t)\}m(X,s)\right].$$

According to the law of large numbers, we know that it converges to $\mathbb{E}\left[\{r(X,t)/n^\alpha + H^0(W,t) - \widehat{H}^0(W,t)\}m(X,s)\right]$. We assert that if $\mathbb{H}^\alpha_{1n}$ holds, then for any $t$ $\mathbb{E}\left[\{r(X,t)/n^\alpha + H^0(W,t) - \widehat{H}^0(W,t)\}|X\right] \neq 0$. Proof by contradiction. If $r(X,t)/n^\alpha = \mathbb{E}\left[\{H^0(W,t) - \widehat{H}^0(W,t)\}|X\right]$ for any $t$, we have

$$\mathbb{E}\{\varphi(Y,t) - H^0(W,t)|X\} = \mathbb{E}\{\widehat{H}^0(W,t) - H^0(W,t)|X\} \text{ for any } t.$$

This implies $\mathbb{E}\{\varphi(Y,t) - \widehat{H}^0(W,t)|X\} = 0$ for any $t$, which is not consistent with $\mathbb{H}^\alpha_{1n}$. Thus, we have $\mathbb{E}\left[\{r(X,t)/n^\alpha + H^0(W,t) - \widehat{H}^0(W,t)\}|X\right] \neq 0$, which implies $\mathbb{E}\left[\{r(X,t) - (A^*A)^{-1}A^*r(X,t)\}m(X,s)\right] \neq 0$. Combining these results, we have

$$\lim_{n\to\infty}\max_{t\in\mathcal{T}}|T_n(s,t)| = \underbrace{O_p(1)}_{(\star)} + \lim_{n\to\infty}\sqrt{n}\left\{\frac{1}{n^\alpha}O_p(1) + o_p\left(\frac{\sqrt{n}}{n^\alpha}\right) + o_p(1)\right\} = \infty.$$

for almost all $s$ under $\mathbb{H}^\alpha_{1n}(0 < \alpha < 1/2)$, where $(\star)$ follows from the Gaussian process.

**(iii). The case of $\mathbb{H}^\alpha_{1n}$ with $\alpha = 1/2$.**

Following the proof of $\mathbb{H}^\alpha_{1n}$ with $0 < \alpha < 1/2$, we have

$$T_n(s,t) = \sqrt{n}\mathbb{P}_n\left(U(W,Y,t)\left[m(X,s) - \{A(A^*A)^{-1}g_s\}(x_i)\right]\right)$$
$$+ \sqrt{n}\mathbb{P}_n\left[\{r(X,t)/n^\alpha + H^0(W,t) - \widehat{H}^0(W,t)\}m(X,s)\right] + o_p\left(\frac{\sqrt{n}}{n^\alpha}\right) + o_p(1).$$

Taking $\alpha = 1/2$, we obtain

$$T_n(s,t) = \sqrt{n}\mathbb{P}_n\left(U(W,Y,t)\left[m(X,s) - \{A(A^*A)^{-1}g_s\}(x_i)\right]\right)$$
$$+ \sqrt{n}\mathbb{P}_n\left[\{r(X,t)/n^{1/2} + H^0(W,t) - \widehat{H}^0(W,t)\}m(X,s)\right] + o_p(1).$$

For the first term, according to Theorem 4.12, we have $\sqrt{n}\mathbb{P}_n\left(U(W,Y,t)\left[m(X,s) - \{A(A^*A)^{-1}g_s\}(x_i)\right]\right)$ converges weakly to $\mathbb{G}(s,t)$. For the second term, we have

$$\sqrt{n}\mathbb{P}_n\left[\{r(X,t)/n^{1/2} + H^0(W,t) - \widehat{H}^0(W,t)\}m(X,s)\right]$$
$$= \sqrt{n}\mathbb{P}_n\left[\{r(X,t)/n^{1/2} + H^0(W,t) - A^*A)^{-1}A^*\{AH^0(W,t) + r(X,t)/n^{1/2}\}\}m(X,s)\right]$$
$$= \mathbb{P}_n\left[\{r(X,t) - (A^*A)^{-1}A^*r(X,t)\}m(X,s)\right].$$

Thus, we have

$$T_n(s,t) = \frac{1}{\sqrt{n}} \sum_{i=1}^{n} \{U(w_i, y_i, t)\} [\{A(A^*A)^{-1}g_s\}(x_i) + m(x_i, s)]$$

$$+ \frac{1}{n} \sum_{i=1}^{n} \{r(x_i, t) - (A^*A)^{-1}A^*r(X,t)\}m(x_i, s) + o_p(1).$$

Besides, we have:

$$\mathbb{E}\left(\{U(w_i, y_i, t) - r(x_i, t)/\sqrt{n}\}\left[m(x_i, s) - \{A(A^*A)^{-1}g_s\}(x_i)\right]\right)^2$$
$$= \mathbb{E}(U(w_i, y_i, t)[m(x_i, s) - \{A(A^*A)^{-1}g_s\}(x_i)])^2 + n^{-1}\mathbb{E}\left(r(x_i, t)^2[m(x_i, s)\{A(A^*A)^{-1}g_s\}(x_i)]^2\right)$$
$$- 2n^{-1/2}\mathbb{E}\left(\mathbb{E}\{U(w_i, y_i, t)|x_i\}r(x_i, t)[m(x_i, s) - \{A(A^*A)^{-1}g_s\}(x_i)]^2\right)$$
$$\overset{(1)}{=} \underbrace{\mathbb{E}(U(w_i, y_i, t)[m(x_i, s) - \{A(A^*A)^{-1}g_s\}(x_i)])^2}_{(I)} - n^{-1}\underbrace{\mathbb{E}\left(r(x_i, t)^2[m(x_i, s) - \{A(A^*A)^{-1}g_s\}(x_i)]^2.\right)}_{(II)},$$

where (1) follows from $\mathbb{E}\{U(w_i, y_i, t)|x_i\} = r(x_i, t)/\sqrt{n}$ under $\mathbb{H}_{1n}^\alpha$. Following 4.12, we have $(I) < \infty$. Besides, for the second term, we have $(II) \leq 2\mathbb{E}\{r(x_i, t)^4\} + 2\mathbb{E}\left([m(x_i, s) - \{A(A^*A)^{-1}g_s\}(x_i)]^4\right) < \infty$ by inequality $a^2b^2 \leq (a^4 + b^4)/2$ and assumption 4.11. Thus, we have $\mathbb{E}(\{U(w_i, y_i, t) - r(x_i, t)/\sqrt{n}\}[m(x_i, s) - \{A(A^*A)^{-1}g_s\}(x_i)])^2 \to \mathbb{E}(\{U(w_i, y_i, t)\}[m(x_i, s) - \{A(A^*A)^{-1}g_s\}(x_i)])^2$ as $n \to \infty$. Therefore, the first term of $T_n(s,t)$ converges weakly to $\mathbb{G}(s,t)$ in $\mathcal{L}^2\{\mathcal{T} \times \mathcal{T}, \mu \times \mu\}$ by Lemma D.15, where $\mathbb{G}(s,t)$ is defined Theorem 4.12.

Besides, we need to prove $\mathbb{E}\left[\{r(X,t) - A(A^*A)^{-1}r(X,t)\}m(X,s)\right] < \infty$. In fact, we only proof $\mathbb{E}\left[r(X,t)m(X,s)\right] < \infty$ and $\mathbb{E}\left[(A^*A)^{-1}A^*r(X,t) \cdot m(X,s)\right] < \infty$.

For $\mathbb{E}[|r(X,t)m(X,s)|]$, we have

$$\mathbb{E}\{|r(X,t)m(X,s)|\} \leq \|r(X,t)\|_{\mathcal{L}^2\{F(x)\}}^{1/2} \cdot \|m(X,s)\|_{\mathcal{L}^2\{F(x)\}}^{1/2}$$
$$\leq \|r(X,t)\|_{\mathcal{H}_X}^{1/2} \cdot \|m(X,s)\|_{\mathcal{L}^2\{F(x)\}}^{1/2},$$

where the last inequality follows from (21) by assumption C.3. Since $\|r(X,t)\|_{\mathcal{H}_X} < \infty$ and the selected weight function $m$ satisfies $\|m(X,s)\|_{\mathcal{L}^2\{F(x)\}} < \infty$, we have $\mathbb{E}\{|r(X,t)m(X,s)|\} < \infty$.

For $\mathbb{E}\left[|(A^*A)^{-1}A^*r(X,t) \cdot m(X,s)|\right]$, we have

$$\mathbb{E}\left[(A^*A)^{-1}A^*r(X,t) \cdot m(X,s)|\right] = \mathbb{E}[|\{(A^*A)^{-1}A^*r(X,t)\}\mathbb{E}\{m(X,s)|W\}|]$$
$$\leq C \cdot \mathbb{E}\{|(A^*A)^{-1}A^*r(X,t)|\}$$
$$\leq C \cdot \|(A^*A)^{-1}A^*r(X,t)\|_{H_W},$$

where the second inequality follows from assumption 4.9 and the last inequality follows from (21) by assumption C.3. Since $(A^*A)^{-1}A^*r(X,t) \in \mathcal{H}_W$, we have $\mathbb{E}\left[\{(A^*A)^{-1}A^*r(X,t)\}m(X,s)\right] < \infty$.

Therefore, the second term of $T_n(s,t)$ converges weakly to $\mu(X,s,t) := \mathbb{E}\left[\{r(X,t) - (A^*A)^{-1}A^*r(X,t)\}m(X,s)\right]$. By Slutsky's theorem, we have $T_n(s,t)$ converges weakly to $\mathbb{G}(s,t) + \mu(X,s,t)$ in $\mathcal{L}^2\{\mathcal{T} \times \mathcal{T}, \mu \times \mu\}$ under $\mathbb{H}_{1n}^\alpha$ with $\alpha = 1/2$. □

**Corollary 4.16.** *Suppose assumptions in Theorem 4.12 hold. If $\varphi(y,t)$ is continuous with respect to $t$ for each $y$, then $\widehat{\Delta}_{\varphi,m}$ is weakly convergent to $\max_{t\in\mathcal{T}} \int_{\mathcal{T}} |\mathbb{G}(s,t)|^2 d\mu(s)$ under $\mathbb{H}_0$, as $n, K \to \infty$. Besides, conditional on the original sample $\{y_i, w_i, x_i\}_{i=1}^n$, the bootstrapped statistics (8) is also weakly convergent to the $\max_{t\in\mathcal{T}} \int_{\mathcal{T}} |\mathbb{G}(s,t)|^2 d\mu(s)$.*

*Proof.* **(i). $\widehat{\Delta}_{\varphi,m}$ is weakly convergent to $\max_{t\in\mathcal{T}} \int |\mathbb{G}(s,t)|^2 d\mu(s)$.**

We denote $X_n(t) := \int_{\mathcal{T}} \{T_n(s,t)\}^2 d\mu(s)$ and $X_n(t)$ weakly converges to $X(t) = \int_{\mathcal{T}} |\mathbb{G}(s,t)|^2 d\mu(s)$. Since integral of the Gaussian process $\mathbb{G}(s,t)$ still a Gaussian process with respect to $t$, we can obtain the variance $\int_{\mathcal{T}} |\text{Var}\{\eta(X,W,Y,s,t)\}|^2 d\mu(s)$. Besides, for the Gaussian process, $X(t)$ is continuous in probability if and only if

its mean and variance are continuous following (Seeger, 2004). Since $\varphi(y,t)$ is continuous with respect to $t$, the variance is continuous. Therefore, $X(t)$ is continuous in probability. Assume that we obtains the maximum value at $t_0$, *i.e.* $\max_{t \in T} X(t) = X(t_0)$. Since the process $X(t)$ is continuous in probability, we have that, for any $\varepsilon > 0$, there exists $\delta$ such that as long as $|t - t_0| < \delta$, $\mathbb{P}(|X(t) - X(t_0)| > \varepsilon/3) < \varepsilon$.

Since $\{t_1, ..., t_K\}$ are evaluated at a grid of equidistant indices, for any $t_0 \in \mathcal{T}$, we have $\lim_{K \to \infty} \min_k |t_0 - t_k| = 0$. That means, for any $\delta > 0$, there exists $K_0$, such that as long as $K > K_0$, there exists $t_k$ with $1 \leq k \leq K$, $|t_k - t_0| < \delta$. Further, for any finite $t_1, ..., t_K$, denote $\mathcal{T}_K := \{k : X(t_k) = \max_{j \leq K} X(t_j)\}$ and set $\delta_0 := X(t_{k_0}) - X(t_{k_1})$, where $t_{k_0} \in \mathcal{T}_K$ and $X(t_{k_1}) := \arg\max_{t_j \notin \mathcal{T}_K} X(t_j)$. For such $K$, there exists $N_K$, such that when $n > N_K$, $\mathbb{P}\left[|X_n(t_k) - X(t_k)| > \min\{\varepsilon/3, \delta_0/2\}\right] < \frac{\varepsilon}{2K}$ for any $k \leq K$. Therefore, for any $\varepsilon > 0$, there exists $K > K_0$ such that $\min_{k \leq K} |t_k - t_0| < \delta$, and $N_K$ such that for any $n > N_K$, we have:

$$
\begin{aligned}
\mathbb{P}(|\max_{k \leq K} X_n(t_k) - X(t_0)| > \varepsilon) &\leq \mathbb{P}(|\max_{k \leq K} X_n(t_k) - X_n(t_{k_0})| > \varepsilon/3) \\
&+ \mathbb{P}(|X_n(t_{k_0}) - X(t_{k_0})| > \varepsilon/3) + \mathbb{P}(|X(t_{k_0}) - X(t_0)| > \varepsilon/3) \\
&\leq \varepsilon + \mathbb{P}(|\max_{k \leq K} X_n(t_k) - X_n(t_{k_0})| > \varepsilon/3) + \mathbb{P}(|X(t_{k_0}) - X(t_0)| > \varepsilon/3).
\end{aligned}
$$

For $\mathbb{P}(|\max_{k \leq K} X_n(t_k) - X_n(t_{k_0})| > \varepsilon/3)$, we have:

$$
\begin{aligned}
\mathbb{P}(|\max_{k \leq K} X_n(t_k) - X_n(t_{k_0})| > \varepsilon/3) &\leq \mathbb{P}(\max_{k \leq K} X_n(t_k) \neq X_n(t_{k_0})) \\
&\leq \mathbb{P}\{\exists t_j \notin \mathcal{T}_K, \max_{k \leq K} X_n(t_k) = X_n(t_j)\} \\
&\leq \sum_{j \leq K} \mathbb{P}\{\max_{k \leq K} X_n(t_k) = X_n(t_j)\} \\
&= \sum_{j \leq K} \mathbb{P}\{X_n(t_j) - X(t_j) + X(t_j) - X(t_{k_0}) + X(t_{k_0}) - X_n(t_{k_0}) > 0\} \\
&\leq \sum_{j \leq K} \mathbb{P}\{X_n(t_j) - X(t_j) + X(t_{k_0}) - X_n(t_{k_0}) > \delta_0\} \\
&\leq \sum_{j \leq K} [\mathbb{P}\{|X_n(t_j) - X(t_j)| > \delta_0/2\} + \mathbb{P}\{|X_n(t_{k_0}) - X(t_{k_0})| > \delta_0/2\}] \\
&\leq \sum_{j \leq K} \left(\frac{\varepsilon}{2K} + \frac{\varepsilon}{2K}\right) = \varepsilon.
\end{aligned}
$$

Denote $k' := \arg\min_{k \leq K} |t_k - t_0|$. Then for $\mathbb{P}(|X(t_{k_0}) - X(t_0)| > \varepsilon/3)$, we have:

$$
\begin{aligned}
\mathbb{P}(|X(t_{k_0}) - X(t_0)| > \varepsilon/3) &= \mathbb{P}\{X(t_0) - X(t_{k_0}) > \varepsilon/3\} \\
&= \mathbb{P}\{X(t_0) - X(t_{k'}) + X(t_{k'}) - X(t_{k_0}) > \varepsilon/3\} \\
&\leq \mathbb{P}\{X(t_0) - X(t_{k'}) > \varepsilon/3\} \leq \varepsilon.
\end{aligned}
$$

Combining these results together, we have $\lim_{n \to \infty} \lim_{K \to \infty} \max_{k \leq K} X_n(t_k) =_d \max_{t \in \mathcal{T}} X(t)$.

**(ii). Bootstrapped statistics** (8) **is also weakly convergent to the** $\max_{t \in \mathcal{T}} \int |\mathbb{G}(s,t)|^2 d\mu(s)$**.**

By Theorem 2.9.2 of (Wellner et al., 2013), $\widehat{T}_n^b(s,t) = \frac{1}{\sqrt{n}} \sum_{i=1}^n \omega_i^b \widehat{U}(w_i, y_i, t) m(x_i, s)$ is weakly convergent to $\mathbb{G}(s,t)$ conditional the original sample. Applying the continuous mapping theorem, $\int |\widehat{T}_n^b(s,t)|^2 d\mu(s)$ is weakly convergent to $\int |\mathbb{G}(s,t)|^2 d\mu(s)$. Using the proof in **(i)** again, we can obtain that $\widehat{\Delta}_{\varphi,m}^b = \max_{k \in [K]} \int_{\mathcal{T}} |\widehat{T}_n^b(s,t_k)|^2 d\mu(s)$ is weakly convergent to $\max_{t \in \mathcal{T}} \int |\mathbb{G}(s,t)|^2 d\mu(s)$, conditional the original sample. □

### D.5. Auxiliary lemmas

We provide some auxiliary lemmas needed for the theoretical analysis above.

**Lemma D.10** (Lemma 2.5 of (Beyhum et al., 2024))**.** *Let* $(\mathcal{W}, \|\cdot\|_{\mathcal{W}})$ *and* $(\mathcal{X}, \|\cdot\|_{\mathcal{X}})$ *be two Hilbert spaces and* $A : \mathcal{W} \mapsto \mathcal{X}$ *be a linear compact operator with singular value decomposition given by* $(\lambda_n, \varphi_n, \phi_n)_{n=1}^{+\infty}$, $\|\cdot\|_{\text{op}}^2$ *be operator norm. Let* $I : \mathcal{W} \mapsto \mathcal{W}$ *be the identity operator. For each* $\lambda > 0$, *we have the following results:*

*(a)*
$$\|A(\lambda I + A^*A)^{-1}A^*\|_{\mathrm{op}} \le 1.$$

*(b)*
$$\|\lambda(\lambda I + A^*A)^{-1}\|_{\mathrm{op}} \le 2.$$

*(c)*
$$\|(\lambda I + A^*A)^{-1}A^*\|_{\mathrm{op}} = \|A(\lambda I + A^*A)^{-1}\|_{\mathrm{op}} \le \frac{1}{2\sqrt{\lambda}}.$$

*(d) For any $\gamma > 0$ and $g \in \mathcal{W}$ such that $\|g\|_\gamma^2 := \sum_j \lambda_j^{-2\gamma}|\langle g, \varphi_j\rangle|^2 < \infty$, there holds:*

$$\|\lambda(\lambda I + A^*A)^{-1}g\|_{\mathcal{W}} = O\left\{\lambda^{\frac{\min(\gamma,2)}{2}}\right\}.$$

**Lemma D.11.** *If $H^0(w,t)$ is the least norm solution to the linear inverse problem and satisfies assumption D.2 (b), then the solution to the Tikhonov regularization $H^\lambda(w,t)$ satisfies that:*

$$\|H^\lambda(w,t) - H^0(w,t)\|_{\mathcal{H}_W}^2 \le O_p\{\lambda^{\min(\theta,2)}\}.$$

*Proof.* For the operator $A : \mathcal{H}_W \to \mathcal{H}_X$ defined in (24), its singular value decomposition given by $(\lambda_n, \varphi_n, \phi_n)_{n=1}^{+\infty}$. Thus, we have $H_t^0 = \sum_j \langle H_t^0, \varphi_j\rangle_{\mathcal{H}_W} \varphi_j$. Besides, according to $A\varphi_n = \lambda_n \phi_n$ and $A^*\phi_n = \lambda_n \varphi_n$, we have $H_t^\lambda = (A^*A + \lambda I)^{-1}A^*f = \sum_j \frac{\lambda_j^2}{\lambda_j^2 + \lambda}\langle H_t^0, \varphi_j\rangle_{\mathcal{H}_W} \varphi_j$. Thus, we have

$$
\begin{aligned}
\|H^\lambda(w,t) - H^0(w,t)\|_{\mathcal{H}_W}^2 &= \left\|\sum_j \left(\frac{\lambda_j^2}{\lambda_j^2 + \lambda} - 1\right)\langle H_t^0, \varphi_j\rangle_{\mathcal{H}_W} \varphi_j\right\|_{\mathcal{H}_W}^2 \\
&= \sum_j \left\{\left(\frac{\lambda_j^2}{\lambda_j^2 + \lambda} - 1\right)\langle H_t^0, \varphi_j\rangle_{\mathcal{H}_W}\right\}^2 \\
&= \sum_j \frac{\lambda^2 \lambda_j^{2\theta}}{(\lambda_j^2 + \lambda)^2}\frac{|\langle H_t^0, \varphi_j\rangle_{\mathcal{H}_W}|^2}{\lambda_j^{2\theta}} \\
&\le \sup_j \left(\frac{\lambda\lambda_j^\theta}{\lambda_j^2 + \lambda}\right)^2 \sum_j \frac{|\langle H_t^0, \varphi_j\rangle_{\mathcal{H}_W}|^2}{\lambda_j^{2\theta}}.
\end{aligned}
$$

Applying assumption D.2 (b) for $\theta \ge 2$, and the maximum singular value of the operator equals $\|A\|_{\mathrm{op}} < \infty$, we have

$$\sup_j \left(\frac{\lambda\lambda_j^\theta}{\lambda_j^2 + \lambda}\right)^2 = \lambda^2 \sup_j \left(\frac{\lambda_j^\theta}{\lambda_j^2 + \lambda}\right)^2 \le \lambda^2 \sup_j \lambda_j^{2\theta-4} = O(\lambda^2).$$

For $0 < \theta < 2$, we define $x = \lambda_j^2$ and $f(x) = \frac{\lambda^2 x^\theta}{(x+\lambda)^2}$. Noted that $f(x)$ is maximized (by using the first order condition) at $x = \lambda\theta(2 - \theta)^{-1}$. Thus, the maximum value of $f(x)$ is

$$\frac{x^\theta \lambda^2}{(x + \lambda)^2} \le \lambda^\theta \frac{\theta^\theta(2 - \theta)^{2-\theta}}{4} \le O(\lambda^\theta).$$

The proof is complete. $\qquad\square$

**Lemma D.12** (Lemma 12 of (Mastouri et al., 2021)). *Suppose assumptions C.2 and C.3 hold for constants $c_Y$ and $\kappa$, respectively. Define $\sigma_f^2$ and $\sigma_A^2$ as follows:*

$$\sigma_f^2 := \mathbb{E}\{\|\varphi(Y,t)\phi_X(X)\|^2\}, \quad \sigma_A^2 := \mathbb{E}\{\|\phi_X(X)\|^2\|\phi_W(W)\|^2\}.$$

*For $A$, $f$ defined in Eq.(24), and $A^*$ in (28), the estimates $\widehat{A}$, $\widehat{f}$ given by (26) satisfy the following properties with probability at least $1 - \delta$:*

$$\|\widehat{b}_t - b_t\|_{\mathcal{H}_X} \leq \frac{2c_Y \kappa^3 \log(2/\delta)}{n} + \sqrt{\frac{2\sigma_f^2 \log(2/\delta)}{n}} = O_p\left(\frac{1}{\sqrt{n}}\right)$$

$$\|\widehat{A} - A\|_{\mathrm{op}} \leq \frac{2\kappa^6 \log(2/\delta)}{n} + \sqrt{\frac{2\sigma_A^2 \log(2/\delta)}{n}} = O_p\left(\frac{1}{\sqrt{n}}\right)$$

$$\|\widehat{A}^* - A^*\|_{\mathrm{op}} \leq \frac{2\kappa^6 \log(2/\delta)}{n} + \sqrt{\frac{2\sigma_A^2 \log(2/\delta)}{n}} = O_p\left(\frac{1}{\sqrt{n}}\right).$$

**Lemma D.13** (Lemma 13 of (Mastouri et al., 2021))**.** *Suppose assumptions C.2 and C.3 hold. For $A$, $A^*$ defined respectively in (24) and (28), the estimates $\widehat{A}$ given by (26) satisfies:*

$$\|\widehat{A}^*\widehat{A} - A^*A\|_{\mathrm{op}} = O_p\left(\frac{1}{\sqrt{n}}\right).$$

**Lemma D.14** (Lemma 2.4 of (Beyhum et al., 2024))**.** *For random variables $X, W$, let $m(\cdot)$ be the function such that $\mathbb{E}[m(X)|W]$ is bounded. Besides, we denote $\mathcal{F}$ as a class of functions of $W$ such that $\int_0^1 \sqrt{N_{[]}(\epsilon, \mathcal{F}, \|\cdot\|_{\mathcal{L}^2\{F(w)\}})}d\epsilon < \infty$. If $\|(\widehat{f} - f_0)m\|_{\mathcal{L}^2\{F(x,w)\}} = o_p(1)$ and $\mathbb{P}(\widehat{f} \in \mathcal{F}) \to 1$, then*

$$\sqrt{n}(\mathbb{P}_n - \mathbb{P})\{(\widehat{f} - f_0)m\} = o_p(1).$$

**Lemma D.15** (Lemma 2.1 of (Li et al., 2003))**.** *Let $Z_1(\cdot), \cdots, Z_n(\cdot)$ be independent and identically distributed zero mean random elements on $\mathcal{L}^2(\mathcal{S}, \nu)$ such that $\mathbb{E}\{\|Z_i(\cdot)\|_{\mathcal{L}^2(\mathcal{S},\nu)}^2\} := \mathbb{E}\left\{\int_s Z_i^2(s)d\nu(s)\right\}$. Here, $\mathcal{L}^2(\mathcal{S}, \nu)$ is square integrable function space with respect to the measure $\nu$. Then $n^{-1/2}\sum_{i=1}^n Z_i(\cdot)$ converges weakly to a zero mean Gaussian process with the covariance function given by $\Omega(s, s') = \mathbb{E}\{Z_i(s)Z_i(s')\}$.*

**Lemma D.16.** *For operators $A$ and $\widehat{A}$ and their adjoint $A^*$ and $\widehat{A}^*$, we have the following transformation:*

$$(\lambda I + \widehat{A}^*\widehat{A})^{-1} - (\lambda I + A^*A)^{-1} = (\lambda I + A^*A)^{-1}(A^*A - \widehat{A}^*\widehat{A})(\lambda I + \widehat{A}^*\widehat{A})^{-1}.$$

*Proof.*

$$(\lambda I + \widehat{A}^*\widehat{A})^{-1} - (\lambda I + A^*A)^{-1} = I \cdot (\lambda I + \widehat{A}^*\widehat{A})^{-1} - (\lambda I + A^*A)^{-1} \cdot I$$

$$= (\lambda I + A^*A)^{-1}(\lambda I + A^*A)(\lambda I + \widehat{A}^*\widehat{A})^{-1} - (\lambda I + A^*A)^{-1}(\lambda I + \widehat{A}^*\widehat{A})(\lambda I + \widehat{A}^*\widehat{A})^{-1}$$

$$= (\lambda I + A^*A)^{-1}\{(\lambda I + A^*A) - (\lambda I + \widehat{A}^*\widehat{A})\}(\lambda I + \widehat{A}^*\widehat{A})^{-1}$$

$$= (\lambda I + A^*A)^{-1}(A^*A - \widehat{A}^*\widehat{A})(\lambda I + \widehat{A}^*\widehat{A})^{-1}.$$

$\square$

**Lemma D.17.** *Suppose that assumptions C.2, C.3, and D.2 hold. The PMCR estimator $\widehat{H}^\lambda(w, t)$ satisfies*

$$\|\widehat{H}^\lambda(w, t) - H^0(w, t)\|_{\mathcal{H}_W} = O_p\left\{\frac{1}{\sqrt{n\lambda}} + \frac{1}{n\lambda} + \lambda^{\frac{\min(\theta, 2)}{2}}\right\}.$$

*In particular, if assumption 4.10 holds, we have $\|\widehat{H}^\lambda(w, t) - H^0(w, t)\|_{\mathcal{H}_W} = o_p(1)$.*

*Proof.* We first decompose the estimation bias into two parts:

$$\|\widehat{H}^\lambda(w, t) - H^0(w, t)\|_{\mathcal{H}_W} \leq \|\widehat{H}^\lambda(w, t) - H^\lambda(w, t)\|_{\mathcal{H}_W} + \|H^\lambda(w, t) - H^0(w, t)\|_{\mathcal{H}_W}.$$

We first consider $\|\widehat{H}^\lambda(w, t) - H^\lambda(w, t)\|_{\mathcal{H}_W}$. In fact, following the decomposition (37), we have

$$\widehat{H}^\lambda(w, t) - H^\lambda(w, t) = G_1 + G_2 + G_3 + G_4,$$

where $G_1, G_2, G_3, G_4$ are defined in (38)-(41). For $G_1$, we can apply Lemma D.10 (c) to have $\|(\lambda I + A^*A)^{-1}A^*\|_{\mathrm{op}} = O_p(1/\sqrt{\lambda})$. Besides, according to (43), we have $\|\widehat{b}_t - \widehat{A}H_t^0\|_{\mathcal{H}_W} = O_p(1/\sqrt{n})$. Combining these together, we get

$$\|G_1\|_{\mathcal{H}_W} \leq \|(\lambda I + A^*A)^{-1}A^*\|_{\mathrm{op}} \cdot \|\widehat{b}_t - \widehat{A}H_t^0\|_{\mathcal{H}_W} = O_p\left(\frac{1}{\sqrt{n\lambda}}\right).$$

For $G_2$, we apply Lemma D.10 (b) to obtain that $\|(\lambda I + A^*A)^{-1}\|_{\mathrm{op}} = O_p(1/\lambda)$. Besides, according to (43) and Lemma D.12, we have $\|\widehat{b}_t - \widehat{A}H_t^0\|_{\mathcal{H}_W} = O_p(1/\sqrt{n})$ and $\|\widehat{A}^* - A^*\|_{\mathrm{op}} = O_p(1/\sqrt{n})$. Combining these inequalities together, we have:

$$\|G_2\|_{\mathcal{H}_W} \leq \|(\lambda I + A^*A)^{-1}\|_{\mathrm{op}} \cdot \|\widehat{A}^* - A^*\|_{\mathrm{op}} \cdot \|\widehat{b}_t - \widehat{A}H_t^0\|_{\mathcal{H}_W} = O_p\left(\frac{1}{n\lambda}\right).$$

For $G_3$, we have:

$$\begin{aligned}
\|G_3\|_{\mathcal{H}_W} &\leq \|\{(\lambda I + \widehat{A}^*\widehat{A})^{-1} - (\lambda I + A^*A)^{-1}\}\widehat{A}^*\|_{\mathrm{op}} \cdot \|\widehat{b}_t - \widehat{A}H_t^0\|_{\mathcal{H}_W} \\
&= \|(\lambda I + \widehat{A}^*\widehat{A})^{-1}\widehat{A}^* - (\lambda I + A^*A)^{-1}A^* - (\lambda I + A^*A)^{-1}(\widehat{A}^* - A^*)\|_{\mathrm{op}} \cdot \|\widehat{b}_t - \widehat{A}H_t^0\|_{\mathcal{H}_W} \\
&\leq \|(\lambda I + \widehat{A}^*\widehat{A})^{-1}\widehat{A}^* - (\lambda I + A^*A)^{-1}A^*\|_{\mathrm{op}} \cdot \|\widehat{b}_t - \widehat{A}H_t^0\|_{\mathcal{H}_W} \\
&\quad + \|(\lambda I + A^*A)^{-1}\|_{\mathrm{op}} \cdot \|\widehat{A}^* - A^*\|_{\mathrm{op}} \cdot \|\widehat{b}_t - \widehat{A}H_t^0\|_{\mathcal{H}_W}.
\end{aligned}$$

Since $\widehat{A}$ and $A$ are compact operators, we can apply Lemma D.10 (b), (c) to obtain that $\|(\lambda I + \widehat{A}^*\widehat{A})^{-1}\widehat{A}^* - (\lambda I + A^*A)^{-1}A^*\|_{\mathrm{op}} = O_p(1/\lambda)$ and $\|(\lambda I + A^*A)^{-1}\|_{\mathrm{op}} = O_p(1/\lambda)$. Besides, according to (43) and Lemma D.12, we have $\|\widehat{b}_t - \widehat{A}H_t^0\|_{\mathcal{H}_W} = O_p(1/\sqrt{n})$ and $\|\widehat{A}^* - A^*\|_{\mathrm{op}} = O_p(1/\sqrt{n})$. Combining all the inequalities, we get

$$\|G_3\|_{\mathcal{H}_W} = O_p\left(\frac{1}{\sqrt{n\lambda}}\right) + O_p\left(\frac{1}{n\lambda}\right).$$

For $G_4$, we have:

$$\begin{aligned}
\|G_4\|_{\mathcal{H}_W} &= \|(\lambda I + \widehat{A}^*\widehat{A})^{-1}\widehat{A}^*\widehat{A}H_t^0 - (\lambda I + A^*A)^{-1}A^*AH_t^0\|_{\mathcal{H}_W} \\
&\overset{(1)}{=} \|\lambda(\lambda I + \widehat{A}^*\widehat{A})^{-1}\{\widehat{A}^*\widehat{A} - A^*A\}(\lambda I + A^*A)^{-1}H_t^0\|_{\mathcal{H}_W} \\
&= \|\lambda(\lambda I + \widehat{A}^*\widehat{A})^{-1}\{\widehat{A}^*(\widehat{A} - A) + (\widehat{A}^* - A^*)A\}(\lambda I + A^*A)^{-1}H_t^0\|_{\mathcal{H}_W} \\
&\leq \|\lambda(\lambda I + \widehat{A}^*\widehat{A})^{-1}\widehat{A}^*(\widehat{A} - A)(\lambda I + A^*A)^{-1}H_t^0\|_{\mathcal{H}_W} \\
&\quad + \|\lambda(\lambda I + \widehat{A}^*\widehat{A})^{-1}(\widehat{A}^* - A^*)A(\lambda I + A^*A)^{-1}H_t^0\|_{\mathcal{H}_W} \\
&\leq \|(\lambda I + \widehat{A}^*\widehat{A})^{-1}\widehat{A}^*\|_{\mathrm{op}} \cdot \|\widehat{A} - A\|_{\mathrm{op}} \cdot \|\lambda(\lambda I + A^*A)^{-1}\|_{\mathrm{op}} \cdot \|H_t^0\|_{\mathcal{H}_W} \\
&\quad + \|\lambda(\lambda I + \widehat{A}^*\widehat{A})^{-1}\|_{\mathrm{op}} \cdot \|\widehat{A}^* - A^*\|_{\mathrm{op}} \cdot \|A(\lambda I + A^*A)^{-1}\|_{\mathrm{op}} \cdot \|H_t^0\|_{\mathcal{H}_W},
\end{aligned}$$

where (1) follows from:

$$\begin{aligned}
&(\lambda I + \widehat{A}^*\widehat{A})^{-1}\widehat{A}^*\widehat{A}H_t^0 - (\lambda I + A^*A)^{-1}A^*AH_t^0 \\
&= \left[(\lambda I + \widehat{A}^*\widehat{A})^{-1}\left\{(\lambda I + \widehat{A}^*\widehat{A}) - \lambda I\right\} - (\lambda I + A^*A)^{-1}\left\{(\lambda I + A^*A) - \lambda I\right\}\right]H_t^0 \\
&= \lambda\left\{(\lambda I + A^*A)^{-1} - (\lambda I + \widehat{A}^*\widehat{A})^{-1}\right\}H_t^0 \\
&= \lambda(\lambda I + \widehat{A}^*\widehat{A})^{-1}\{\widehat{A}^*\widehat{A} - A^*A\}(\lambda I + A^*A)^{-1}H_t^0.
\end{aligned}$$

Since $\widehat{A}$ and $A$ are compact operators, we can apply Lemma D.10 (b), (c) to obtain that $\|(\lambda I + \widehat{A}^*\widehat{A})^{-1}\widehat{A}^*\|_{\mathrm{op}}O_p(1/\sqrt{\lambda})$, $\|(\lambda I + A^*A)^{-1}A^*\|_{\mathrm{op}} = O_p(1/\sqrt{\lambda})$, $\|\lambda(\lambda I + A^*A)^{-1}\|_{\mathrm{op}} \leq 2$, $\|\lambda(\lambda I + \widehat{A}^*\widehat{A})^{-1}\|_{\mathrm{op}} \leq 2$. Besides, according to Lemma D.12, we have $\|\widehat{A}^* - A^*\|_{\mathrm{op}} = \|\widehat{A} - A\|_{\mathrm{op}} = O_p(1/\sqrt{n})$. Combining all the inequalities, we get:

$$\|G_4\|_{\mathcal{H}_W} = O_p\left(\frac{1}{\sqrt{n\lambda}}\right).$$

Combining these results for $G_1$ to $G_4$, we have

$$\|\widehat{H}^\lambda(w,t) - H^\lambda(w,t)\|_{\mathcal{H}_W} = O_p\left(\frac{1}{\sqrt{n\lambda}} + \frac{1}{n\lambda}\right).$$

Next, we consider $\|H^\lambda(w,t) - H^0(w,t)\|_{\mathcal{H}_W}$. By assumption D.2, we can employ Lemma D.11 to obtain that:

$$\|H^\lambda(w,t) - H^0(w,t)\|_{\mathcal{H}_W} = O_p\left(\lambda^{\frac{\min(\theta,2)}{2}}\right).$$

Thus, we have

$$\|\widehat{H}^\lambda(w,t) - H^0(w,t)\|_{\mathcal{H}_W} = O_p\left\{\frac{1}{\sqrt{n\lambda}} + \frac{1}{n\lambda} + \lambda^{\frac{\min(\theta,2)}{2}}\right\}.$$

By assumption 4.10, we have $n\lambda \to \infty$ and $\lambda \to 0$, which gives $\|\widehat{H}^\lambda(w,t) - H^0(w,t)\|_{\mathcal{H}_W} = o_p(1)$. $\qquad\square$

# E. Existence of solutions with two proxies

### E.1. Proof of Theorem 6.3

**Theorem 6.3.** *Suppose assumptions 6.1, 6.2 hold. For any $h(w, y)$ that satisfies (1), $\mathbb{H}_0$ holds if and only if $h(w, y)$ also satisfies the integral equation (10) for any fixed $x$.*

*Proof.* Suppose $h(w, y)$ satisfies $p(y|x) = \int h(w, y)p(w|x)dw$. Under $\mathbb{H}_0$, we have $X \perp (W, Y)|U$, which leads to:

$$\int p(y|u)p(u|x)du = p(y|x) = \int h(w, y)p(w|x)dw$$
$$= \int \left\{ \int h(w, y)p(w|u)dw \right\} p(u|x)du.$$

By the completeness in assumption 6.1, $h(w, y)$ solves the following integral equation for all $(u, y)$.

$$p(y|u) = \int h(w, y)p(w|u)dw.$$

Since $\mathbb{H}_0$ holds, we have $Y \perp (Z, X)|U$. Therefore, for any fixed $x$, taking expectation over $p(u|z, x)$ on both sides, we have:

$$p(y|z, x) = \int p(y|u)p(u|z, x)du = \int \left\{ \int h(w, y)p(w|u)dw \right\} p(u|z, x)du \overset{(1)}{=} \int h(w, y)p(w|z, x)dw,$$

where "(1)" is due to $W \perp (Z, X)|U$. That means, $h(w, y)$ solves the integral equation (10). To prove the contrary, note that if $h(w, y)$ (1) satisfies (10), by $W \perp (Z, X)|U$ and $Y \perp Z|(U, X)$, we have

$$\int p(y|u, x)p(u|z, x)du = p(y|z, x)$$
$$= \int h(w, y)p(w|z, x)dw$$
$$= \int \left\{ \int h(w, y)p(w|u)dw \right\} p(u|z, x)du.$$

By the completeness condition in assumption 6.2, we obtain

$$p(y|u, x) = \int h(w, y)p(w|u)dw.$$

Since the right side of the equation is independent of $x$, we get $p(y|u, x) = p(y|u)$, and thus $\mathbb{H}^0$ holds. $\square$

### E.2. Discussions of causal inference and causal discovery

In this section, we explore the distinction between causal discovery and causal inference, focusing on why the causal relation cannot be identified solely through the causal effect. We begin by presenting a counter-example that demonstrates that even when the intervention distribution for each $x$ is identical, the independence $X \perp Y|U$ may still fail to hold. Following this, we provide an in-depth discussion of the differences between causal inference and causal discovery.

We first introduce the notations. For any discrete variables $X, Y, Z$ with $k$ categories, we denote $\mathbb{P}(X) := \{p(x_1), ..., p(x_k)\}^\top$, $\mathbb{P}(Y|X) = \{p(y_i|x_j)\}_{i,j}$, and $\mathbb{P}(Y = y|X, Z) = \{p(y|x_i, z_j)\}_{i,j}$.

**Example E.1.** Suppose $U, X, Y$ are binary, and the causal diagram over $(U, X, Y)$ is $U \to X, U \to Y, X \to Y$. The conditional probability matrices $\mathbb{P}(U), \mathbb{P}(X|U), \mathbb{P}(Y|X, U)$ are given by:

$$\mathbb{P}(U) = \begin{pmatrix} 0.4 \\ 0.6 \end{pmatrix}, \ \mathbb{P}(X|U) = \begin{pmatrix} 0.2 & 0.4 \\ 0.8 & 0.6 \end{pmatrix}, \ \mathbb{P}(Y = 0|X, U) = \begin{pmatrix} 0.5 & 0.1 \\ 0.2 & 0.3 \end{pmatrix}.$$

By the definition, we know $X \not\perp Y|U$. However, the intervention distribution is the same, *i.e.*, $\mathbb{P}\{y|do(X = 0)\} = \mathbb{P}\{y|do(X = 1)\}$ for any $y$.

*Proof.* According to the backdoor formula, we have

$$\mathbb{P}\{Y = y|do(X = x)\} = \sum_{u \in \{0,1\}} \mathbb{P}(Y = y|U = u, X = x)\mathbb{P}(U = u).$$

Plugging $\mathbb{P}(Y = 0|X, U)$ into the formula, we have:

$$\mathbb{P}\{Y = 0|do(X = 0)\} = 0.5 \times 0.4 + 0.1 \times 0.6 = 0.26$$
$$\mathbb{P}\{Y = 0|do(X = 1)\} = 0.2 \times 0.4 + 0.3 \times 0.6 = 0.26$$
$$\mathbb{P}\{Y = 1|do(X = 0)\} = 0.5 \times 0.4 + 0.9 \times 0.6 = 0.74$$
$$\mathbb{P}\{Y = 1|do(X = 1)\} = 0.8 \times 0.4 + 0.7 \times 0.6 = 0.74,$$

which implies intervention distributions are equal. However, through data generation, we know $X \not\perp Y|U$. $\qquad\square$

Next, we will verify that in this example, $\mathbb{P}(Y = y|X = x) \neq \sum_u \mathbb{P}(Y = y|U = u)\mathbb{P}(U = u|X = x)$, which implies the example contradicts our assumption that there is no solution in (1) under $\mathbb{H}_1$. To this end, we need to obtain probability matrix $\mathbb{P}(Y|X), \mathbb{P}(Y|U)$, and $\mathbb{P}(U|X)$. First, by $\mathbb{P}(U)$ and $\mathbb{P}(X|U)$, we can get the probability matrix $\mathbb{P}(X)$ and $\mathbb{P}(U|X)$.

$$\mathbb{P}(X) = \mathbb{P}(X|U)\mathbb{P}(U) = \begin{pmatrix} 0.2 & 0.4 \\ 0.8 & 0.6 \end{pmatrix}\begin{pmatrix} 0.4 \\ 0.6 \end{pmatrix} = \begin{pmatrix} 0.32 \\ 0.68 \end{pmatrix}, \mathbb{P}(U|X) = \begin{pmatrix} 0.25 & 8/17 \\ 0.75 & 9/17 \end{pmatrix}.$$

Besides, we calculate the probability of $\mathbb{P}(y|x)$ for any $y, x$. According to the Bayesian formula, we have

$$\mathbb{P}(Y = y|X = x) = \sum_u \mathbb{P}(Y = y|X = x, U = u)\mathbb{P}(U = u|X = x)$$
$$= \sum_u \mathbb{P}(Y = y|X = x, U = u)\frac{\mathbb{P}(X = x|U = u)\mathbb{P}(U = u)}{\mathbb{P}(X = x)}.$$

Therefore, we have

$$\mathbb{P}(Y|X) = \begin{pmatrix} 0.2 & 43/170 \\ 0.8 & 127/170 \end{pmatrix}.$$

According to the Bayesian formula, we have

$$\mathbb{P}(Y = y|U = u) = \sum_x \mathbb{P}(Y = y|X = x, U = u)\mathbb{P}(X = x|U = u).$$

Therefore, we have

$$\mathbb{P}(Y|U) = \begin{pmatrix} 0.26 & 0.22 \\ 0.74 & 0.78 \end{pmatrix}.$$

Thus, we can verify

$$\mathbb{P}(Y = 0|X = 0) = 0.2 \neq 0.23 = 0.26 \times 0.25 + 0.22 \times 0.75 = \sum_u \mathbb{P}(Y = 0|U = u)\mathbb{P}(U = u|X = 0)$$

$$\mathbb{P}(Y = 0|X = 1) = \frac{43}{170} \neq \frac{203}{850} = 0.26 \times \frac{8}{17} + 0.22 \times \frac{9}{17} = \sum_u \mathbb{P}(Y = 0|U = u)\mathbb{P}(U = u|X = 1)$$

$$\mathbb{P}(Y = 1|X = 0) = 0.8 \neq 0.64 = 0.22 \times 0.25 + 0.78 \times 0.75 = \sum_u \mathbb{P}(Y = 1|U = u)\mathbb{P}(U = u|X = 0)$$

$$\mathbb{P}(Y = 1|X = 1) = \frac{127}{170} \neq \frac{439}{850} = 0.22 \times \frac{8}{17} + 0.78 \times \frac{9}{17} = \sum_u \mathbb{P}(Y = 1|U = u)\mathbb{P}(U = u|X = 1).$$

**More discussions about casual discovery and causal inference.** Causal inference and causal discovery address fundamentally different problems (Guo et al., 2020). Causal inference focuses on quantifying the effects of interventions, often

requiring strong assumptions and additional information to ensure accurate estimation. In contrast, causal discovery aims to uncover the underlying causal structure, emphasizing the identification of causal relationships rather than their magnitudes.

It may not be feasible to infer the causal relationship from the causal effect. One key reason is that the inference is often complicated by noise in the estimates, making it hard to determine whether a nonzero effect arises from an actual causal relationship or random noise perturbing the estimation. Even if we can estimate a confidence interval for the effect at each treatment value (Robins, 1988; Robins et al., 2003; Calonico et al., 2018; Colangelo & Lee, 2020), there are no valid statistics to determine whether the relation exists. Moreover, as shown in the previous example, a causal effect of zero does not necessarily imply the existence of the causal relation. Additionally, estimating causal effects often requires satisfying other conditions. For example, proximal causal inference depends on additional completeness assumptions (Miao et al., 2018; Mastouri et al., 2021). In our scenario, such conditions are assumed on $Z|X, W$ (*i.e.*, for any square-integrable function $g$, $\mathbb{E}\{g(z)|x, w\} = 0$ almost surely if and only if $g(z) = 0$ almost surely) and $\{X, W\}|\{X, Z\}$ (Mastouri et al., 2021).

### E.3. Proof of Proposition 5.1 and example 5.3

We first prove Proposition 5.1.

**Proposition E.2.** *We consider the linear Gaussian generation mechanism:*

$$\begin{cases} U \sim \mathcal{N}(0, 1) \\ X = \alpha_0 + \alpha_U U + \mathcal{N}(0, 1) \\ W = \beta_0 + \beta_U U + \mathcal{N}(0, 1) \\ Y = \gamma_0 + \gamma_U U + \gamma_X X + \gamma_W W + \mathcal{N}(0, 1). \end{cases}$$

*When $\gamma_W = 0$, as long as $|\gamma_X| > \frac{|B| + \sqrt{\Delta}}{2A}$, where $A = 1 + \frac{1}{\alpha_U^2} + \frac{2}{\beta_U^2} + \frac{1}{\alpha_U^2 \beta_U^2} + \frac{\alpha_U^2}{\beta_U^2}$, $B = \frac{2\gamma_U}{\alpha_U} + \frac{2\gamma_U}{\alpha_U \beta_U^2} + \frac{2\alpha_U \gamma_U}{\beta_U^2}$ and $\Delta = \frac{4(1+\alpha_U^2+\beta_U^2)(1+\alpha_U^2+\gamma_U^2)}{\alpha_U^2 \beta_U^2}$, the integration equation (1) has no solution. Further, if $|\gamma_W| > \frac{|C| + |B||\gamma_X| + A\gamma_X^2}{2|D|}$, where $C = 1 - \gamma_U^2/\beta_U^2$ and $D = \frac{\gamma_X}{\alpha_U \beta_U} + \frac{\alpha_U}{\beta_U} \gamma_X + \frac{\beta_U}{\alpha_U} \gamma_X + \frac{\gamma_U}{\beta_U}$, (1) has a solution.*

*Proof.* Based on the data generation structure, we can obtain joint distribution

$$\begin{pmatrix} U \\ X \\ W \\ Y \end{pmatrix} \sim \mathcal{N} \left\{ \begin{pmatrix} 0 \\ \alpha_0 \\ \beta_0 \\ \gamma_0 + \gamma_X \alpha_0 + \gamma_W \beta_0 \end{pmatrix}, \begin{pmatrix} 1 & \alpha_U & \beta_U & \text{Cov}(U, Y) \\ \alpha_U & 1 + \alpha_U^2 & \alpha_U \beta_U & \text{Cov}(X, Y) \\ \beta_U & \alpha_U \beta_U & 1 + \beta_U^2 & \text{Cov}(W, Y) \\ \text{Cov}(U, Y) & \text{Cov}(X, Y) & \text{Cov}(W, Y) & \text{Var}(Y) \end{pmatrix} \right\},$$

where covariance $\text{Cov}(U, Y), \text{Cov}(X, Y), \text{Cov}(W, Y)$ and $\text{Var}(Y)$ are respectively

$$\begin{cases} \text{Cov}(U, Y) = \gamma_U + \gamma_X \alpha_U + \gamma_W \beta_U \\ \text{Cov}(X, Y) = \alpha_U (\gamma_U + \gamma_W \beta_U + \gamma_X \alpha_U) + \gamma_X \\ \text{Cov}(W, Y) = \beta_U (\gamma_U + \alpha_U \gamma_X + \gamma_W \beta_U) + \gamma_W \\ \text{Var}(Y) = (\gamma_U + \gamma_X \alpha_U + \gamma_W \beta_U)^2 + \gamma_X^2 + \gamma_W^2 + 1. \end{cases}$$

We can therefore derive the explicit form of the conditional distributions $p(w|x)$ and $p(y|x)$:

$$W|X = x \sim \mathcal{N} \left\{ \mu_W + \frac{\text{Cov}(W, X)}{\text{Var}(X)} (x - \mu_X), \text{Var}(W) \left( 1 - \frac{\text{Cov}^2(W, X)}{\text{Var}(X) \cdot \text{Var}(W)} \right) \right\}$$

$$\sim \mathcal{N} \left\{ \mu_X^{W|X} x + \mu_0^{W|X}, \sigma_{W|X}^2 \right\}$$

$$Y|X = x \sim \mathcal{N} \left\{ \mu_Y + \frac{\text{Cov}(Y, X)}{\text{Var}(X)} (x - \mu_X), \text{Var}(Y) \left( 1 - \frac{\text{Cov}^2(Y, X)}{\text{Var}(X) \cdot \text{Var}(Y)} \right) \right\}$$

$$\sim \mathcal{N} \left\{ \mu_X^{Y|X} x + \mu_0^{Y|X}, \sigma_{Y|X}^2 \right\},$$

where $\mu_X^{W|X}, \mu_0^{W|X}, \sigma_{W|X}^2, \mu_X^{Y|X}, \mu_0^{Y|X}$ and $\sigma_{Y|X}^2$ are defined as follows

$$
\begin{cases}
\mu_X^{W|X} = \frac{\alpha_U \beta_U}{1+\alpha_U^2} \\
\mu_0^{W|X} = \beta_0 - \frac{\alpha_0 \alpha_U \beta_U}{1+\alpha_U^2} \\
\sigma_{W|X}^2 = 1 + \beta_U^2 - \frac{(\alpha_U \beta_U)^2}{1+\alpha_U^2} \\
\mu_X^{Y|X} = \frac{\alpha_U(\gamma_U + \gamma_W \beta_U + \gamma_X \alpha_U) + \gamma_X}{1+\alpha_U^2} \\
\mu_0^{Y|X} = \gamma_0 - \frac{\alpha_0 \alpha_U(\gamma_U + \gamma_W \beta_U + \gamma_X \alpha_U) + \alpha_0 \gamma_X}{1+\alpha_U^2} \\
\sigma_{Y|X}^2 = (\gamma_U + \gamma_X \alpha_U + \gamma_W \beta_U)^2 + \gamma_X^2 + \gamma_W^2 + 1 - \frac{(\alpha_U(\gamma_U + \gamma_W \beta_U + \gamma_X \alpha_U) + \gamma_X)^2}{1+\alpha_U^2}.
\end{cases}
$$

By applying Lemma B.4, the solution of (1) is given by:

$$
h(w,y) = \frac{1}{\sqrt{\sigma_{Y|X}^2 - \left(\mu_X^{Y|X}\right)^2 \sigma_{W|X}^2 / \left(\mu_X^{W|X}\right)^2}} \phi \left( \frac{y - \left(\mu_0^{Y|X} - \mu_X^{Y|X} \mu_0^{W|X}/\mu_X^{W|X}\right) - \mu_X^{Y|X}/\mu_X^{W|X} w}{\sqrt{\sigma_{Y|X}^2 - \left(\mu_X^{Y|X}\right)^2 \sigma_{W|X}^2 / \left(\mu_X^{W|X}\right)^2}} \right),
$$

where $\phi$ is the standard normal distribution's probability density function (pdf).

For $h(w,y)$ to be meaningful, we need $\sigma_{Y|X}^2 - \left(\mu_X^{Y|X}\right)^2 \sigma_{W|X}^2 / \left(\mu_X^{W|X}\right)^2 > 0$, which implies

$$
1 - \frac{\gamma_U^2}{\beta_U^2} - \left(\frac{2\gamma_U}{\alpha_U} + \frac{2\gamma_U}{\alpha_U \beta_U^2} + \frac{2\alpha_U \gamma_U}{\beta_U^2}\right)\gamma_X - \left(1 + \frac{1}{\alpha_U^2} + \frac{2}{\beta_U^2} + \frac{1}{\alpha_U^2 \beta_U^2} + \frac{\alpha_U^2}{\beta_U^2}\right)\gamma_X^2
$$
$$
-2\left(\frac{1}{\alpha_U \beta_U} + \frac{\alpha_U}{\beta_U} + \frac{\beta_U}{\alpha_U}\right)\gamma_X \gamma_W - 2\frac{\gamma_U}{\beta_U}\gamma_W > 0. \tag{54}
$$

We discuss the following two cases: **(i)** $X \to Y$ ($\gamma_X \neq 0$) and $W \not\to Y$ ($\gamma_W = 0$); **(ii)** $X \to Y$ ($\gamma_X \neq 0$) and $W \not\to Y$ ($\gamma_W = 0$).

**1.** $\gamma_X \neq 0, \gamma_W = 0$**.**

We first rewrite (54) as:

$$
\underbrace{1 - \frac{\gamma_U^2}{\beta_U^2}}_{C} - \underbrace{\left(\frac{2\gamma_U}{\alpha_U} + \frac{2\gamma_U}{\alpha_U \beta_U^2} + \frac{2\alpha_U \gamma_U}{\beta_U^2}\right)}_{B}\gamma_X - \underbrace{\left(1 + \frac{1}{\alpha_U^2} + \frac{2}{\beta_U^2} + \frac{1}{\alpha_U^2 \beta_U^2} + \frac{\alpha_U^2}{\beta_U^2}\right)}_{A}\gamma_X^2 > 0.
$$

Noting that this is a quadratic function, we can get its discriminant

$$
\Delta := B^2 - 4AC = \frac{4\left(1 + \alpha_U^2 + \beta_U^2\right)\left(1 + \alpha_U^2 + \gamma_U^2\right)}{\alpha_U^2 \beta_U^2} > 0.
$$

Besides, we can find $1 + \frac{1}{\alpha_U^2} + \frac{2}{\beta_U^2} + \frac{1}{\alpha_U^2 \beta_U^2} + \frac{\alpha_U^2}{\beta_U^2} > 0$. Therefore, this is a quadratic function whose discriminant is always positive and opens downward. When $\gamma_X$ satisfies $\frac{-B+\sqrt{\Delta}}{2A} < \gamma_X < \frac{-B-\sqrt{\Delta}}{2A}$, (1) will have a solution. When $\gamma_X \geq \frac{-B-\sqrt{\Delta}}{2A}$ or $\gamma_X \leq \frac{-B+\sqrt{\Delta}}{2A}$, (1) will have no solution.

Without loss of generality, we consider the case where $\alpha_U$ and $\gamma_U$ have the same sign. First, we can find $B = -(\frac{2\gamma_U}{\alpha_U} + \frac{2\gamma_U}{\alpha_U \beta_U^2} + \frac{2\alpha_U \gamma_U}{\beta_U^2}) < 0$ since $\beta_U^2 > 0$. Thus, we have $|-B - \sqrt{\Delta}| < |-B + \sqrt{\Delta}|$. Thus, when $|\gamma_X| > \frac{-B+\sqrt{\Delta}}{2A}$, (1) will have no solution. If $\alpha_U$ and $\gamma_U$ have the different sign, we have $B = -(\frac{2\gamma_U}{\alpha_U} + \frac{2\gamma_U}{\alpha_U \beta_U^2} + \frac{2\alpha_U \gamma_U}{\beta_U^2}) > 0$ since $\beta_U^2 > 0$. Thus, we have $|-B - \sqrt{\Delta}| > |-B + \sqrt{\Delta}|$. Thus, when $|\gamma_X| > \frac{-B-\sqrt{\Delta}}{2A}$, (1) will have no solution.

Combining the two cases, we can obtain that as long as $|\gamma_X| > \frac{|B|+\sqrt{\Delta}}{2A}$, the integration equation (1) has no solution.

**2.** $\gamma_X \neq 0, \gamma_W \neq 0$.

We consider the case $|\gamma_X| > \frac{-B+\sqrt{\Delta}}{2A}$ under $\alpha_U \gamma_U > 0$, since (1) have no solution. We can rewrite (54) as

$$2\left(\frac{\gamma_X}{\alpha_U \beta_U} + \frac{\alpha_U}{\beta_U}\gamma_X + \frac{\beta_U}{\alpha_U}\gamma_X + \frac{\gamma_U}{\beta_U}\right)\gamma_W < 1 - \frac{\gamma_U^2}{\beta_U^2} - \left(\frac{2\gamma_U}{\alpha_U} + \frac{2\gamma_U}{\alpha_U \beta_U^2} + \frac{2\alpha_U \gamma_U}{\beta_U^2}\right)\gamma_X - \left(1 + \frac{1}{\alpha_U^2} + \frac{2}{\beta_U^2} + \frac{1}{\alpha_U^2 \beta_U^2} + \frac{\alpha_U^2}{\beta_U^2}\right)\gamma_X^2.$$

Thus, if $\frac{\gamma_X}{\alpha_U \beta_U} + \frac{\alpha_U}{\beta_U}\gamma_X + \frac{\beta_U}{\alpha_U}\gamma_X + \frac{\gamma_U}{\beta_U} < 0$, we can obtain that (1) may still have a solution, as long as

$$\gamma_W > \frac{1 - \frac{\gamma_U^2}{\beta_U^2} - \left(\frac{2\gamma_U}{\alpha_U} + \frac{2\gamma_U}{\alpha_U \beta_U^2} + \frac{2\alpha_U \gamma_U}{\beta_U^2}\right)\gamma_X - \left(1 + \frac{1}{\alpha_U^2} + \frac{2}{\beta_U^2} + \frac{1}{\alpha_U^2 \beta_U^2} + \frac{\alpha_U^2}{\beta_U^2}\right)\gamma_X^2}{2\left(\frac{\gamma_X}{\alpha_U \beta_U} + \frac{\alpha_U}{\beta_U}\gamma_X + \frac{\beta_U}{\alpha_U}\gamma_X + \frac{\gamma_U}{\beta_U}\right)}.$$

We find that if $\frac{\gamma_X}{\alpha_U \beta_U} + \frac{\alpha_U}{\beta_U}\gamma_X + \frac{\beta_U}{\alpha_U}\gamma_X + \frac{\gamma_U}{\beta_U} < 0$, the right-hand side of the above inequality is positive. That means, as long as $|\gamma_W|$ is sufficiently large, the solution to (1) will still exist when $|\gamma_X| > \frac{-B+\sqrt{\Delta}}{2A}$.

If $\frac{\gamma_X}{\alpha_U \beta_U} + \frac{\alpha_U}{\beta_U}\gamma_X + \frac{\beta_U}{\alpha_U}\gamma_X + \frac{\gamma_U}{\beta_U} > 0$, we can obtain (1) may still have a solution, as long as

$$\gamma_W < \frac{1 - \frac{\gamma_U^2}{\beta_U^2} - \left(\frac{2\gamma_U}{\alpha_U} + \frac{2\gamma_U}{\alpha_U \beta_U^2} + \frac{2\alpha_U \gamma_U}{\beta_U^2}\right)\gamma_X - \left(1 + \frac{1}{\alpha_U^2} + \frac{2}{\beta_U^2} + \frac{1}{\alpha_U^2 \beta_U^2} + \frac{\alpha_U^2}{\beta_U^2}\right)\gamma_X^2}{2\left(\frac{\gamma_X}{\alpha_U \beta_U} + \frac{\alpha_U}{\beta_U}\gamma_X + \frac{\beta_U}{\alpha_U}\gamma_X + \frac{\gamma_U}{\beta_U}\right)}.$$

We find that if $\frac{\gamma_X}{\alpha_U \beta_U} + \frac{\alpha_U}{\beta_U}\gamma_X + \frac{\beta_U}{\alpha_U}\gamma_X + \frac{\gamma_U}{\beta_U} > 0$, the right-hand side is negative. That also means, as long as $|\gamma_W|$ is sufficiently large, the solution to (1) will still exist when $|\gamma_X| > \frac{-B+\sqrt{\Delta}}{2A}$.

If $\alpha_U \gamma_U < 0$, the proof is similar. Besides, in the above cases, as long as $|\gamma_W| > \frac{|C|+|B||\gamma_X|+A\gamma_X^2}{2|D|}$ with $D := \frac{\gamma_X}{\alpha_U \beta_U} + \frac{\alpha_U}{\beta_U}\gamma_X + \frac{\beta_U}{\alpha_U}\gamma_X + \frac{\gamma_U}{\beta_U}$, (1) has a solution. $\square$

*Remark* E.3. If $\gamma_X = \gamma_W = 0$, (54) will become $1 - \frac{\gamma_U^2}{\beta_U^2} > 0$. This means that if the strength between $W - U$ is greater than the confounder strength between $W - U$, (1) will have a solution under $\mathbb{H}^0$. Otherwise, similar to the case when $\gamma_X \neq 0$, if the effect of $W$ on $Y$ is strong enough (*i.e.*, $|\gamma_W|$), the solution exists again. Specifically, if $\gamma_X = 0, \gamma_W \neq 0$, (54) will become $1 - \frac{\gamma_U^2}{\beta_U^2} - 2\frac{\gamma_U}{\beta_U}\gamma_W > 0$. If $-2\frac{\gamma_U}{\beta_U}\gamma_W$ is large enough, (1) still have a solution. If we $\gamma_U/\beta_U > 0$, we need $\gamma_W$ to be as negative as possible; if $\gamma_U/\beta_U < 0$, we need $\gamma_W$ to be as positive as possible.

Next, we prove the claims in example 5.3. We show that as long as the coefficient of $W \to Y$ is strong enough in example 5.3, the solution of the integral equation $p(y|x) = \int h(w,y)p(w|x)dw$ exists. As an explanation, we will show that a key condition in Picard's theorem B.2 holds, namely, the series $\sum_{n=1}^{\infty} \lambda_n^{-2}|\langle p(y|x), \phi_n\rangle|^2$ converges.

To compute the series, we need the singular value decomposition of the operator $T : \mathcal{L}^2\{F(w)\} \to \mathcal{L}^2\{F(x)\}$, where $Th = \mathbb{E}\{h(W,y)|x\} = p(y|x)$ for all $(x,y)$. Based on the data-generating process in example 5.3, both $\mathcal{L}^2\{F(w)\}$ and $\mathcal{L}^2\{F(x)\}$ are square-integrable spaces with respect to the standard Gaussian measure. For such spaces, (Carrasco et al., 2007) derived the form of the eigenvectors $\phi_n$, as stated in Lemma E.4. As this result builds upon the concept of generalized Hermite polynomials, we include a brief introduction to facilitate understanding.

We first introduce the concept of Hermite polynomial, which is defined in the square-integrable function space with respect to the standard Gaussian measure. Specifically, we say that a function $f : \mathbb{R} \to \mathbb{R}$ is square integrable w.r.t. the standard Gaussian measure $\gamma = e^{-x^2/2}/\sqrt{2\pi}$ if $\mathbb{E}_{x \sim \mathcal{N}(0,1)}\{f^2(x)\} < \infty$. We denote by $\mathcal{L}^2(\mathbb{R})$ the space of all such functions, whose basis functions are characterized by probabilist's Hermite polynomials

$$\text{He}_n(x) := (-1)^k e^{x^2/2} \frac{d^k}{dx^k} e^{-x^2/2}.$$

The first three Hermite polynomials are $\text{He}_0(x) = 1, \text{He}_1(x) = x, \text{He}_2(x) = x^2 - 1$. Let $\text{he}_k(x) := \frac{\text{He}_k(x)}{\sqrt{k!}}$ denote the normalized Hermite polynomials, which form a complete orthonormal basis in $\mathcal{L}^2(\mathbb{R})$. Thus, the Hermite expansion of a

function $f \in \mathcal{L}^2(\mathbb{R})$ is given by

$$f(x) = \sum_{k=1}^{\infty} \mu_{k-1}(f)\mathrm{he}_{k-1}(x), \ \mu_{k-1}(f) = \mathbb{E}_{X \sim \mathcal{N}(0,1)}\{f(X)\mathrm{he}_{k-1}(X)\}.$$

Indeed, Hermite polynomials can be equivalently defined by identifying $e^{xt-t^2/2} = \sum_{k=0}^{\infty} \frac{\mathrm{He}_n(x)}{k!}t^k$. We can define the generalized Hermite polynomials $H_n(x, y)$ as those that satisfy

$$e^{xt+yt^2} = \sum_{k=0}^{\infty} \frac{t^k}{k!} H_n(x, y).$$

In the case when $y = -\frac{1}{2}$, we have $H_n(x, -1/2) = \mathrm{He}_n(x)$. Generally, the generalized Hermite polynomials $H_n(x, y)$ is related to the Hermite canonical form $\mathrm{He}_n(x)$ as the following way:

$$H_n(x, y) = \mathrm{i}^k (2y)^{\frac{n}{2}} \mathrm{He}_n \left( \frac{x}{\mathrm{i}\sqrt{2y}} \right). \tag{55}$$

We are now ready to introduce the eigenvalue system of the operator $T : \mathcal{L}^2(W) \to \mathcal{L}^2(X)$ derived by (Carrasco et al., 2007).

**Lemma E.4** ((Carrasco et al., 2007)). *Let $T : \mathcal{L}^2(W) \to \mathcal{L}^2(X), Tf = \mathbb{E}\{f(W)|X = \cdot\}$, where $\mathcal{L}^2(\cdot)$ is square integrable space with respect to the standard Gaussian measure, i.e., $(W, X)$ is jointly Gaussian with zero mean, unit variance, and correlation $\rho_{WX}$. We have $T$ is a self-adjoint operator, and the eigenvalue system for $T$ is given by $\varphi_j(w) = \mathrm{he}_j(w), \phi_j(x) = \mathrm{he}_j(x), \lambda_j = \rho_{WX}^j$, where $\rho_{WX}$ is the correlation coefficient between $W$ and $X$ and $\mathrm{he}_j$ is the normalized Hermite polynomials.*

Now we prove the result in example 5.3.

**Example E.5.** Suppose that $X, Y, U, W$ satisfy the linear Gaussian model, *i.e.* $U = \varepsilon_U, X = 2U + \varepsilon_X, W = -2U + \varepsilon_W, Y = X + U + \gamma_W W + \varepsilon_Y$, where $\varepsilon_U, \varepsilon_Y, \varepsilon_W, \varepsilon_X \sim \mathcal{N}(0, 1)$. The integral equation (1) has a solution if and only if $\gamma_W > \frac{-15+36\sqrt{5}}{72+16\sqrt{5}} \approx 0.61$. Besides, when $\gamma_W > \frac{-15+36\sqrt{5}}{72+16\sqrt{5}}$, the series $\sum_{n=1}^{\infty} \lambda_n^{-2} |\langle p(y|x), \phi_n \rangle|^2$ converges.

*Proof.* We first show that even under $\mathbb{H}_1$, the integral equation $p(y|x) = \int h(w, y)p(w|x)dw$ has a solution when the coefficient $\gamma_W$ is large enough. Specifically, since $X$ and $W$ are normalized, based on the data generation structure, we have

$$\begin{pmatrix} U \\ X \\ W \\ Y \end{pmatrix} \sim \mathcal{N} \left\{ \begin{pmatrix} 0 \\ 0 \\ 0 \\ 0 \end{pmatrix}, \begin{pmatrix} 1 & \frac{2}{\sqrt{5}} & -\frac{2}{\sqrt{5}} & -\frac{2}{\sqrt{5}}\gamma_W + 1 + \frac{2}{\sqrt{5}} \\ \frac{2}{\sqrt{5}} & 1 & -\frac{4}{5} & -\frac{4}{5}\gamma_W + 1 + \frac{2}{\sqrt{5}} \\ -\frac{2}{\sqrt{5}} & -\frac{4}{5} & 1 & \gamma_W - \frac{2}{5}(2 + \sqrt{5}), \\ -\frac{2}{\sqrt{5}}\gamma_W + 1 + \frac{2}{\sqrt{5}} & -\frac{4}{5}\gamma_W + 1 + \frac{2}{\sqrt{5}} & \gamma_W - \frac{4}{5} - \frac{2}{\sqrt{5}} & \gamma_W^2 - \frac{4}{5}(2 + \sqrt{5})\gamma_W + 3 + \frac{4}{\sqrt{5}} \end{pmatrix} \right\}.$$

We can therefore derive the explicit form of the conditional distributions $p(w|x)$ and $p(y|x)$:

$$W|X = x \sim \mathcal{N} \left\{ \mu_W + \frac{\mathrm{Cov}(W, X)}{\mathrm{Var}(X)}(x - \mu_X), \mathrm{Var}(W) \left( 1 - \frac{\mathrm{Cov}^2(W, X)}{\mathrm{Var}(X) \cdot \mathrm{Var}(W)} \right) \right\}$$

$$\sim \mathcal{N} \left( -\frac{4}{5}x, \frac{9}{25} \right);$$

$$Y|X = x \sim \mathcal{N} \left\{ \mu_Y + \frac{\mathrm{Cov}(Y, X)}{\mathrm{Var}(X)}(x - \mu_X), \mathrm{Var}(Y) \left( 1 - \frac{\mathrm{Cov}^2(Y, X)}{\mathrm{Var}(X) \cdot \mathrm{Var}(Y)} \right) \right\},$$

$$\sim \mathcal{N} \left\{ \left( -\frac{4}{5}\gamma_W + 1 + \frac{2}{\sqrt{5}} \right) x, \frac{9}{25}\gamma_w^2 - \frac{4}{5\sqrt{5}}\gamma_W + \frac{6}{5} \right\}. \tag{56}$$

By applying Lemma B.4, the solution of (1) is given by:

$$h(w, y) = \frac{1}{\sqrt{\frac{9+2\sqrt{5}}{10}\gamma_W + \frac{3}{16} - \frac{9\sqrt{5}}{20}}} \phi \left\{ \frac{y - \left( \gamma_W + \frac{2\sqrt{5}-5}{4} \right) w}{\frac{9+2\sqrt{5}}{10}\gamma_W + \frac{3}{16} - \frac{9\sqrt{5}}{20}} \right\}. \tag{57}$$

For $h(w, y)$ to be meaningful, we need $\frac{9+2\sqrt{5}}{10}\gamma_W + \frac{3}{16} - \frac{9\sqrt{5}}{20} > 0$, which implies $\gamma_W > \frac{-15+36\sqrt{5}}{72+16\sqrt{5}} \approx 0.61$.

Next, we need to verify the conditions for the series in Picard's theorem B.2, which requires proving that $\sum_{n=0}^{+\infty} \lambda_n^{-2}|\langle f, \phi_n\rangle|^2 < +\infty$ for the singular system $(\lambda_n, \varphi_n, \phi_n)_{n=1}^{+\infty}$ associated with the compact operator $Th = f$. In our data generation process, operator $T : \mathcal{L}^2(W, \gamma) \to \mathcal{L}^2(X, \gamma)$ satisfies $Th = \mathbb{E}\{h(W, y)|x\} = p(y|x)$ for all $(x, y)$ and is characterized by the integral kernel (12). Thus, by Lemma E.4, we have $T : \mathcal{L}^2(W, \gamma) \to \mathcal{L}^2(X, \gamma)$ is a self-adjoint operator and the eigenvalue system of operator $T$ is given by $\varphi_j(w) = \text{he}_j(w), \phi_j(x) = \text{he}_j(x), \lambda_j = \rho_{WX}^j$, where $\rho_{WX}$ is the correlation coefficient between $W$ and $X$ and $\text{he}_j$ is the normalized Hermite polynomials. Thus, we show that as long as $\gamma_W > \frac{-15+36\sqrt{5}}{72+16\sqrt{5}}$, the following series converges, which can explain why the solution may exist under $\mathbb{H}_1$:

$$\sum_{n=0}^{\infty} \frac{|\langle p(y|x), \text{he}_n(x)\rangle|^2}{\rho_{WX}^{2n}}.$$

Let

$$I_n := \langle p(y|x), \text{he}_n(x)\rangle = \frac{1}{\sqrt{2\pi}} \int p(y|x)\text{he}_n(x)e^{-x^2/2}dx = \frac{1}{\sqrt{2\pi n!}} \int p(y|x)\text{He}_n(x)e^{-x^2/2}dx.$$

We will use the generating function method to derive the analytic form of integrals. By equation (6) in (Babusci et al., 2012), we have:

$$\int H_n(ax + b, m)e^{-cx^2+\alpha x}dx = \frac{\sqrt{\pi}}{\sqrt{c}} \exp\left(\frac{\alpha^2}{4c}\right) H_n\left(b + \frac{\alpha a}{2c}, m + \frac{a^2}{4c}\right).$$

Let $\mu := -\frac{4}{5}\gamma_W + 1 + \frac{2}{\sqrt{5}}$ and $\sigma^2 := \frac{9}{25}\gamma_w^2 - \frac{4}{5\sqrt{5}}\gamma_W + \frac{6}{5}$. By (56), we can obtain

$$I_n = \frac{1}{\sqrt{2\pi n!}} \int \frac{1}{\sqrt{2\pi\sigma^2}} e^{-\frac{(y-\mu x)^2}{2\sigma^2}} H_n(x)e^{-x^2/2}dx \overset{(1)}{=} \frac{e^{-\frac{y^2}{2\sigma^2}}}{2\pi\sqrt{\sigma^2 n!}} \int e^{-\left(\frac{\sigma^2+\mu^2}{2\sigma^2}\right)x^2 + \frac{\mu y}{\sigma^2}x} H_n\left(x, -1/2\right)dx,$$

where (1) follows from $H_n(x, -1/2) = \text{He}_n(x)$ by (55). By taking $a = 1, b = 0, c = \frac{\sigma^2+\mu^2}{2\sigma^2}, \alpha = \frac{\mu y}{\sigma^2}$ and $m = -1/2$, we have:

$$I_n = \frac{1}{\sqrt{2\pi n!(\sigma^2 + \mu^2)}} e^{\frac{-y^2}{2(\sigma^2+\mu^2)}} H_n\left\{\frac{\mu y}{\sigma^2 + \mu^2}, -\frac{\mu^2}{2(\sigma^2 + \mu^2)}\right\}.$$

We consider the case when $\mu \neq 0$ and $\mu = 0$.

**1. When $\mu \neq 0$.**

By (55) we have

$$I_n = \frac{1}{\sqrt{2\pi n!(\sigma^2 + \mu^2)}} e^{\frac{-y^2}{2(\sigma^2+\mu^2)}} \left\{i^n \left(-\frac{\mu^2}{\sigma^2 + \mu^2}\right)^{\frac{n}{2}} \text{He}_n\left(\frac{\frac{\mu y}{\sigma^2+\mu^2}}{i\sqrt{-\frac{\mu^2}{\sigma^2+\mu^2}}}\right)\right\}$$

$$= \frac{1}{\sqrt{2\pi n!(\sigma^2 + \mu^2)}} \left\{(-1)^n \left(\frac{\mu^2}{\sigma^2 + \mu^2}\right)^{\frac{n}{2}} e^{\frac{-y^2}{2(\sigma^2+\mu^2)}} \text{He}_n\left(-\frac{\mu}{|\mu|\sqrt{\sigma^2 + \mu^2}}y\right)\right\}.$$

Therefore, the series can be written as

$$\sum_{n=0}^{\infty} \left(\frac{I_n}{\rho_{WX}^n}\right)^2 = \sum_{n=0}^{\infty} \left\{\frac{(-1)^n}{\rho_{WX}^n\sqrt{2\pi(\sigma^2 + \mu^2)}} \left(\frac{\mu^2}{\sigma^2 + \mu^2}\right)^{\frac{n}{2}} e^{\frac{-y^2}{2(\sigma^2+\mu^2)}}\right\}^2 \left\{\frac{1}{\sqrt{n!}}\text{He}_n\left(-\frac{\mu}{|\mu|\sqrt{\sigma^2 + \mu^2}}y\right)\right\}^2. \quad (58)$$

According to equation (18.15.27) in (Olver, 2010), it is known that for fixed $M = 0, 1, 2...,$

$$H_n(x) = \lambda_n e^{\frac{1}{2}x^2}\left[\sum_{m=0}^{M-1} \frac{u_m(x)\cos\omega_{n,m}(x)}{(2n+1)^{\frac{1}{2}m}} + O\left\{\frac{1}{(2n+1)^{\frac{1}{2}M}}\right\}\right].$$

where $H_n(x) := (-1)^n e^{x^2} \frac{d^n}{dx^n} e^{-x^2}$ is physicist's Hermite polynomials, the coefficients $u_m(x)$ are polynomials in $x$, $u_0(x) = 1, u_1(x) = 1/6x^3, \omega_{n,m}(x) = (2n+1)^{\frac{1}{2}}x - \frac{1}{2}(m+n)\pi$ and

$$\lambda_n = \begin{cases} \Gamma(n+1)/\Gamma\left(\frac{1}{2}n+1\right) & \text{if } n \text{ is even,} \\ \Gamma(n+2)/\left\{(2n+1)^{\frac{1}{2}}\Gamma\left(\frac{1}{2}n+\frac{3}{2}\right)\right\} & \text{if } n \text{ is odd.} \end{cases}$$

By taking $M = 1$ and $\text{He}_n(x) = 2^{-\frac{n}{2}}H_n\left(\frac{x}{\sqrt{2}}\right)$, we have

$$\text{He}_n(x) = 2^{-\frac{n}{2}}\lambda_n e^{\frac{1}{4}x^2}\left[\cos\omega_{n,m}(\frac{x}{\sqrt{2}}) + O\left\{\frac{1}{(2n+1)^{\frac{1}{2}M}}\right\}\right] \le 2^{-\frac{n}{2}}\lambda_n e^{\frac{1}{4}x^2} + 2^{-\frac{n}{2}}\lambda_n e^{\frac{1}{4}x^2} \times O\left\{\frac{1}{(2n+1)^{\frac{1}{2}}}\right\}$$

$$\le 2^{-\frac{n}{2}+1}\lambda_n e^{\frac{1}{4}x^2}.$$

Since each term in the series $\sum_{n=0}^{\infty} I_n^2$ is positive, its odd-term series and even-term series are both positive term series. If both subseries converge, then $\sum_{n=0}^{\infty} I_n^2$ also converges.

**(i). Even term series.** By Stirling's approximation $n! \sim \sqrt{2\pi n}\left(\frac{n}{e}\right)^n$, we have:

$$\frac{\Gamma(n+1)}{\Gamma(\frac{n}{2}+1)} \sim \sqrt{2\pi n}\left(\frac{n}{e}\right)^n\left(2\pi\frac{n}{2}\right)^{-\frac{1}{2}}\left(\frac{n}{2e}\right)^{-\frac{n}{2}} = 2^{\frac{n+1}{2}}\left(\frac{n}{e}\right)^{\frac{n}{2}}.$$

Thus, we can obtain $\text{He}_n(x) \lesssim 2^{\frac{1}{2}+1}\left(\frac{n}{e}\right)^{\frac{n}{2}}e^{\frac{x^2}{4}}$ for large values of $n$ even. Then for even $n$, we have

$$\frac{\text{He}_n(x)\text{He}_n(x)}{n!} \lesssim \frac{2^3}{\sqrt{2\pi n}}e^{x^2/2}.$$

Thus, we have

$$\left(\frac{I_n}{\rho_{WX}^n}\right)^2 \lesssim \left[\frac{1}{2\pi(\sigma^2+\mu^2)}\left\{\frac{\mu^2}{\rho_{WX}^2(\sigma^2+\mu^2)}\right\}^n \frac{2^3}{\sqrt{2\pi n}}\right]e^{-\frac{1}{2(\sigma^2+\mu^2)}y^2} := J_n.$$

We only need to prove the convergence of the series $\sum_{n \text{ is even}} J_n$. By the ratio test, we can obtain:

$$\lim_{n\to\infty}\left|\frac{J_{n+2}}{J_n}\right| = \lim_{n\to\infty}\left|\frac{\left[\frac{1}{2\pi(\sigma^2+\mu^2)}\left\{\frac{\mu^2}{\rho_{WX}^2(\sigma^2+\mu^2)}\right\}^{n+2}\frac{2^3}{\sqrt{2\pi(n+2)}}\right]e^{-\frac{1}{2(\sigma^2+\mu^2)}y^2}}{\left[\frac{1}{2\pi(\sigma^2+\mu^2)}\left\{\frac{\mu^2}{\rho_{WX}^2(\sigma^2+\mu^2)}\right\}^n\frac{2^3}{\sqrt{2\pi n}}\right]e^{-\frac{1}{2(\sigma^2+\mu^2)}y^2}}\right|$$

$$= \lim_{n\to\infty}\left\{\frac{\mu^2}{\rho_{WX}^2(\sigma^2+\mu^2)}\right\}^2\sqrt{\frac{n}{n+2}}.$$

This series absolutely converges if and only if

$$\frac{\mu^2}{\rho_{WX}^2(\sigma^2+\mu^2)} < 1,$$

which holds if and only if $\gamma_W > \frac{-15+36\sqrt{5}}{72+16\sqrt{5}} \approx 0.61$ by taking $\mu = -\frac{4}{5}\gamma_W + 1 + \frac{2}{\sqrt{5}}, \sigma^2 = \frac{9}{25}\gamma_w^2 - \frac{4}{5\sqrt{5}}\gamma_W + \frac{6}{5}$ and $\rho_{WX} = -\frac{4}{5}$.

**(ii). Odd term series.** By Stirling's approximation $n! \sim \sqrt{2\pi n}\left(\frac{n}{e}\right)^n$, we have

$$\frac{\Gamma(n+2)}{\sqrt{2n+1}\cdot\Gamma(\frac{n}{2}+\frac{3}{2})} \sim \frac{\sqrt{2\pi(n+1)}}{\sqrt{2n+1}}\left(\frac{n+1}{e}\right)^{n+1}\left(2\pi\frac{n+1}{2}\right)^{-\frac{1}{2}}\left(\frac{n+1}{2e}\right)^{-\frac{n+1}{2}} = \sqrt{\frac{2}{2n+1}}2^{\frac{n+1}{2}}\left(\frac{n+1}{e}\right)^{\frac{n+1}{2}}.$$

Thus, we can obtain $\text{He}_n(x) \lesssim \sqrt{\frac{2}{2n+1}}2^{\frac{3}{2}}\left(\frac{n+1}{e}\right)^{\frac{n+1}{2}}e^{\frac{x^2}{4}}$ for large values of $n$ odd. Then for odd $n$, we have

$$(n+1)\frac{\text{He}_n(x)\text{He}_n(x)}{(n+1)!} \lesssim \frac{16(n+1)}{e(2n+1)}\frac{e^{x^2/2}}{\sqrt{2\pi(n+1)}}.$$

Thus, we have

$$\left(\frac{I_n}{\rho_{WX}^n}\right)^2 \lesssim \sum_{n=0}^{\infty}\left\{\frac{1}{2\pi(\sigma^2+\mu^2)}\left(\frac{\mu^2}{\rho_{WX}^2(\sigma^2+\mu^2)}\right)^2\frac{16(n+1)}{2n+1}\frac{1}{\sqrt{2\pi(n+1)}}\right\}e^{-\frac{y^2}{2(\sigma^2+\mu^2)}} := J_n.$$

Again, we only need to show the convergence for the series $\sum_{n \text{ is odd}} J_n$. By the ratio test, we can obtain:

$$\lim_{n\to\infty}\left|\frac{J_{n+2}}{J_n}\right| = \lim_{n\to\infty}\left|\frac{\left\{\frac{1}{2\pi(\sigma^2+\mu^2)}\left(\frac{\mu^2}{\rho_{WX}^2(\sigma^2+\mu^2)}\right)^{n+2}\frac{16(n+3)}{2n+5}\frac{1}{\sqrt{2\pi(n+3)}}\right\}e^{-\frac{y^2}{2(\sigma^2+\mu^2)}}}{\left\{\frac{1}{2\pi(\sigma^2+\mu^2)}\left(\frac{\mu^2}{\rho_{WX}^2(\sigma^2+\mu^2)}\right)^{n}\frac{16(n+1)}{2n+1}\frac{1}{\sqrt{2\pi(n+1)}}\right\}e^{-\frac{y^2}{2(\sigma^2+\mu^2)}}}\right|$$

$$= \lim_{n\to\infty}\left|\left\{\frac{\mu^2}{\rho_{WX}^2(\sigma^2+\mu^2)}\right\}^2\sqrt{\frac{(2n+1)^2(n+3)}{(2n+5)^2(n+1)}}\right|.$$

We can verify $\sqrt{\frac{(2n+1)^2(n+3)}{(2n+5)^2(n+1)}} = \sqrt{\frac{4n^3+16n^2+13n+3}{4n^3+24n^2+45n+25}} < 1$ for positive integers. Thus, this sub-series of odd terms absolutely converges if and only if

$$\frac{\mu^2}{\rho_{WX}^2(\sigma^2+\mu^2)} < 1,$$

which holds if and only if $\rho_{WX} = -\frac{4}{5}$, we have $\gamma_W > \frac{-15+36\sqrt{5}}{72+16\sqrt{5}} \approx 0.61$, by taking $\mu = -\frac{4}{5}\gamma_W + 1 + \frac{2}{\sqrt{5}}$, $\sigma^2 = \frac{9}{25}\gamma_w^2 - \frac{4}{5\sqrt{5}}\gamma_W + \frac{6}{5}$.

Combining two results, we know that when $\gamma_W > \frac{-15+36\sqrt{5}}{72+16\sqrt{5}} \approx 0.61$, the original series converges.

**2. When $\mu = 0$.**

Since $\mu = 0$, we have $\gamma_W = \frac{5+2\sqrt{5}}{4} > \frac{-15+36\sqrt{5}}{72+16\sqrt{5}}$. Thus, the distribution of $p(y|x)$ becomes

$$Y|X = x \sim \mathcal{N}\left\{0, \frac{1}{16}(29+4\sqrt{5})\right\}.$$

Thus, if we define $\sigma_{\text{con}}^2 := \frac{1}{16}(29+4\sqrt{5})$, we have

$$I_n = \frac{1}{\sqrt{2\pi n!}}\int\frac{1}{\sqrt{2\pi\sigma_{\text{con}}^2}}e^{-\frac{y^2}{2\sigma_{\text{con}}^2}}H_n(x)e^{-x^2/2}dx = \frac{e^{-\frac{y^2}{2\sigma_{\text{con}}^2}}}{2\pi\sqrt{\sigma_{\text{con}}^2 n!}}\int \text{He}_n(x)e^{-x^2/2}dx.$$

Following Lemma 2.6 of (Davis, 2024), the integral of the stretched Hermite polynomial $S_n = \frac{1}{\sqrt{2\pi}}\int \text{He}_n(\gamma x)e^{-x^2/2}dx$ is only non-zero for even $n$ and has the value $S_n = (n-1)!!(\gamma^2-1)^{n/2}$. We use the above results and take $\gamma = 1$, we have $I_n = 0$ for $n \geq 1$. Thus, the series is:

$$\sum_{n=0}^{\infty}\left(\frac{I_n}{\rho_{WX}^n}\right)^2 = \left(\frac{I_0}{\rho_{WX}^0}\right)^2 = (I_0)^2 \overset{(1)}{=} \frac{e^{-\frac{y^2}{\sigma_{\text{con}}^2}}}{4\pi^2\sigma_{\text{con}}^2}\left(\int e^{-x^2/2}dx\right)^2 \overset{(2)}{=} \frac{e^{-\frac{y^2}{\sigma_{\text{con}}^2}}}{4\pi^2\sigma_{\text{con}}^2}2\pi = \frac{1}{2\pi\sigma_{\text{con}}^2}e^{-\frac{y^2}{\sigma_{\text{con}}^2}} < \infty.$$

where (1) follows from $\text{He}_0(x) = 1$ and (2) follows from $\int e^{-x^2/2}dx = \sqrt{2\pi}$. $\qquad\square$

### E.4. Proof of asymptotic properties with two proxies

**Assumption E.6.** We assume $\mathbb{E}_X\{m(X, Z, s)|W\}$ and $\mathbb{E}_X\{|m(X, Z, s)|^2|W\}$ are uniformly bounded for all $s$.

**Assumption E.7.** For any $s, t \in \mathcal{T}$, $\mathbb{E}\{U(W, Y, t)^4|X\} < \infty$ and $\mathbb{E}(|m(X, Z, s) - \{A(A^*A)^{-1}g_s\}(X)|^4) < \infty$, where $g_s(\cdot) = \mathbb{E}[m(X, Z, s)\phi_W(W)](\cdot)$.

**Assumption E.8.** Let $(\lambda_j, \varphi_j, \phi_j)_j$ be the singular value decomposition of the operator $A$ described in section C. Then we assume: (a). For some $\eta \geq 2$, $\sum_j \lambda_j^{-2\eta}|\langle g_s, \varphi_j\rangle_{\mathcal{H}_W}|^2 < \infty$; (b) For some $\theta \geq 2$, $\sum_j \lambda_j^{-2\theta}|\langle H_t^0, \varphi_j\rangle_{\mathcal{H}_W}|^2 < \infty$.

**Theorem 6.6.** *Denote* $\overline{\eta}_{s,t}(O,x) := U(W,Y,t)\left[\{m(Z,x,s)-\{A(A^*A)^{-1}g_s\}(X,x)\right]$, *where* $g_s(\cdot,x) := \mathbb{E}\{m(Z,x,s)\phi_W(W)\}(\cdot)$ *and* $O := (W,Z,Y,X)$. *Suppose assumptions in Theorem 4.12 hold. If Asm. 6.1-6.2, and E.6-E.8 hold, under* $\mathbb{H}_0$ *we have, (i).* $T_n^{(Z)}(s,t)$ *converges weakly to* $\mathbb{G}(s,t)$ *s.t.* $\iint |\mathbb{G}(s,t)|^2 d\mu(s)d\mu(t) < \infty$, *where* $\mathbb{G}(s,t)$ *is a mean-zero Gaussian process with covariance* $\Sigma\{(s,t),(s',t')\} = \mathbb{E}\{\overline{\eta}_{s,t}(O,x)\overline{\eta}_{s',t'}(O',x)\}$, *where* $O' := (W',Z',Y',X')$ *is an independent copy of* $O$; *(ii).* $\Delta_{\varphi,m}^{(Z)}$ *converges weakly to* $\max_{t\in\mathcal{T}} \int_{\mathcal{T}} |\mathbb{G}(s,t)|^2 d\mu(s)$.

*Proof.* We need to replace the weight function $m(x,s)$ with $m(Z,x,s)$ over $Z$. By (36), we have

$$T_n^{(Z)}(s,t) = \sqrt{n}\mathbb{P}_n\{U(W,Y,t)m(Z,x,s)\} + \text{(Expected risk difference)} + \text{(Empirical process)}.$$

By Propositions D.3, for fixed $x$, we can obtain the empirical process

$$\sqrt{n}(\mathbb{P}_n - \mathbb{P})[\{H^0(W,t)-\widehat{H}^\lambda(W,t)\}m(Z,x,s)] = o_p(1).$$

By Propositions D.4, for fixed $x$, we have

$$\sqrt{n}\mathbb{P}\left\{(H^0(W,t)-\widehat{H}^\lambda(W,t))m(Z,x,s)\right\} = -\frac{1}{\sqrt{n}}\sum_{i=1}^n U(w_i,y_i,t)\{A(A^*A)^{-1}g_s\}(x_i) + o_p(1).$$

Therefore, combining all the inequalities, we have

$$T_n^{(Z)}(s,t) = \frac{1}{\sqrt{n}}\sum_{i=1}^n U(w_i,y_i,t)\left[m(x,z_i,s)-\left\{A(A^*A)^{-1}g_s\right\}(x_i,x)\right] + o_p(1).$$

Next, we apply Lemma D.15 to $\left\{U(w_i,y_i,t)\left[m(x,z_i,s)-\left\{A(A^*A)^{-1}g_s\right\}(x_i,x)\right]\right\}_i$ to obtain $T_n^{(Z)}(s,t)$ converges weakly to $\mathbb{G}(s,t)$ in $\mathcal{L}^2\{\mathcal{T}\times\mathcal{T},\mu\times\mu\}$, where $\mathbb{G}(s,t)$ is a Gaussian process with zero-mean and covariance:

$$\Sigma\{(s,t),(s',t')\} = \mathbb{E}\{\overline{\eta}_{s,t}(W,Z,Y,X,x)\overline{\eta}_{s,t}(W',Z',Y',X',x)\}.$$

To show $\mathbb{G}(s,t)$ is zero-mean, noted that

$$\begin{aligned}
&\mathbb{E}\left[U(W,Y,t)\left\{m(Z,x,s)-\{A(A^*A)^{-1}g_s\}(X,x)\right]\right] \\
&= \mathbb{E}\left[U(W,Y,t)m(Z,x,s)\right] - \mathbb{E}\left[U(W,Y,t)\{A(A^*A)^{-1}g_s\}(X,x)\right] \\
&= \mathbb{E}\left[m(Z,x,s)\mathbb{E}\left[U(W,Y,t)|Z,x\right]\right] - \mathbb{E}\left[\mathbb{E}\left[U(W,Y,t)|X\right]\{A(A^*A)^{-1}g_s\}(X,x)\right] \\
&= 0,
\end{aligned}$$

where the last equation follows from (11) and (4).

Besides, by assumption E.7, we have $\text{Var}(U(w_i,y_i,t)[m(x,z_i,s)-\{A(A^*A)^{-1}g_s\}(x_i,x)]) = \mathbb{E}(U(w_i,y_i,t)[m(x,z_i,s)-\{A(A^*A)^{-1}g_s\}(x_i,x)])^2 < \infty$ for any $(x,s,t)$. Therefore, by continuous mapping theorem, we have $\Delta_{\varphi,m}^{(Z)}$ converges weakly to $\max_{t\in\mathcal{T}} \int |\mathbb{G}(s,t)|^2 d\mu(s)$. $\square$

For power analysis, we define the global alternative $\mathbb{H}_1^{\text{fix}}$ and $\mathbb{H}_{1n}^\alpha$ $(0 < \alpha \le 1/2)$ of (11), in terms of $\mathbb{E}\{\varphi(Y,t) - H(W,t)|Z,x\}$ for fixed $x$.

$$\mathbb{H}_1^{\text{fix}} : \mathbb{E}\{\varphi(Y,t) - H(W,t)|Z,x\} \ne 0 \text{ for some } t \in \mathcal{T},$$

for any $H \in \mathcal{H}_W$. For the local alternative $\mathbb{H}_{1n}^\alpha$, there exists $H^0 \in \mathcal{H}_W$, such that

$$\mathbb{H}_1^\alpha : \mathbb{E}\{\varphi(Y,t)|Z,x\} = \mathbb{E}\{H^0(W,t)|Z,x\} + \frac{r(Z,x,t)}{n^\alpha}, \ \forall t,$$

where $0 < \alpha \le 1/2$, and for any $H$, $\frac{r(Z,x,t)}{n^\alpha}$ cannot be written as $\mathbb{E}\{H(\cdot,t) - H^0(\cdot,t)|Z,x\}$ for some $t$.

**Theorem E.9.** *Suppose assumptions in Theorem 6.6 hold. Besides, we assume* $\mathbb{E}\{r(Z,x,t)^4\} < \infty$ *for fixed $x$ and any $t$. Then, we have:*

(i) ***Global alternative.*** $\lim_{n\to\infty} \max_{t\in\mathcal{T}} |T_n^{(Z)}(s,t)| = \infty$ *for almost all $s$ under $\mathbb{H}_1^{\text{fix}}$.*

(ii) ***Local alternative*** *($\alpha < 1/2$).* $\lim_{n\to\infty} \max_{t\in\mathcal{T}} |T_n^{(Z)}(s,t)| = \infty$ *for almost all $s$ under $\mathbb{H}_{1n}^{\alpha}$.*

(iii) ***Local alternative*** *($\alpha = 1/2$).* $T_n^{(Z)}(s,t)$ *converges weakly to* $\mathbb{G}(s,t)+\mu(Z,X,x,s,t)$ *in* $\mathcal{L}^2\{\mathcal{T}\times\mathcal{T}, \mu\times\mu\}$ *under $\mathbb{H}_{1n}^{\alpha}$,* *where $\mathbb{G}(s,t)$ is defined in Theorem 6.6 and $\mu(Z,X,x,s,t) := \mathbb{E}\left[\{r(Z,x,t) - (A^*A)^{-1}A^*r(Z,x,t)\}m(Z,x,s)\right]$.*

*Proof.* The proof is similar to that of theorem 4.14, with the weight function $m(X,s)$ replaced with $m(Z,x,s)$. $\square$

# F. Additional experiments

In this section, we evaluate our two-proxy procedure to a nonlinear setting, where $W \to Y$ and both $W$ and $Z$ begin available. Similar to Fig. 5, the two-proxy method outperforms the single-proxy approach by leveraging information provided from the additional proxy, *i.e.*, NCE.

**Data generation.** We generate $U$ following a normal distribution with mean 0 and variance 1, denoted by $U \sim \mathcal{N}(0, 1)$. Similarly, we simulate $W = -2\sin(U) + \varepsilon_W$ and $Z = 2\sin(U) + \varepsilon_Z$. The treatment assignment mechanism follows the generation process: $X = 2\sin(U) + \varepsilon_X$. Under the alternative hypothesis $X \not\perp\!\!\!\perp Y|U$, the outcome is generated from $Y = X + \sin(U) + 2W^2 + \varepsilon_Y$; while under the null hypothesis $X \perp\!\!\!\perp Y|U$, the outcome is generated from $Y = \sin(U) + 2W^2 + \varepsilon_Y$. In both hypotheses, the noise terms $\varepsilon_X, \varepsilon_Z, \varepsilon_W, \varepsilon_Y$ are independently drawn from a standard normal distribution. We repeat the process 20 times, where at each time we generate 100 replications under $\mathbb{H}_0$ and $\mathbb{H}_1$.

**Type-I error and power.** The average results are presented in Fig. 6. As observed, while our single-proxy procedure effectively controls the type-I error, it exhibits low power in identifying causal relationships. In contrast, by incorporating additional information from the NCE, the power improves significantly, demonstrating its efficacy in learning causal connections.

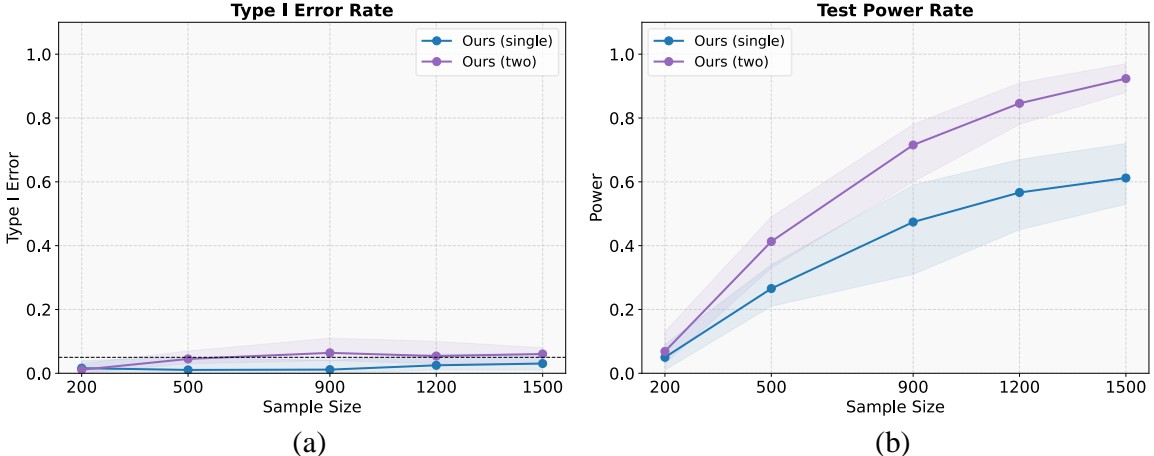

*Figure 6.* Type-I error rate (left) and power rate (right) of our procedure and baselines in the nonlinear setting with two proxies.

