# OpenReview forum: "Bivariate Causal Discovery with Proxy Variables: Integral Solving and Beyond"
_ICML.cc/2025/Conference — ICML 2025 poster_

### Official Review · Reviewer_g83S · 2025-03-12

**Overall Recommendation:** 4

**Summary:**

This paper aims to test the conditional independence relation 𝑋⊥𝑌∣𝑈 in the presence of an unobserved latent confounder U. Due to the unobservability of U, this conditional independent relation cannot be directly tested. The authors show that conditional independence can still be assessed using a proxy variable Z or W, thereby transforming the hypothesis test into solving an integral equation and evaluating whether the residuals are zero. The paper provides theoretical results on the solution of such an integral equation and an asymptotic analysis of the corresponding test statistic. These theoretical contributions extend previous discretization-based methods to a more robust approach.

**Claims And Evidence:**

The claims are generally well-supported by theoretical analysis.

**Essential References Not Discussed:**

The author gives a well-reviewed on the related literature.

**Experimental Designs Or Analyses:**

The experimental design and analyses are generally sound.

**Methods And Evaluation Criteria:**

The proposed method is reasonable and well-motivated.

**Other Comments Or Suggestions:**

NAN

**Other Strengths And Weaknesses:**

Strengths:

1. This paper addresses an important yet challenging problem in causal discovery involving latent confounders.

2. The proposed method is supported by rigorous theoretical analysis.

3. Experimental results demonstrate that the proposed approach outperforms state-of-the-art methods.


Weaknesses:

1. The method relies on a partially known prior structure, specifically that U causes both X and Y, and that Z and W serve as proxy variables for U. This assumption may be restrictive, as in practice, we may not have direct knowledge of the existence of U or whether Z and W are valid proxies.

2. The asymptotic properties of the proposed method depend on a set of assumptions, which may limit its applicability in real-world scenarios.

**Questions For Authors:**

1. The uniqueness of the solution (not only the existence) to the integral equation seems crucial for the hypothesis test. Could the authors provide an intuitive explanation or illustration of its role?

2. If U is a vector rather than a single variable, does the proposed theoretical result still hold?

3. The theoretical contribution is intriguing! I am curious whether the proposed method can be viewed as a generalization of Tetrad constraints when considering Z and  W as proxy variables.

**Relation To Broader Scientific Literature:**

The proposed method contributes to the problem of CI testing in  the presence of latent confounders.

**Theoretical Claims:**

The theoretical claims are clearly presented, and the proofs are well-structured.

---

> ### Author Rebuttal · Authors · 2025-03-31
>
> We appreciate your efforts and valuable suggestions in reviewing our paper. We address your concerns below.
>
> **Q1.** The method relies on a partially known prior structure, specifically that $U$ causes both $X$ and $Y$, and that $Z$ and $W$ serve as proxy variables for $U$. This assumption may be restrictive, as in practice, we may not have direct knowledge of the existence of $U$ or whether $Z$ and $W$ are valid proxies.
>
> **A.** This is a standard setting in proximal causal inference. Indeed, since the observational causal inference/discovery typically involves unobserved measurements, it is essential to introduce substitution variables, such as instrumental variables and proxy variables, as considered here. In many scenarios, these assumptions naturally hold and such proxy variables are easy to obtain, for example, they can be noisy measurements of the confounders [1] or as descendants in time-series data [2].
>
> **Q2.** The asymptotic properties of the proposed method depend on a set of assumptions, which may limit its applicability in real-world scenarios.
>
> **A.** The most key assumption is completeness, which is standard in proximal causal inference and can easily hold in our scenario. Others are regularity conditions that are also standard for kernel regression.
>
> **Q3.** The uniqueness of the solution (not only the existence) to the integral equation seems crucial for the hypothesis test. Could the authors provide an intuitive explanation or illustration of its role?
>
> **A.** As claimed in lines 168-176 (right column), uniqueness is not required in our procedure, since our goal is to determine whether a solution exists. Among all solutions to the integral equation, our estimate converges to the least-norm solution (Lemma D.19 in Appendix D.5).
>
> **Q4.** If $U$ is a vector rather than a single variable, does the proposed theoretical result still hold?
>
> **A.** Our procedure remains valid even when $U$ is a high-dimensional variable, but requires the proxy variable $W$ to satisfy the completeness condition (Assumption 4.1). It implies that the dimension of $W$ is greater than that of $U$. When this holds, the completeness is easy to hold.
>
> **Q5.** The theoretical contribution is intriguing! I am curious whether the proposed method can be viewed as a generalization of Tetrad constraints when considering $Z$ and $W$ as proxy variables.
>
> **A.** Thank you for your insightful question. Our methods can be viewed as a generalization of Tetrad constraints. Specifically, the tetrad constraint was introduced to test whether $X, Y, Z, W$ are conditionally independent given $U$, under linear models. Specifically, the constraint is:
> $$
> \mathrm{cov}( X,Y ) \mathrm{cov}( Z,W )= \mathrm{cov}( X,Z ) \mathrm{cov}( Y,W )= \mathrm{cov}( X,W ) \mathrm{cov}( Y,Z ).
> $$
> When proxies $W,Z$ are assumed to satisfy $W \perp Y|U$ and $Z \perp X|U$, the constraint degenerates to:
> $$
> \mathrm{cov}( X,Y ) \mathrm{cov}( Z,W )= \mathrm{cov}( X,W ) \mathrm{cov}( Y,Z ),
> $$
> and it can be used to test $\mathbb{H}_0: X \perp Y|U$. In contrast, our constraint for testing $\mathbb{H}_0$ is formulated by integral equations (1) and (10). Under the linear Gaussian setting with standard normal exogenous noise, our constraint gives rise to the one in \#. Specifically, suppose $U \sim \mathcal{N}(0,1)$, $W=\mu_UU +\varepsilon_W$, $Z=\beta_UU +\varepsilon_Z$, $X=\alpha_UU +\varepsilon_X$, and $Y=\gamma_UU+\varepsilon_Y$, where  $\varepsilon_W,\varepsilon_Z, \varepsilon_X, \varepsilon_Y$ are standard normal. We have $h(W) = \frac{\gamma_U}{\mu_U}W$. Since $\mathrm{cov}( Z,W ) = \beta_U \mu_U$ and $\mathrm{cov}( Z,Y ) = \beta_U \gamma_U$, we have $\mathrm{cov}( X,Y ) \mathrm{cov}( Z,W )=\gamma_U \alpha_U \mu_U \beta_U = \mathrm{cov}( X,W ) \mathrm{cov}( Y,Z )$.
>
> [1] Kuroki, M. and Pearl, J. Measurement bias and effect restoration in causal inference. Biometrika, 101(2):423-437, 2014.
>
> [2] Liu, M., Sun, X., Hu, L., and Wang, Y. Causal discovery from subsampled time series with proxy variables. Advances in neural information processing systems, 2023.

---

> > ### Comment · Reviewer_g83S · 2025-04-04
> >
> > Thank you for the author response—The rebuttal addressed my questions.
> >
> >
> > Maybe one can establish the graphical implication of your "generalized Tetrad constraints", which would be interesting in the causal discovery task. I believe this is an important contribution of this work.
> >
> > Overall, I will keep my score leaning towards acceptance.

---

### Official Review · Reviewer_nuHH · 2025-03-13

**Overall Recommendation:** 2

**Summary:**

The paper concerns bivariate causal discovery in the presence of unobserved confounders with the assumption that certain "proxy variables" are observed. Existing literature translates the absence of a direct effect of treatment on outcome to the existence of a solution to a certain integral equation and proposes parametric approaches that involve discretization for continuous variables given that a proxy variable that affects the outcome is observed. This paper proposes a nonparametric approach to test the existence. While the integral equation is only a necessary condition (under certain assumptions) for the absence of a direct effect, the paper shows that the integral equation is not sufficient in the linear models case. By assuming that another proxy variable that affects the treatment, they show that a condition involving the existence of a solution for a modified integral equation is necessary and sufficient for the absence of direct effect.

**Claims And Evidence:**

Most of the theoretical claims are clear in the statement but I have not checked the proofs. There are some unsubstantiated claims.
1) For e.g. in Section 3, lines 125-127 state that the proposed test is sample-efficient compared to discretization. I don't see any evidence in the form of any comparison for this claim to hold.
2) "Besides our power approximates to one as n increases" (line 422) - Figure 3b does not show the same.

**Essential References Not Discussed:**

None that I am familiar with.

**Experimental Designs Or Analyses:**

1) I am not sure why the orange line in Figure 3 does not achieve type I error level in the paper's experiments whereas it does in Liu et. al. 2023. Apart from the sqrt function, the function classes and the noise types are the same in both papers.

**Methods And Evaluation Criteria:**

The paper draws heavily from previous experimental setups in Liu et. al. 2023 which makes comparison easier. However, it is to be noted that these are synthetic datasets and perhaps evaluations on real-world datasets could make the paper stronger.

**Other Comments Or Suggestions:**

1. Line 101 - "allowing test" should be "allowing to test"?
2. Please specify $\mathbb{H}_1$ in line 118.

**Other Strengths And Weaknesses:**

The paper finds an appropriate set of well-motivated assumptions under which a non-parametric test is constructed for bivariate causal discovery assuming access to a proxy variable that affects the outcome. It seems believable that under the nonparametric setup, nonidentifiability is an issue which the paper shows is true even under a linear model. Despite this a condition is proposed under which identifiability is restored. The paper combines multiple ideas that come together for the non-parametric test such as using the characteristic function instead of first order moments like previous work, using a weight function to convert the conditional restriction to an unconditional one.
Regarding weaknesses, 1) I think the writing of the paper is a major weakness and needs to be improved greatly. Currently, the flow of ideas is abrupt and a few claims/choices don't seem to be well-explained in the text. For e.g., Pg 4, the part before "Equivalently speaking" does not seem to be relevant to the final estimate $\hat{H}^{\lambda}(w,t)$, the regularization in (5) is not motivated, the term 'bridge function' is never defined in line 172  (see questions for more such instances).
2) Through each successive approximation, the null hypothesis is being enlarged. While, the paper provides a power analysis, it was not clear to me what whether there are any theoretical guarantees for a type I error control with finite-sample guarantees which makes an earlier claim about sample-efficiency untestable.
3) Some claims don't seem to be supported by evidence (see claims and evidence section)

**Questions For Authors:**

1) Does (4) hold if (1) does not hold under the assumptions in Thm 4.5?
2) Like assumption 4.1, does assumption 6.1 say that all variability in U is captured by X? That seems like a very strong assumption and makes a "confounder" immaterial.
3) "There exists $H^0(w,t)$" (line 234). Could you explain why this is a "there exist" statement. The motivation behind this form of the local alternative seems unclear.
4) It is not clear to me why in Line 129, you need h(w,y) to be square-integrable? The text prior does not seem to motivate this requriement.
5) What does "feasibility of solutions" mean in Line 204? Feasibility in what sense?

**Relation To Broader Scientific Literature:**

This work continues the recent thread of research in proximal causal inference where the exchangeability assumption is relaxed by assuming that certain proxy variables are observed. The absence of a direct effect is then translated in terms of conditions on joint distribution of the treatment, outcome and proxy variables. Recently, Miao et. al. 2023 introduced one such condition that concerned the existence of a solution of an integral equation but used a parametric approach to testing. This work introduces a nonparametric approach.

**Theoretical Claims:**

No

---

> ### Author Rebuttal · Authors · 2025-03-31
>
> We appreciate your efforts and suggestions in reviewing our paper. We address your concerns below.
>
> **Q1.** About the type I error level in Liu's paper.
>
> **A.** Liu's type-I error control requires a Lipschitz smooth function, but sqrt does not satisfy this. To make it hold, we conduct experiments with the sigmoid function. As shown, Liu's method approximates the type-I error as $n$ increases. It is also seen that our method has better sample efficiency than theirs.
>
> |Sample|200|400|600|800|1000|1200|
> |-|-|-|-|-|-|-|
> |Our|0.06|0.05|0.05|0.06|0.05|0.05|
> |Liu|0.15|0.13|0.09|0.09|0.08|0.08|
>
> **Q2.** No evidence to support sample efficiency.
>
> **A.** The table above shows our method's superior sample efficiency in type-I error control. As claimed in lines 107-109, this is because Liu's method requires the bin number to go to infinity to make the discretization error vanish.
>
> **Q3.** The power does not approach to one in Fig. 3b.
>
> **A.** We further conduct experiments as $n$ increases. Combined with Fig. 3b, they show a clear trend of power approaching one. When $n=4000$, the power is 0.97.
>
> |Sample|2000|3000|4000|
> |-|-|-|-|
> |Power|0.89|0.94|0.97|
>
> **Q4.** About definitions and claims.
>
> **A.** **About $\hat{H}^\lambda(w,t)$, its relation to the paragraph before "equivalent speaking".** As mentioned in lines 201-211, $\hat{H}^\lambda(w,t)$ is the minimizer of (5), which is designed to solve the least-norm solution as introduced in lines 176-181. From lines 182-208, we explain how the risk (5) is derived. We first introduce the risk in lines 192-193, followed by its equivalent form in lines 197-198. Indeed, the risk (5) is its empirical version, augmented with a regularization term. Additionally, we provide a more detailed explanation in Appx. C.4-C.5 due to space limitations. We will offer further clarifications in the updated version.
>
> **About the motivation of Tikhonov regularization.** The regularization provides a way to solve ill-posedness problems and is commonly adopted in the literature of kernel regression. We have explained in more detail in Appx. C.5.
>
> **About the bridge function.**  The bridge function is the solution to the integral equation used in proximal causal inference, serving a similar role to $H$ in (4). It is a fundamental concept in proximal causal inference, and we have provided references in lines 167-168 due to space limits.
>
> **Q5.** About power analysis after successive approximation.
>
> **A.** Our power analysis focuses on alternatives in (4), which are defined based on characteristic restrictions. By utilizing the characteristic function, our approach demonstrates significantly better power compared to first-order moment methods in the literature (Fig. 4).
>
> **Q6.** Does (4) hold if (1) does not hold under the assumptions in Thm 4.5?
>
> **A.** We are sorry that we're unsure if we fully understand the question. Under assumptions in Thm. 4.5, both (1) and (4) should hold, as (1) is the conclusion of Thm. 4.5, and (4) naturally follows if (1) holds. We would appreciate your clarification if we've misunderstood.
>
> **Q7.** Does assump. 6.1 say that all variability in U is captured by X? That seems to make a "confounder" immaterial.
>
> **A.** Yes, it means variability in $U$ is captured by $X$. However, we want to clarify that this assumption is easy to hold and will not make $U$ immaterial.
>
> First, the assumption applies to a wide range of models, as long as the dimension of $X$ is greater than that of $U$. Under this condition, [Andrews et al. 2017] demonstrated that completeness generically holds. Even if this condition holds, the confounding problem still matters as it also causes bias since completeness does not imply that $p(x)$ would determine (up to transformation) $p(x,u)$ (this is known as the ``equivalence condition" in Miao et. al. 2023) and hence the distribution of $p(u)$. In this regard, we still cannot determine whether $p(y|do(x)) \neq p(y|x)$ when $U$ is unobserved.
>
> **Q8.** About "there exist" statement in $H^0(w,t)$" (line 234).
>
> **A.** Local alternative refers to alternatives that are very close to the null hypothesis. That means, it adds a small deviation $r(X)/n^\alpha (\alpha > 0)$ to some $H^0$. To ensure that it does not degenerate to $\mathbb{H}_0$, we require $r(X)/n^\alpha$ cannot be written as $E(H-H^0|X)$ for any $H$, as claimed in lines 239-241.
>
> **Q9.** Why need h(w,y) to be square-integrable?
>
> **A.** If $h(w, y)$ is not square-integrable, it cannot be solved using regression methods, as regression approaches minimize the squared loss that involves the second-order moment of the solution. Besides, our method and analysis are built upon the completeness assumption, which is fundamental in proximal causal inference and is imposed on the square-integrable function class.
>
> **Q10.** What does "feasibility of solutions" mean in Line 204?
>
> **A.** It refers to the capability of regression methods to achieve consistency for integral solving (Lines 200-203).

---

### Official Review · Reviewer_EyGH · 2025-03-14

**Overall Recommendation:** 4

**Summary:**

The paper proposes a nonparametric procedure for bivariate causal discovery for determining $X \perp Y \mid U$, where $U$ is an unmeasured confounder of $X$ and $Y$. It introduces the Proxy Maximum Characteristic Restriction (PMCR) method to solve an integral equation where a proxy variable is available to determine $X \perp Y \mid U$. The paper also introduces a second proxy (NCE) and derive an additional integral equation for identifiability. Theoretical results establish asymptotic validity and power, while experiments demonstrate improved type-I error control and power compared to baselines in synthetic settings.

**Claims And Evidence:**

Yes, claims are backed up by theoretical statements (e.g., Thm. 4.5) and experimental results.

**Essential References Not Discussed:**

None to the best of my knowledge.

**Experimental Designs Or Analyses:**

Yes, the ones in the main text.

**Methods And Evaluation Criteria:**

Synthetic datasets generated under varying conditions (with/without direct proxy effects) are used to evaluate type-I error and power. The evaluation criteria are appropriate for assessing causal discovery methods in the bivariate setting.

**Other Comments Or Suggestions:**

1. In line 289 it should read "bootstrapped statistic."
2.  The numbering of lemmas in the main text and the appendix is inconsistent (e.g., Theorem 4.5 in the main text is Theorem B.4 in the appendix). Consider using a LaTeX package like `restatable` to maintain consistent numbering.

**Other Strengths And Weaknesses:**

**Strengths**
- Comprehensive theoretical analysis with detailed asymptotic results.
- Clear identification and resolution of non-identifiability issues via a second proxy.
- Thorough experimental results (albeit all synthetic).

**Weaknesses**
- The paper is very dense and details are hard to grasp without reading parts of the 45+ page appendix!

**Questions For Authors:**

1. How sensitive is PMCR to kernel and bandwidth choices?
2. What is the computational efficiency of the proposed approach? How well does it scale to larger datasets and/or high-dimensional variables?
3. Are the non-identifiability results (Prop. 5.1) generalizable beyond linear models?

**Relation To Broader Scientific Literature:**

It extends prior work by addressing the sample efficiency limitations of discretization and by clarifying non-identifiability conditions when using a single proxy.

**Theoretical Claims:**

Yes, I checked the proof of Theorem 4.5 in Appendix B.

---

> ### Author Rebuttal · Authors · 2025-03-31
>
> Thank you for the positive assessment and valuable suggestions about our paper. We address your questions below. We will correct typos and make the number of lemmas consistent in the updated version.
>
> **Q1.** How sensitive is PMCR to kernel and bandwidth choices?
>
> **A.** We would like to clarify that PMCR requires kernels to be bounded, continuous, and integrally strictly positive definite (Assumptions C.2 and C.3). Typical types of kernels satisfying these conditions include Gaussian RBF and Laplacian kernels. We adopt the Gaussian kernel because it is commonly used in the literature. Its bandwidth is typically initialized via the median distance heuristic, ensuring robust performance.
>
> **Q2.** What is the computational efficiency of the proposed approach? How well does it scale to larger datasets and/or high-dimensional variables?
>
> **A.** PMCR has a time complexity of $ O(pn^3) $, where $ n $ is the sample size and $ p $ is the dimension of $ W $. The complexity is linearly proportional to $ p $ because, for multivariate $ W $, the kernel can be constructed as the product of scalar kernels applied to each input dimension $ w $, i.e., $ k(w, w') = \prod_{j=1}^{p} k(w[:, j], w[:, j]') $, where $ w[:, j] $ denotes the $ j $-th feature. The complexity with respect to $ n $ can be reduced using standard Cholesky techniques when computing the inverse. We record the running time of a single trial as $ n $ varies (when $ W $ is univariate). As shown, the actual running time increases much more slowly than $ O(n^3) $.
>
> | Sample                | 200  | 400  | 600  | 800  | 1000 | 1200 |
> |----------------------|------|------|------|------|------|------|
> | Computational time (s) | 0.39 | 0.97 | 2.64 | 5.62 | 9.65 | 14.84 |
>
> **Q3.** Are the non-identifiability results (Prop. 5.1) generalizable beyond linear models?
>
> **A.** We believe that nonlinear models also suffer from the non-identifiability issue when $W$ has a strong effect on $Y$, but the theoretical analysis is challenging as there is no closed-form solution for non-linear models.
>
> To empirically verify this point, we consider a non-linear setting: $U \sim \mathcal{N}(0,1)$, $W=U +\varepsilon_W$, $X=U +\varepsilon_X$, and $Y=X^2+\gamma_W W^2+U+\varepsilon_Y$, where $\varepsilon_W, \varepsilon_X, \varepsilon_Y$ are standard normal. We generate $n=800$ samples and record the average power over 20 times. We can see that the power decreases as $\gamma_W$ increases, which suggests the difficulty of identifying the causal relation when $W$ strongly affects $Y$.
>
> | $\gamma_W$ | 0    | 0.5  | 1    | 3    | 5    | 7    | 9    | 11   |
> |---------------|------|------|------|------|------|------|------|------|
> | Power        | 0.85 | 0.88 | 0.85 | 0.68 | 0.58 | 0.32 | 0.14 | 0.07 |

---

> > ### Comment · Reviewer_EyGH · 2025-04-02
> >
> > I thank the authors for their response and for adequately addressing my questions and comments.

---

### Decision · Program_Chairs · 2025-05-01

**Decision:**

Accept (poster)

**Comment:**

This paper presents a nonparametric method, Proxy Maximum Characteristic Restriction (PMCR), for bivariate causal discovery in the presence of unobserved confounders. By leveraging proxy variables, it reformulates the causal inference problem as an integral equation and proposes a second proxy-based condition for identifiability.

Pros:

+ Strong theoretical foundation with rigorous asymptotic analysis and well-structured proofs.

+ Innovative use of multiple proxy variables to resolve non-identifiability issues.

+ Experimental results show performance improvements over baseline methods in synthetic settings.

Cons:

+ The writing is dense and very hard to follow, requiring reference to a lengthy appendix.

+ Assumptions such as access to valid proxy variables and certain model structures may limit real-world applicability.